# Deep autoencoder for interpretable tissue-adaptive deconvolution and cell-type-specific gene analysis

Yanshuo Chen[1,2,9], Yixuan Wang [1,3,9], Yuelong Chen[4,5], Yuqi Cheng [6], Yumeng Wei[1], Yunxiang Li[1], Jiuming Wang[1], Yingying Wei[7], Ting-Fung Chan [4,5] ✉ & Yu Li [1,8] ✉

Single-cell RNA-sequencing has become a powerful tool to study biologically significant characteristics at explicitly high resolution. However, its application on emerging data is currently limited by its intrinsic techniques. Here, we introduce Tissue-AdaPtive autoEncoder (TAPE), a deep learning method connecting bulk RNA-seq and single-cell RNA-seq to achieve precise deconvolution in a short time. By constructing an interpretable decoder and training under a unique scheme, TAPE can predict cell-type fractions and cell-type-specific gene expression tissue-adaptively. Compared with popular methods on several datasets, TAPE has a better overall performance and comparable accuracy at cell type level. Additionally, it is more robust among different cell types, faster, and sensitive to provide biologically meaningful predictions. Moreover, through the analysis of clinical data, TAPE shows its ability to predict cell-type-specific gene expression profiles with biological significance. We believe that TAPE will enable and accelerate the precise analysis of high-throughput clinical data in a wide range.

Bulk RNA sequencing (RNA-seq), a widely used high-throughput sequencing technique, provides a powerful tool to investigate transcriptome variation of biological events[1]. RNA-seq measures averaged expression levels, which gives a macro atlas of large samples from transcription levels without cell-specific information. However, it is also important to study the cellular composition and proportion of the sample in some cases, especially in a system with cellular development and proliferation (e.g., cancer)[2,3].

Recently, single-cell RNA sequencing (scRNA-seq) has given unprecedented opportunities to identify and analyze the cell heterogeneity of complex tissues[4]. While scRNA-seq provides impressive resolution in cell granularity, it is still costly and vulnerable to noise,

prohibiting sequencing the large-scale samples[5,6]. To overcome these obstacles, we may combine the abundant bulk RNA-seq data with the scRNA-seq data, performing cell-type deconvolution from the bulk RNA-seq samples with reference to a small scRNA-seq dataset.

Many single-cell profile-assisted algorithms have sprung up to dissect bulk RNA-seq data in recent years. The existing methods can be roughly divided into two categories: statistical learning-based and deep learning-based methods. Based on traditional regression models like non-negative least squares (NNLS) and support vector regression (SVR), a series of methods like CIBERSORT (CS)[7], MuSiC[8], CIBERSORTx (CSx)[9], Bisque[10], DWLS[11], RNA-Sieve[12], and BLADE[13] have been developed. All these tools need a pre-selected cell-type-specific gene

[1]Department of Computer Science and Engineering, CUHK, Hong Kong SAR, China. [2]School of Life Sciences, Tsinghua University, 100084 Beijing, China. [3]Department of Mathematics, HIT, Weihai 264209, China. [4]School of Life Sciences, CUHK, Hong Kong SAR, China. [5]State Key Laboratory of Agrobiotechnology, The Chinese University of Hong Kong, Hong Kong SAR, China. [6]Weill Cornell Graduate School of Medical Sciences, Weill Cornell Medicine, New York, NY 10065, USA. [7]Department of Statistics, The Chinese University of Hong Kong, Hong Kong SAR, China. [8]The CUHK Shenzhen Research Institute, Hi-Tech Park, Nanshan, Shenzhen 518057, China. [9]These authors contributed equally: Yanshuo Chen, Yixuan Wang. ✉e-mail: tf.chan@cuhk.edu.hk; liyu@c-se.cuhk.edu.hk

expression profile (GEP) or allocating different weights to different genes based on statistic value (e.g., mean and variance). In contrast, Scaden[14], a deep learning method, utilizes simulated bulk data for training without relying on a pre-defined GEP, and it can automatically extract features from GEP. Despite this progress, these methods ignore the running time cost, especially regarding the growing demands of dealing with big datasets. Moreover, except CSx, other methods, like Scaden, cannot predict the crucial cell-type-specific gene expression. This limitation leads to the poor interpretability of Scaden and other methods. Even for CSx, it requires multiple samples (>15) to purify expression[9].

To overcome these limitations, we propose an accurate, efficient, and interpretable deep-learning algorithm, Tissue-AdaPtive auto-Encoder (TAPE), using deep neural networks (DNNs). The basic idea is that the encoder can learn higher-order latent representations and decoder can realize the interpretability of the output in the framework of autoencoder. Moreover, we introduce a new training scheme named adaptive training to optimize the GEP tissue-adaptively. Empirically, our method could achieve a better overall performance than previous state-of-the-art methods. When evaluated on cell-type level, TAPE has the best performance of MAE, and comparable CCC with relatively small variance on real datasets (Supplementary Table 8). To demonstrate the clinical application of TAPE, we use three datasets to show that TAPE is sensitive to biological changes. To be specific, TAPE is the only method that predicts the increasing tendency of monocytes-to-lymphocytes ratio (MLR) value which is suitable within the clinical report (0.29–0.88)[15] in the COVID-19 peripheral blood mononuclear cells (PBMC) dataset. Furthermore, TAPE can predict cell-type-specific GEP tissue-adaptively with the minimum input data size requirement, inferring cell-type-specific GEP for only one sample. Thus, TAPE can provide a valuable reference for biologists to more conveniently investigate differentially expressed genes (DEGs). More importantly, TAPE can predict DEGs even when these genes are not signature genes, which would probably fail with previous methods. We further combine TAPE with single-sample gene set enrichment analysis (ssGSEA)[16] on the virus-infected PBMC dataset to prove its capability of analyzing cross-viral functional differences among cells.

In this work, we build TAPE to precisely predict cellular fractions and cell-type-specific gene expression. The novelty lies in adopting the architecture of autoencoder as well as introducing a new training scheme at the adaptive stage. Compared with the state-of-art methods, TAPE shows a competitive performance and has almost the fastest processing speed on benchmarking datasets. In addition, TAPE predicts the cell-type-specific gene expression tissue-adaptively, allowing the dissection of bulk gene expression into different cell types and discovering potential differential gene expressions among cell types.

## Results
### Method overview
As shown in Fig. 1, the basic architecture of our method is a DNN-based autoencoder (AE), taking bulk GEPs as input and outputting cell-type proportions and cell-type-specific GEPs. There are three stages of using TAPE. The first stage is to create training data through simulation. Simulated bulk data is the sum of selected single-cell GEPs with the pre-defined cell fractions and the total cell numbers, where the single-cell profile and the real bulk profile should come from the same tissue. The next is the training stage. We want to train the model to output the proper cell fractions after the encoder and use the cell fractions to reconstruct the bulk profile. More than only using the reconstruction loss in the classic AE model, we try to minimize the mean absolute error (MAE) between the ground truth and the predicted cell fractions to make it supervised. When the model is required to predict the cell fractions and the cell-type-specific GEPs on the real bulk data, it enters the adaptive stage. In this process, inspired by the classic AE's training process, we only use real bulk data to train the model in an unsupervised manner. More specifically, the model is iteratively greedily optimized on the decoder and the encoder. That is, it would not optimize the parameters of the encoder until it achieves the temporally best parameters on the decoder (see Table 1). The intuition is that training the encoder and decoder separately can directly lead to the adaption to the new coming bulk data. As for the decoder, we require it to reconstruct the real bulk data and maintain the concordance with itself, while the encoder is required to predict the proper cell fractions, which should be similar to the primary prediction after the training stage. Since we require the decoder to output the reconstructed bulk gene expression based on the cell fractions, the parameters of the decoder are the cell-type-specific GEPs, so we could directly output those parameters as the GEPs after the adaptive stage. See more details in the Methods part.

### Performance evaluation on pseudo-bulk data
Since a real bulk dataset with its corresponding cell type fractions assessed by traditional experimental methods (e.g., flow cytometry) is rare, and it is hard to analyze how the batch effect would affect deconvolution performance, it is necessary to conduct a pseudo-bulk test for an initial estimation. The pseudo-bulk data are generated in silico from single-cell GEPs with ground truth (pre-defined cell type proportions). That is, pseudo-bulk data are the summation of many single-cell profiles. To make this pseudo-bulk test as difficult as the real bulk test instead of trivial linear regression task, we added Gaussian noise[17] (0.01 times random value generated from a Gaussian distribution with gene expression mean and variance for each gene) and randomly masked 20% genes for each pseudo-bulk sample. The single-cell profiles are from *Tabular Muris*[18], a cell atlas for mouse with two different sequencing techniques, 10X-seq (UMI-based method) and Smart-seq (counts-based method). This cell atlas is a good resource for us to simulate the batch effect. Thus, in the following experiments, we used one protocol's single-cell data as the reference to predict another protocol's pseudo-bulk data. Here we only selected three tissues/organs from *Tabular Muris* because they have the largest number of shared cell types across different protocols in all the tissues/organs. Specifically, "Limb Muscle" has 6 cell types, "Marrow" has 7 cell types, and "Lung" has 9 cell types. To fully exploit the advantages of pseudo-bulk data, we defined three deconvolution scenarios: "normal", "rare", and "similar". For the "normal" scenario, all the cell type proportions are randomly generated, while in the "rare" scenario, some cell types' fractions are set below 3%. To be specific, skeletal muscle satellite cells and endothelial cells are set to be rare cell types in "Limb Muscle"; monocyte and hematopoietic precursor cells are set to be rare cell types in "Marrow"; T cells, natural killer cells and ciliated columnar cells of tracheobronchial trees are set to be rare cell types in "Lung". In the "similar" task, we only used "Marrow" because there are two similar subtypes of B cell in it: "late-pro B cell" and "immature B cell". Here, we expect that if we delete one kind of B cell from the single-cell reference, the predicted fraction of the other type of B cell would still be similar to the summation of the two kinds of B cell. That is, we expect the method could correctly transfer the weight of one kind of B cell to another. Performance was evaluated by MAE and Lin's concordance correlation coefficient (CCC)[19] between the prediction and the ground truth for each cell type. More details of the simulation process, dataset, and metrics for evaluation are in the Methods section.

In the "normal" scenario, we find that DWLS achieves the best performance on both metrics (Fig. 2c), and TAPE is comparable to DWLS. We also notice that deep learning methods like Scaden and TAPE are more robust than statistical methods, which lead to the smallest performance variance among all the cell types. In the "rare" scenario, we only display the metrics for pre-defined rare cell types. The results show that all the methods can not result in a satisfying concordance between prediction and ground truth in this scenario (Fig. 2c). Interestingly, although the CCC values are pretty low with

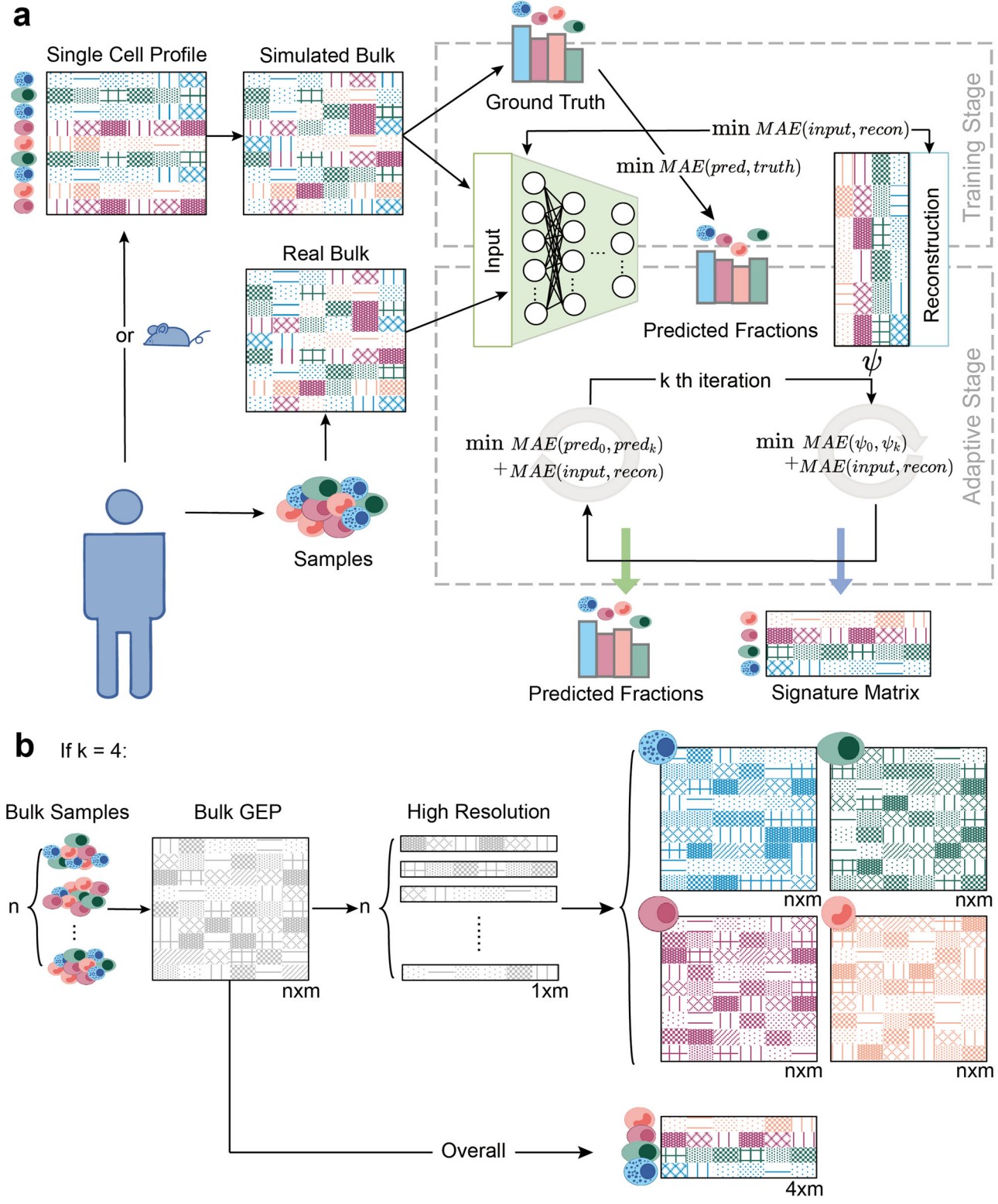

**Fig. 1 | TAPE workflow and clarification of adaptive stage. a** TAPE takes scRNA-seq data from human or mouse and RNA-seq data from the homologous tissue as input, then performs the deconvolution as well as the prediction of cell-type-specific GEPs via a training stage and an adaptive stage. **b** Generation of the cell-type-specific GEPs has two separate modes. The first is the "high-resolution" mode: TAPE takes the RNA-seq data from one sample at a time as input and outputs the adapted cell-type-specific signature matrix for each sample. The second is the "overall" mode: TAPE takes all the RNA-seq data at one time as input and outputs one signature matrix adapted to all samples. $n$ is the number of samples, $m$ is the number of genes, $k$ is the number of cell types.

those methods, their MAEs are comparable to those in the "normal" scenario, which indicates that those methods can predict a value near ground truth but are not correlated with each other. Though TAPE is not the best algorithm in this scenario, its performance is comparable

to DWLS, which focuses on rare cell types. As for the "similar" scenario, we investigate the performance on two kinds of B cells in the "normal" scenario and what would happen if we delete one cell type from the reference. The results show that TAPE is the most robust algorithm and

**Table 1 | Adaptive training procedure**

input  : Encoder parameters $E$ and decoder parameters $D$ from the initial training stage,
            GEPs of bulk RNA-seq $\mathbf{B}$ of size $n \times m$, step number $\alpha$, max iteration $\beta$
output: signature matrix $\mathbf{S}$ of size $k \times m$,
            predicted fractions $\mathbf{X}$ of size $n \times k$,
            training loss $L$

1  $\tilde{\mathbf{S}}_0, \tilde{\mathbf{X}}_0 \leftarrow \text{model}(\mathbf{B})$;
2  **for** $k \leftarrow 1$ **to** $\beta$ **do**
3 **for** $i \leftarrow 1$ **to** $\alpha$ **do**
4 $\tilde{\mathbf{B}}, \mathbf{X} \leftarrow \text{model}(\mathbf{B})$;
5 $L \leftarrow \text{MAE}(\tilde{\mathbf{B}}, \mathbf{B}) + \text{MAE}(\mathbf{S}, \tilde{\mathbf{S}}_0)$;
6 $D \leftarrow D - \frac{\partial L}{\partial D}$;
7 **end**
8 **for** $j \leftarrow 1$ **to** $\alpha$ **do**
9 $\tilde{\mathbf{B}}, \mathbf{S} \leftarrow \text{model}(\mathbf{B})$;
10 $L \leftarrow \text{MAE}(\tilde{\mathbf{B}}, \mathbf{B}) + \text{MAE}(\mathbf{X}, \tilde{\mathbf{X}}_0)$;
11 $E \leftarrow E - \frac{\partial L}{\partial E}$;
12 **end**
13 **end**
14 $\mathbf{S}, \mathbf{X} \leftarrow \text{model}(\mathbf{B})$;

can distinguish cell subtypes when both kinds of the B cell are in the reference (Fig. 2c, "similar distinguishment"). Moreover, TAPE can transfer one B cell's proportion to another if this kind of B cell is missing from the reference (Fig. 2c, "similar transferring"). Meanwhile, we find that deep learning methods are robust for the cross-protocol prediction, while the performance of some statistical methods will drop if the reference and pseudo-bulk data type (UMI to counts or counts to UMI) are exchanged (Supplementary Figs. 1 and 2). When we compare the performance among algorithms for each tissue/organ, we find that TAPE outperforms the other methods for "Limb Muscle" and "Marrow". Nevertheless, we notice that the performances of deep learning methods such as TAPE and Scaden drop for "Lung" as compared to MuSiC and Bisque. One potential reason is that given a fixed sample size, as traditional statistical methods requires fewer parameters, they perform better when the number of cell types increases compared to deep learning methods.

**Accurate and stable deconvolution on real bulk data**
Since previous studies have shown that cell type proportions in single-cell data are not concordant with bulk samples[9], we further evaluated TAPE and the other representative deconvolution methods on real tissue expression datasets with the corresponding ground truth obtained from traditional experimental methods (e.g. flow cytometry). First, we assessed deconvolution performance on two human PBMC bulk RNA-seq datasets, SDY67[20] and the S13 cohort from Monaco et al.[21]. Another PBMC microarray dataset was obtained from Newman et al.[7]. All the ground truth of PBMC datasets was measured by flow cytometry. Second, we deconvolved the ROSMAP human brain RNA-seq dataset[22] with both human brain single-cell RNA-seq and mouse brain single-cell RNA-seq as references. Through immunohistochemistry analysis, the cell-type fractions of 41 samples of the ROSMAP dataset were recently given[23]. Detailed deconvolution software comparison and settings are in the Methods.

Among all the real datasets considered, TAPE achieves the best MAE and the smallest variance (Fig. 2d). For the CCC metric, although other methods like CIBERSORTx and Scaden surpass TAPE, TAPE still shows comparable performance with a relative small variance,

indicating that the prediction performance of TAPE is similar for all the cell types and hence robust. To be specific, for ROSMAP_human dataset, the median CCC of TAPE is the best (0.140). While Scaden achieves the best median CCC of 0.326 and 0.202 on SDY67 dataset and ROSMAP_mouse dataset, and CIBERSORTx achieves the best median CCC on Monaco's PBMC dataset and microarray PBMC dataset. Though TAPE's median CCC on these four datasets is not the highest, the values are comparable with the difference smaller than 0.07. Considering the interquartile range, we can see that the performance of DWLS is close to the best on SDY67 dataset and ROSMAP_mouse dataset. Detailed comparison results are available in Supplementary Table 8. In the benchmarking procedure, we also considered different scenarios as the "similar" scenario, the "missing cell types" scenario and the "unknown cell type" scenario. For the "similar" scenario, we investigated TAPE's performance on distinguishing similar cell types (CD4 T cell and CD8 T cell) in all the three PBMC datasets (Supplementary Fig. 3). The results show that TAPE is the best algorithm and can distinguish them well. Moreover, we test all the methods' ability of deconvolving immune cell subtypes. With 13 defined cell subtypes (Supplementary Table 1), all the methods can not achieve satisfying results (Supplementary Fig. 4, median CCC < 0.1), which clearly shows the common limitation of current methods. For the "missing cell types" scenario, the ROSMAP dataset using mouse brain as reference is a good demonstration. The single-cell dataset of mouse brain has more cell types than the measured bulk ROSMAP dataset (Fig. 2a). So, we directly filtered out extra cell types predicted by these methods and re-scaled the predicted fraction to make the summation is 1. For the "unknown" cell type, it is dependent on the single-cell data. If researchers can not label some cells with proper cell types, they can label them as "unknown" cell type, and then this "unknown" fraction will be considered accordingly when deconvolving bulk data. See more details in the Methods part.

Furthermore, since TAPE and Scaden are both DNN-based methods, we made a head-to-head comparison between them using different random seeds to evaluate TAPE's stability. In practice, the original Scaden program provided by its authors has a very slow simulation speed, so we implemented a PyTorch version of Scaden and

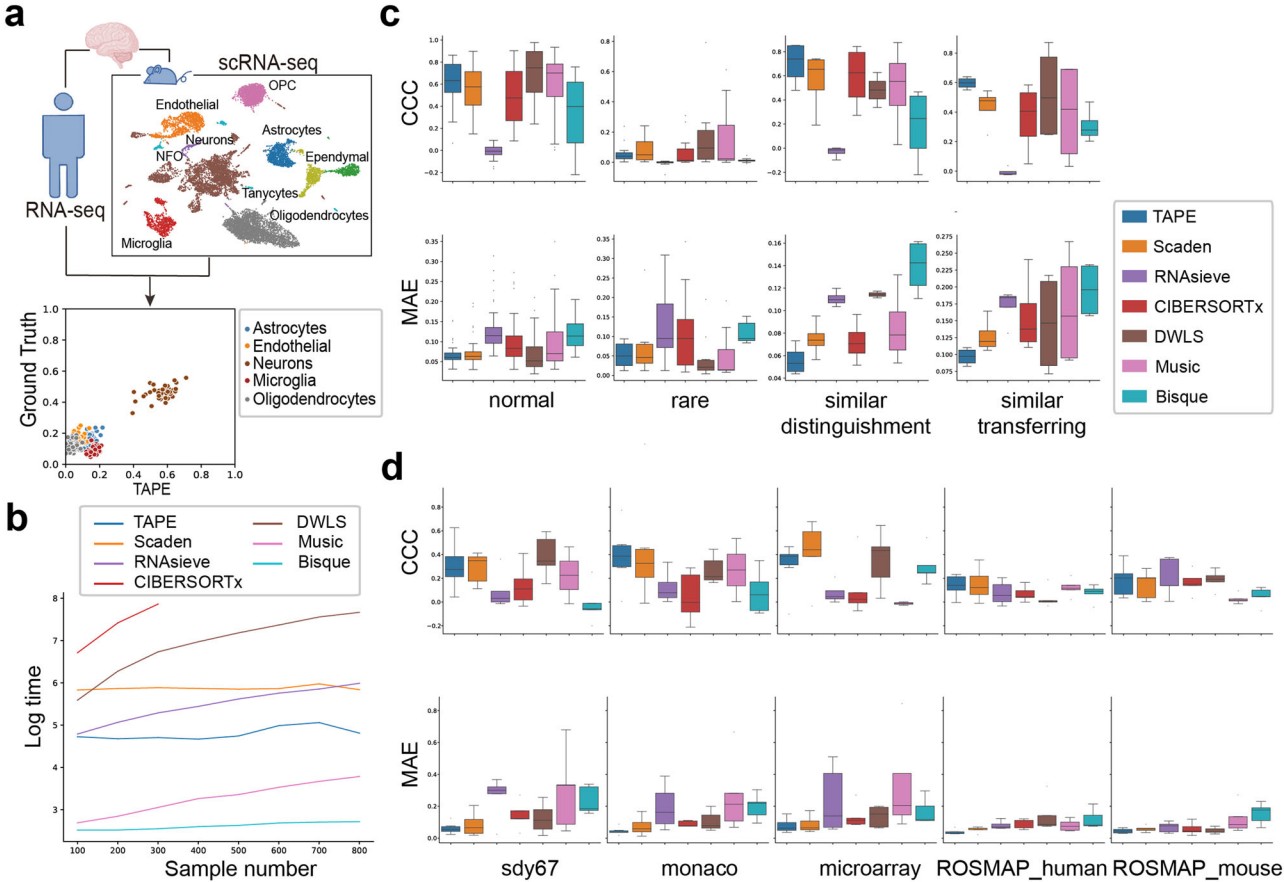

**Fig. 2 | Comparison of deconvolution algorithms on benchmark datasets.**
**a** Deconvolution procedure diagram. Bulk RNA-seq data and single-cell data should come from a homologous tissue. **b** Time complexity analysis of different methods (including pre-processing steps). Time measured by seconds is scaled by Log() to show the differences more clearly. These tests are conducted on the simulated data. The time limit is set to 2500 s. Any longer test was not conducted. **c** Deconvolution results on simulated data. CCC represents the Lin's concordance correlation coefficient, measuring the concordance between the predicted fraction and the ground truth. MAE represents mean absolute error, measuring the accuracy of prediction. Higher CCC and lower MAE are better. Each box contains metric values for all the cell types considered in all the tissues. Different color refers to different methods. Sample size in the four scenarios from left to right is 44, 12, 4, and 4, respectively. Sample size of different method is the same. **d** Deconvolution results on real data. The columns' labels refer to the datasets. CCC and MAE are used as metrics. Sample size of each method on each dataset consistently equals to 5. In **c**, **d**, the boxes represent interquartile range (IQR) while the solid line represents the median. The whiskers extend to points that lie within 1.5 IQRs of the lower and upper quartile, and then observations that fall outside this range are displayed as points independently. Source data are provided as a Source Data file.

used it to test how the random seeds affect deep methods. In Supplementary Fig. 6, the two colors stand for the different methods, and the two coordinates of each dot represent overall MAE and CCC for all cell types respectively. As shown in the figure, the dots of TAPE occupy the upper-right of the figures with low variance, showing TAPE's higher accuracy and stability than Scaden.

## Efficient deconvolution on large cohort RNA-seq data

Besides accuracy and stability, the methods' scalability is also important in practice. Therefore, we evaluated the time consumption of the representative methods mentioned above on the same pseudo-bulk samples from tabular-formed sequencing data to the prediction of cell type fractions (including the step of constructing signature matrix or training data if needed). We ran TAPE, Scaden, RNA-Sieve, DWLS, MuSiC and Bisque on the same workstation with Intel(R) Xeon(R) Gold 6226 CPU @ 2.70GHz, CentOS Linux release 7.9.2009 (Core), Nvidia 3090 GPU. CSx was tested on the web-based application. Detailed implementations are in the Methods.

Among all the methods tested (Fig. 2b), Bisque is the fastest algorithm, and it can deconvolve 800 samples in 15 s. For TAPE, it takes about 120 s in total to construct the training data and train the deep learning model for 5000 iterations. But its inference speed is very fast

and its time complexity is $O(n)$ with a very small coefficient due to the inherent advantage of using deep learning. Thus, TAPE's time consumption would not increase markedly with a larger cohort size. Besides the time complexity, TAPE only needs about 1900MB GPU memory during the training stage. When deconvolving new bulk samples, the memory consumption will increase along with the number of samples, but this increment is really small in practice. Compared with Scaden, another deep learning method, TAPE is faster because of its highly optimized training data simulation procedure and a smaller model size. Of note, the deconvolution step of DWLS is not slow, but the step of constructing signature matrix using MAST[24] is really time- and memory-consuming. As for CIBERSORTx, its slow prediction speed is not justified because its speed is limited by the web server. We would expect a much better performance if users can acquire the source program from the developers. Generally, within the test settings, algorithms that do not require complicated preprocessing steps (Bisque and MuSiC) achieve a better performance on speed.

## Biologically significant deconvolution on clinical RNA-seq data

We further evaluated whether TAPE could predict cell-type proportions consistent with prior clinical knowledge. Here, we selected three datasets with clinical information or related prior knowledge: (1) the

ROSMAP dataset[25] that is obtained from patients with Alzheimer's Disease (AD); (2) the COVID-19 PBMC dataset[26] containing clinical information about different severity of COVID-19 (mild, moderate, and serious) and different stages of patients (treatment stage, convalescence stage, and rehabilitation stage); (3) the cultured pancreatic islet dataset[27] which has RNA-seq data of islets from three different conditions (normal, SARS-CoV-2 infected, and SARS-CoV-2 infected tissue with *Remdesivir* treatment). Detailed information on those datasets is in the Methods.

For the ROSMAP dataset, we used the human brain single-cell profile from Darmanis et al.[28] as a reference. As we know, neuron cell loss is a significant symptom in patients with AD. In the ROSMAP dataset, the Braak stage is given as a measurement of the severity of AD[29]. So we expected neuron fraction would decrease with the development of AD. Additionally, we investigated each sample's Braak stage to the estimated fraction of microglia whose proportion will increase with AD severity and decrease at stage 6 as shown by previous studies[30,31]. The results (Fig. 3a, b) show that TAPE can predict the tendency of neuron loss and have an accurate prediction of microglia activation and deactivation among 532 samples with clinical information. Moreover, according to the immunohistochemistry analysis of 41 AD patients from a previous study, the proportion of neurons or microglia cells ranged from 0.32–0.55 and 0.06–0.12, respectively[23]. Impressively, if we accept the assumption that the cell type proportions' ranges of the 41 patients are the same as those of the 532 patients, only TAPE could predict proportions in this range, which shows the remarkable accuracy of TAPE's prediction.

Next, we used the PBMC data8k[32] dataset as the reference to deconvolve the COVID-19 PBMC dataset. According to existing clinical observations and research, metrics like neutrophil-to-lymphocyte ratio (NLR) and monocyte-to-lymphocyte ratio (MLR) are closely associated with the progression of the COVID-19, thus which are used to indicate its severity clinically[33,34]. In practice, patients with the higher NLR or MLR show a more serious symptom of COVID-19. Since the data we used are obtained from PBMC and do not contain neutrophils, we only tested the correlation between MLR and the severity of COVID-19 patients. The MLR is calculated by the fraction of monocytes divided by the sum of fractions of CD4 T cell, CD8 T cell and B cell. We used the estimated MLR value predicted from different models to compare the tendency between different severity (Fig. 3c). Although Scaden, CIBERSORTx, DWLS and TAPE predict an increasing tendency correctly, after hypothesis tests, only TAPE predicts the increasing tendency of MLR value with statistical significance, and the value range is suitable for the clinical report (0.29–0.88)[15].

To deconvolve the cultured islet dataset, we selected the endocrine cells (alpha, beta, gamma, delta, and epsilon cells) from Baron et al.[35] to generate the training data because pancreatic islet only contains endocrine cells. Since the infection of SARS-CoV-2 usually causes metabolic dysregulation and *Mellitus*[27], we expected a decrease in beta cell fraction in COVID-19 patients. Here, we used the sequencing data of in vitro cultured islets to deconvolve and expected TAPE to predict the decrease of beta cell proportion after infection. Furthermore, the proportion of beta cell should restore after treatment with *Remdesivir*, a very famous antiviral medication used to treat COVID-19 (Fig. 3d). Though Scaden and TAPE can predict both beta cell loss and restoration in this experiment among the three conditions, after one-sided *t*-test, only TAPE's predictions show a statistical significance. The accurate deconvolution results of these controlled experiments demonstrate that TAPE is sensitive to the biological changes in the bulk RNA-seq data and can produce biologically significant results, which are consistent with the previous research and reports. All the clinical deconvolution results show that TAPE's prediction is stable, with potential clinical applications for disease early screening and treatment outcome prediction.

## Tissue-adaptive cell-type-specific gene expression prediction

More than only predicting cell fractions of bulk RNA-seq data like the existing deep-learning method, TAPE could also predict the cell-type-specific gene expression tissue-adaptively. That is, TAPE only needs simulated data from healthy samples to train, but it can also predict the cell-type-specific gene expression in pathological conditions if the corresponding bulk RNA-seq data is given. This feature enables TAPE to dissect bulk gene expression into different cell types and discover some potentially differentially expressed genes in different cell types.

We began with testing the correctness of the predicted cell-type-specific GEPs. To test this, we measured the concordance between the predicted gene expression value of each cell type and the original gene expression value obtained from single-cell RNA-seq (Fig. 4a, b). Here, the PBMC bulk data are from Monoco et al.[21], while the single-cell data from the data8k dataset from the 10X website[32]. Since we transformed the input RNA-seq data into 0−1 values using $Log_2$ and MinMaxScaler() in the training stage (see more in the Methods), the sums of gene expression values grouped by cell types are also transformed in this way to compare with the predicted relative gene expression value. Note that only gene expression in monocytes does not have a good concordance (Fig. 4b). After testing TAPE on a simulated dataset with a single-cell profile as ground truth (Fig. 4a) and considering the good concordance in other five cell types, we draw the conclusion that this distortion is caused by the individual difference. The concordance shown in the figures proves that TAPE predicts the signature matrix correctly and establishes the base for further gene expression analysis. In contrast, this disconcordance in monocytes also shows the adaptiveness of our method.

Besides the concordance, we also expect that TAPE can assign the gene expression value in bulk data to different values at a cell-type level. To test this, we used the ROSMAP RNA-seq dataset[25] and human brain single-cell profile[28] to perform adaptive training in the "overall" mode. The deconvolution result (Fig. 4d) of cell-type-specific GEPs shows that TAPE indeed predicted the differentially expressed genes in different cell types. However, since TAPE takes single-cell gene expression as input, these differences may be inherent from single-cell data. So, we compared the original signature matrix from single-cell data to the adapted signature matrix using the heatmap (Fig. 4c). We further investigated whether TAPE just inherits the data distribution from bulk RNA-seq data and whether the different distributions of different cell types are randomly assigned. We selected the *NRGN* gene to study it. Since the *NRGN* gene has been shown to be closely associated with AD[36], we expected it to have a high gene expression level in neurons and other nerve cells. Interestingly, for the predicted values (Fig. 4g, blue columns), the gene expression value in Endothelial is low compared with the high-level gene expression values in ExNeurons, InNeurons, and Astrocytes. In contrast, for the healthy single-cell profiles (Fig. 4g, red columns), expression values of *NRGN* in these four cell types don't have such big differences. Thus, TAPE can successfully predict a high expression value of *NRGN* in neurons while a low expression value of *NRGN* in endothelial cells. More specifically, this shows that the prediction of cell-type-specific GEPs is a product of two-sided information from both bulk and single-cell profiles, not randomly assigned or guessed. In this test, we also used the "group" mode of CIBERSORTx to predict the expression value of *NRGN* in different cell types. The results show that although CIBERSORTx can predict a high expression value of *NRGN* in InNeurons, it can not predict an expected high value in ExNeurons.

## Cell-type-specific differentially gene expression profiling at high-resolution

Since TAPE has shown its ability to predict cell-type-specific GEPs correctly and selectively given a group of bulk samples, we continued to use TAPE to predict cell-type-specific GEP per sample at high-resolution. To test TAPE's capability under "high-resolution" mode, we

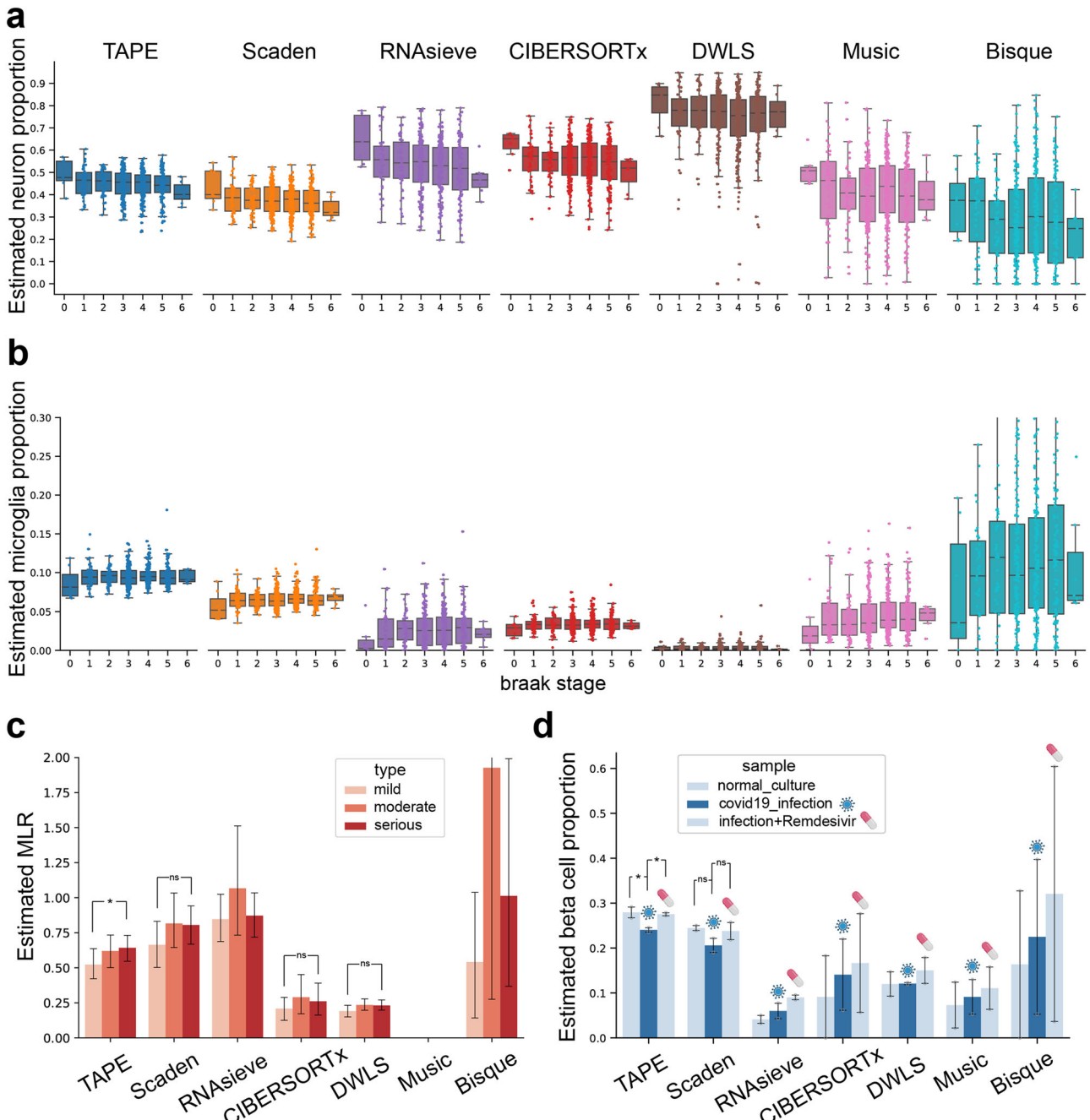

**Fig. 3 | Deconvolution benchmark on datasets with clinical information.**
**a** Comparison of estimated neuron cell proportion on different Braak stages between different models on the ROSMAP dataset. Neuron content is expected to decrease along with the development of AD. **b** Microglia content estimated by different methods on Braak stage. The fraction is expected to increase from stage 0 to 5 followed by a decrease from stage 5 to 6. In **a**, **b**, sample size of each stage from 0 to 6 is 7, 43, 47, 150, 174, 104, and 7, respectively. The boxes represent IQR while the solid line represents the median. The whiskers extend to points that lie within 1.5 IQRs of the lower and upper quartile, and then observations that fall outside this range are displayed as points independently. **c** Estimated MLR value calculated from the estimated monocytes fraction divided by the sum of estimated proportions of CD4$^+$ T cell, CD8$^+$ T cell, and B cell. We expect MLR value increases from

mild ($n = 12$) stage to moderate ($n = 14$) and serious ($n = 12$) stage. After one-sided Wilcoxon signed-rank test, we find that MLR increasement from mild to serious stage of TAPE has significance with $p = 0.0461$. **d** Estimated beta cell fractions of cultured islet in different conditions. The middle column represents samples infected with SARS-CoV-2, and the right one means samples treated with *Remdesivir* after infection. Sample size of each bar is 2. The model should predict the restoration of beta cell content after being treated with medication. One-sided *t*-test was used due to the small sample size. For TAPE's prediction, the *p* value is 0.0475 and 0.0142 for normal versus infected and treated versus infected, respectively. In **c**, **d**, these data are presented as mean values ± standard error of the mean. *p* value with notation * means $p < 0.05$, with notation ns means no significance. Source data are provided as a Source Data file.

synthesized a series of pseudo-bulk samples with known differentially expressed genes (DEGs) (Fig. 4e). Following the settings in CIBERSORTx[9], we selected 100 cells across four cell types (CD8 T cell, Natural Killer (NK) cell, B cell, and Monocyte) from PBMC single-cell

data and another 10 cells from human brain single-cell dataset as noise to compose the pseudo-bulk data. Then we randomly selected 100 genes among 10,000 genes in CD8 T cells as up-regulated genes to adjust their expression. Each pseudo-bulk dataset contains 50 pseudo-

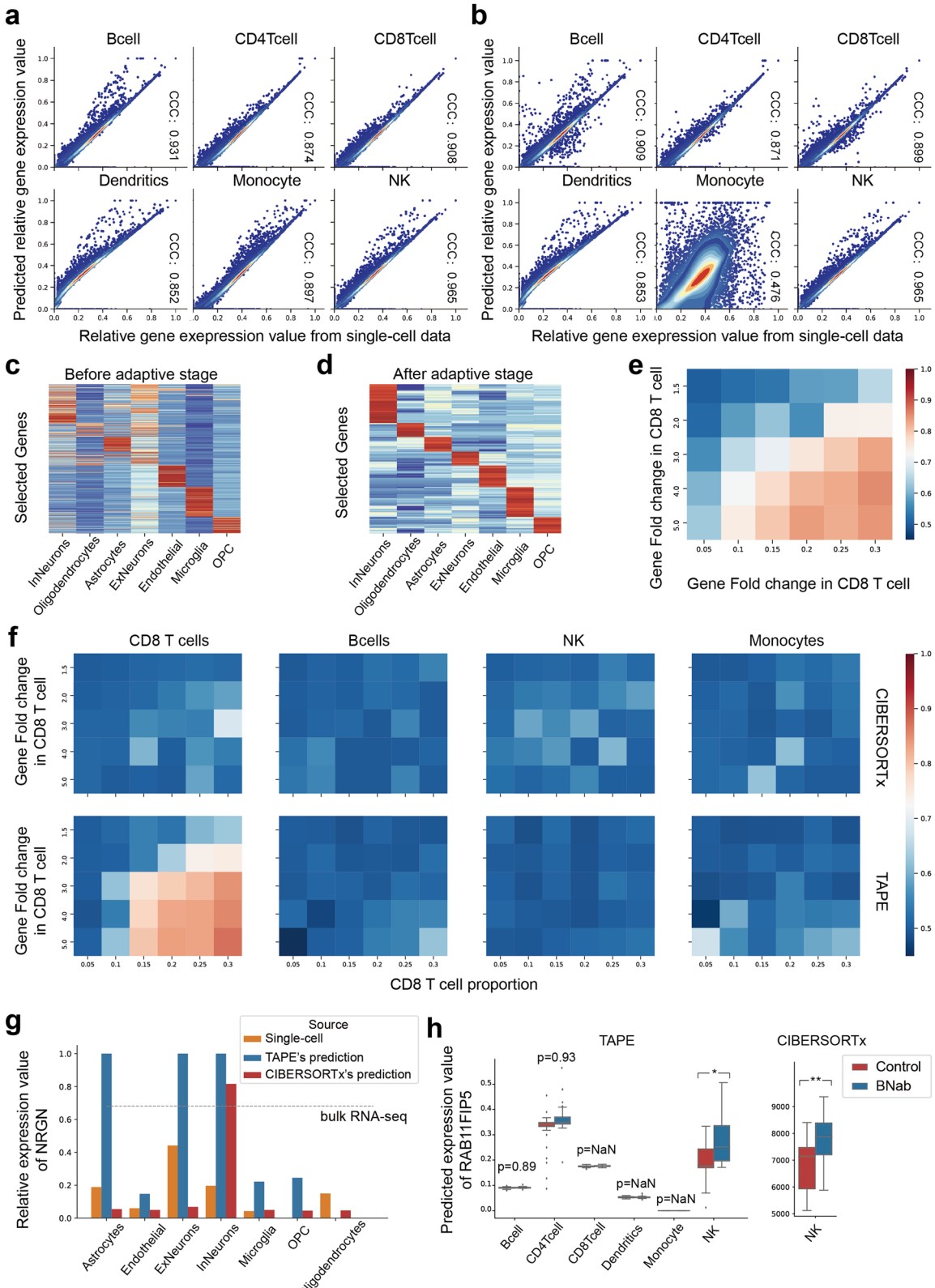

bulk samples and half of them are composed of up-regulated CD8 T cells. The cell proportion of CD8 T cells in pseudo-bulk data ranges from 5 to 30%, and the foldchange of up-regulated genes ranges from 1.5 to 5. In total, we created a series of pseudo-bulk datasets with foldchange gradients and cell proportion gradients. After obtaining the GEP of CD8 T cells, we used a two-sided *t*-test to detect DEGs ($p < 0.05$). So, this task is essentially a binary classification task, and we

naturally chose area under receiver operating characteristic curve (AUROC) as the criterion. The results show that (Fig. 4f), TAPE can successfully predict cell-type-specific DEGs correctly (with good sensitivity) and selectively (with good specificity) while CIBERSORTx almost fails on this task. The overall trend is that algorithms can easily recognize DEGs in one cell type if the proportion of this cell type or the foldchange of DEGs is high. Interestingly, using DEGs in bulk as the

**Fig. 4 | Cell-type-specific gene expression analysis. a** Concordance between the predicted relative gene expression value in simulated bulk data and the relative gene expression value in single-cell data. The relative gene expression value is the original expression value after Log$_2$ and MinMaxScaler() transformation. **b** Concordance between the predicted relative gene expression value in real bulk data and the relative gene expression value in single-cell data. **c**, **d** Estimated signature matrix after the adaptive stage in the "overall" mode. The gene expression is normalized with Z-Score. The genes are selected by the differential expression in different cell types after the adaptive stage. The differences between before and after the adaptive stage indicate that TAPE could not only make the signature matrix adapted to new data but also maintain concordance with the original one. **e** Differentially expressed genes detected from bulk RNA-seq data. The color indicates the AUROC value, red means better classification performance. Each row corresponds to different up-regulated foldchanges of randomly selected genes in CD8 T cells. Each column refers to CD8 T cell proportion in simulated bulk data.

**f** Differentially expressed genes detected by CIBERSORTx and TAPE in different cell types. DEGs should only be detected from CD8 T cell. **g** The relative gene expression value of *NRGN* from different sources. The dashed line represents the total relative *NRGN* expression value in the AD patients' brain tissue. The missing column means the relative gene expression value of prediction or single-cell data is zero. **h** Boxplots of the estimated *RAB11FIP5* gene expression values in different cell types by different methods. Both control group and BNab group have 46 samples in it. The estimated *RAB11FIP5* values by CIBERSORTx in other cell types are NaN (not shown). *p* value is calculated from two-sided *t*-test. *p* value has been adjusted by the false discovery rate. *p* value with notation * means *p* < 0.05 (exact value for TAPE is 0.025), with notation ** means *p* < 0.01 (exact value for CIBERSORTx is 0.00041). The boxes represent IQR while the solid line represents the median. The whiskers extend to points that lie within 1.5 IQRs of the lower and upper quartile, and then observations that fall outside this range are displayed as points independently. Source data are provided as a Source Data file.

reference, we can see that TAPE can even predict DEGs not shown up in bulk samples but in CD8 T cells (the maximum AUROC of TAPE is higher than the maximum AUROC of bulk samples in Fig. 4e, f). In the original article[9], CIBERSORTx has also demonstrated its great ability in DEGs prediction, the reason why it failed in this task is that CIBERSORTx usually focuses on signature genes which have bigger statistical power and are easily detectable if they are differentially expressed, but in this task, we randomly selected 100 genes which are probably not signature genes; therefore, it is hard for CIBERSORTx to infer the 100 DEGs properly. In contrast, TAPE has shown its ability in predicting DEGs even when they are not signature genes, which means TAPE has a broader application potential than CIBERSORTx. Of note, the recently published method, BLADE[13], can do this task too, but we did not benchmark BLADE in our experiments, considering its high time complexity.

In addition to the normal scenario where there are only 100 randomly selected DEGs with four non-similar cell types in simulated bulk samples, we designed comprehensive tests with four scenarios to benchmark TAPE and CIBERSORTx's performances. The four scenarios are: "randomly selected DEGs without similar cell type", "randomly selected DEGs with similar cell type", "signature genes as DEGs without similar cell type", and "signature genes as DEGs with similar cell type". In detail, we set up a series of simulated bulk data to detect DEGs as we mentioned before. However, we used similar cell types or changed the number of randomly selected genes, or used signature genes as DEGs in this test. Specifically, for the "similar" scenario, we used similar cell types like CD4 T cells and CD8 T cells together with two other cell types, namely monocytes and NK cells. In the scenarios where DEGs are randomly selected, the number of DEGs ranges from 100 to 5000. For the "signature genes as DEGs" scenario, we up-regulated the signature genes of CD8 T cells produced by CIBERSORTx in the simulated bulk samples. From the results (Supplementary Fig. 10), we can have four conclusions: (1) TAPE's predictive power is better than CIBERSORTx when the randomly selected DEGs are less than 1000; (2) both methods can achieve good performance when the DEGs are signature genes and there are not any similar cell types; (3) both methods can not distinguish DEGs from CD8 T cell rather than CD4 T cell if the DEGs are randomly selected; (4) CIBERSORTx is better than TAPE if the DEGs are signature genes and there exist similar cell types. Interestingly, from points 2 and 4, it seems that TAPE can learn the signature genes between distinguished cell types but not exactly enough to distinguish similar cell types. In all, considering all the scenarios, we display that each method has its own advantages and disadvantages and it can be seen as a guide for researchers to decide which method to use.

To further evaluate each method's "high-resolution" mode in the real-life scenario, we take HIV infection as an example where the researchers want to determine which cell type differentially express a gene between two conditions. In this case, HIV-infected patients can be classified into two different classes based on the existence of broadly

neutralizing antibodies (BNab). Recently, a study about the development mechanism of BNab in HIV patients used bulk RNA-seq and population sorted RNA-seq to investigate the most differentially expressed gene[37]. In this study, researchers initially found about 270 DEGs between two conditions using DESeq2[38]. After filtering non-related DEGs by controlling non-related information like age, sex, country, and viral load, researchers made the conclusion that *RAB11-FIP5* is the only differentially expressed gene in bulk samples. Then they used qPCR to find that *RAB11FIP5* is differentially expressed in NK cells rather than other cell types and leads to the development of BNab. The steps they used to find the relation between *RAB11FIP5* and NK could be replaced with the cell-type-specific gene expression analysis in the "high-resolution" mode. So, we used TAPE and CIBERSORTx to tissue-adaptively deconvolve the HIV PBMC data[37]. To avoid batch effects and harmful effects caused by the low-quality single-cell data, we combined data6k, data8k, and data10k PBMC single-cell data[32,39,40] as the reference. After obtaining the predicted GEPs for each sample at high resolution, we calculated the adjusted *p* value and foldchange for each cell type (Fig. 4h). The results show that both TAPE and CIBERSORTx successfully predict that *RAB11FIP5* is differentially expressed in NK cells. Considering that there are about 270 pseudo-DEGs in bulk samples, we further validated whether TAPE can distinguish them as pseudo-DEGs by checking the DEGs in each cell type. The results show that TAPE only predicts *NOP2* and *RAB11FIP5* as DEGs in NK cell and no DEGs for other cell types (Supplementary Fig. 8). We can see that the prediction is not perfect, but our method can correctly predict that NK cells have DEGs rather than other cell types and reduce the number of possible DEGs (including pseudo-DEGs if there is not any filter) from 270 to 2. All the results displayed prove that our methods can be applied to the real-life scenario and accelerate biological discoveries by identifying which cell type has DEGs and reducing the number of possible DEGs.

**Functional investigation across various types of virus infection**

To further prove the versatility of TAPE, we applied TAPE on the PBMC RNA-seq data of three kinds of virus-infected samples, including the SARS-CoV-2 infection, which is the severe acute respiratory syndrome coronavirus 2 that has been sweeping the world, hepatitis C virus (HCV) infection, which caused 290,000 death in 2019, and human immunodeficiency virus (HIV) infection, which is the cause of acquired immunodeficiency syndrome (AIDS). These three virus infections will damage the host's immune system but lead to different syndromes. Knowing the specific function in specific cells could help us in both the treatment and prevention of these infections.

Besides the differential expressed genes, we also investigated the functions of each cell type by incorporating cell-specific GEPs and ssGSEA[16]. Since the ssGSEA algorithm only needs the gene rank which can be provided by our method. We could predict the activities of each function pathway for each sample without positive or negative

controls. Considering function pathways that is significantly ($p_{adj} < 0.05$) activated in at least one sample, we found the samples that were infected by different viruses clustered (Pearson for distance, ward.D2 for cluster) together (Fig. 5a). Besides functional pathways that are differently activated at the population level, there also existed a diversity of activated functional pathways at the sample level, especially in dendritic cells (Fig. 5a).

Compared SARS-CoV-2 infected samples with the other two virus-infected samples, functional pathways had more potential to activate than inactivate . Also, the HIV-infected samples are similar to the HCV-infected ones, showing the difference between the SARS-CoV-2 infection and other ones. Besides, subsets of samples within each virus-infected sample could be also be identified, presenting the heterogeneous samples within the same virus infection.

Even the activities of the significant function pathways show differences among the three virus infections. The proportions of common significant enriched pathways were large in different cell types (Fig. 5b–d). More significant enriched function pathways were observed in SARS-CoV-2 infected samples, than in the other two virus-infection samples. In the B cells, HIV-infected samples shared 99% of significantly enriched pathways with HCV-indected samples, while SARS-CoV-2 occupied more than 40% of the significantly enriched pathways privately (Fig. 5b). These SARS-CoV-2 private enriched pathways contributed to the identification of the subset samples (Fig. 5e).

Of note, Monocytes and NK cells contributed to distinguishing these three kinds of virus-infected samples (Fig. 5f, g). We noticed that the number of common enriched pathways in these two cell types is much larger than the numbers of mono-enriched or di-enriched pathways, indicating the activation differences, rather than functional differences, make the various three virus-infection samples.

Combining with prior knowledge, some pathways we found are highly relevant to these diseases. For instance, most of the commonly activated pathways within the three infections in the B cells are general immune response pathways, including BIOCARTA_IL2_PATHWAY, BIOCARTA_IL4_PATHWAY, BIOCARTA_IL6_PA- THWAY, and BIOCARTA_IL7_PATHWAY. Interestingly, out of these pathways, BIO-CARTA_MAPK_PATHWAY, BIOCARTA_LONGEVITY_PATHWAY, and BIOCARTA_CELLCYCLE_PATHWAY have already been linked to SARS-CoV-2 infections. BIOCARTA_MAPK_PATHWAY (MAPKinase Signaling Pathway) activation has been proved to cause an overwhelming inflammatory response in SARS-CoV-2 infections[41]. The blockage of the BIOCARTA_LONGEVITY_PATHWAY (The IGF-1 Receptor and Longevity) has also been reported to mitigate lung injury and decrease the risk of death in patients with SARS-CoV-2[42]. Recent studies also found the coronavirus would induce the cell cycle arrest, which did not exist in other kinds of virus infection, but was discovered by our algorithm[43]. Generally, these examples show that the combination of TAPE and ssGSEA can indeed discover some significant pathways as clues for further experimental validation.

## Discussion

We develop TAPE as a deep-learning algorithm for digital tissue dissection. Key features distinguishing it from previous methods include (1) highly accurate and sensitive deconvolution to capture the biologically significant changes in clinical data, and (2) tissue-adaptive cell-type-specific gene expression profile prediction to identify potential gene expression differences at the cell-type level. TAPE benefits from the architecture of the autoencoder and the unique training method in the adaptive stage. The encoder-decoder architecture enables an interpretable decoder to answer why the encoder makes such predictions. More interestingly, the decoder is a natural cell-type-specific signature matrix that can be learned after the training stage and then adapted to the bulk data after the adaptive stage. Notice that the special training process of TAPE makes it fundamentally different from

other methods, which only predict cell fractions or need large cohort bulk RNA-seq data to impute cell-type-specific GEPs or are hard to infer insignificant gene expression in cell-type-specific GEPs. Another advantage of TAPE is its super fast inference when deconvolving a large number of samples. Running on a commonly-used graphics processing unit (GPU), TAPE has comparable speed to the fastest statistical method and even faster than the previous deep-learning method.

Although we have shown that TAPE's deconvolution performance is pretty good in many scenarios, we find that it would perform poorly in the "rare" scenario since it shows a low CCC value. But, in the benchmarking process (Fig. 2c), the results show that other tools' performance also drops in the "rare" scenario. This phenomenon indicates the "rare" scenario has not been solved well by current methods and needs to be addressed in future works. In the scenario of clinical data prediction, TAPE is capable of predicting the ratio change for most cell types in clinical cases stably with statistical power, whose results are consistent with the previous related clinical studies[15,27,30,31,33,34]. During real-life usage, to make the study more focused, we recommend that users select the cell types they want to analyze further from the TAPE output based on the existing experimental evidence.

As is previously highlighted, TAPE can predict cell-type-specific GEPs tissue-adaptively. But admittedly, it can be improved further. Firstly, when we study the correlation at the gene level using "overall" mode (Supplementary Fig. 7), about 30% of the predicted genes have the negative correlation. Although our method's performance (median CCC 0.2127) is better than CSx (median CCC 0.0627), there is still large room for improvement. Secondly, when we use it to predict DEGs, it is hard for TAPE to predict a proper foldchange, this is partially caused by the normalization method since the gene value is normalized between 0 and 1. However, this phenomenon can also be observed in CIBER-SORTx (Supplementary Fig. 8), which indicates the information loss between bulk samples and inferred cell-type-specific GEP is hard to be reconstructed. In our tests, considering the fact that the predicted foldchange is not proper, we only uses $t$-test to find DEGs with $p < 0.05$ and we can obtain plausible DEG results from this criterion. Thirdly, we notice that both CIBERSORTx and our method can not distinguish DEGs from similar cell subtypes correctly if the DEGs are not signature genes (Supplementary Fig. 9) which means that their resolution is still limited. But CIBERSORTx has displayed its advantages in distinguishing signature DEGs from similar cell types because of the incorporation of the signature matrix (Supplementary Fig. 10). Though our method cannot precisely predict DEGs from cell subtypes or have better performance than CIBERSORTx if all signature genes are DEGs which probably does not occur in the real world, it still reduces the potential candidates by excluding irrelated cell types. So, our method is still useful and can be applied in real-life scenarios to accelerate biological research.

Benefited from the predicted cell-type-specific GEPs in the "high-resolution" mode, we could identify specific activated functional pathways in each cell type for each sample, which could be another potential advantage of our algorithm. According to the results above, we could identify cell types involved in the dysfunctional pathways. Combining ssGSEA and TAPE could help identify the specific dysfunctional pathways in particular cell types using the bulk RNA-seq data, which will essentially make use of previous population transcriptome datasets.

In summary, TAPE represents a widely applicable framework for deciphering the heterogeneity of tissues at a cell-type level, and provides a practical training scheme for supervised autoencoder to perform domain adaptation. Considering the fact that it can be integrated with other tools seamlessly, we believe that TAPE will be helpful to investigate the connection between the single-cell data and the abundant bulk data.

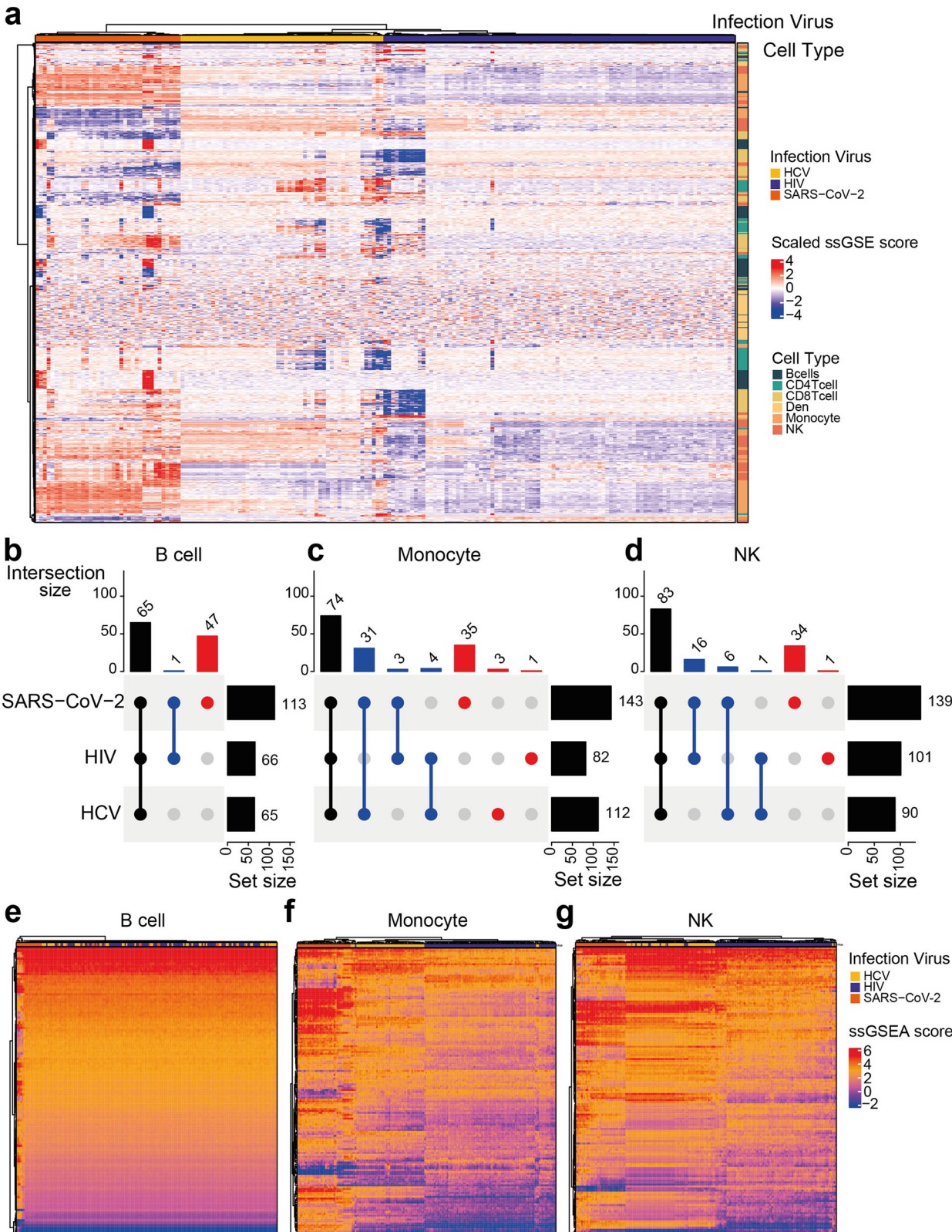

**Fig. 5 | Function enrichment of cell-specific GEP. a** Heatmap of enrichment scores for various cell types, including B cells, CD4 T cells, CD8 T cells, Dendritic cells, Monocytes, and NK cells within different virus infection samples. The enrichment scores have been scaled by the Z-score. The top row annotation represents the virus types of the infection. The left column annotation represents the corresponding cell types of the enriched pathway. Significantly enriched pathway upset plots for **b** B cells, **c** Monocytes, and **d** NK cells in three kinds of virus infection. Heatmaps of enrichment scores for **e** B cells, **f** Monocytes, and **g** NK cells. Source data are provided as a Source Data file.

## Methods

### Datasets and preprocessing

In this work, we used several public single-cell RNA-seq datasets, bulk RNA-seq datasets, and microarray datasets to perform our experiments. In the pseudo-bulk test, a single-cell dataset of mouse atlas from *Tabular Muris*[18] was used. This dataset consists of 20 organs and tissues with cell type labels provided by the authors. Only three tissues' (Limb_Muscle, Marrow and Lung) data in both protocols were used to perform the pseudo-bulk test. Other data were not selected because the shared cell types across protocols are very limited (less than four cell types), which can not simulate the real-life scenario.

In the experiments of real bulk data with ground truth, we used several real bulk datasets with the corresponding cell fractions. The first PBMC dataset SDY67 was created by Zimmermann et al., but it was indirectly obtained from Scaden's training data with unknown fractions. The second PBMC dataset created by Monaco et al. could be downloaded from the GEO database with accession number GSE107011. The corresponding cell fractions data were provided as supplementary information of the original paper. More specifically, the unknown fraction was calculated by one minus the sum of known proportions, and cell types of the same kind were added together to fit the cell types in training data. For example, monocytes C, monocytes I, and monocytes NC are different kinds of monocytes, so their fractions will be added together as the total fraction of monocytes. When we used it to test whether deconvolution methods can achieve good performance with immune cell subtypes, we merged all 30 cell types into 13 cell subtypes (Supplementary Table 1). The similar subtypes are defined as "mDC" and "pDC", "naive CD4 T cell" and "non-naive CD4 T cell", "naive CD8 T cell" and "non-naive CD8 T cell", and "naive B cell" and "memory B cell". The third PBMC dataset was created by Newman et al. Its expression data were downloaded from GEO with accession number GSE65133 and its cell fractions were provided on the webpage of CIBERSORT[7]. Next, the dataset we used to deconvolve human tissue with Alzheimer's Disease (AD) was obtained from a project called Religious Orders Study and Memory and Aging Project (ROSMAP)[25]. This dataset consists of about 600 samples of RNA-seq data from AD patients, while 41 of them have cell-type proportion information measured by immunohistochemistry in another study[23]. The gene expression data were obtained from the supplementary data of Scaden rather than the original program of ROSMAP to maintain consistency during the test. As for the single-cell datasets, 8k PBMC dataset from healthy donors was downloaded from 10X Genomics[32], mouse and human brain datasets were obtained from the GEO database with accession numbers GSE87544 and GSE67835 respectively[28,44]. All of these datasets were preprocessed to generate cell-type labels using the same procedure in Scaden. Notably, if the training data were available in Scaden, like PBMC and mouse brain datasets, we just used the training data provided by the authors of Scaden to assess performance.

In the advanced analysis of real bulk data with clinical information, three different datasets were involved. Since the ROSMAP dataset has been introduced above, here we only describe the other two datasets. The first is the COVID-19 PBMC dataset[26] from a longitudinal study of patients with COVID-19. This dataset has 39 RNA-seq samples of PBMC consisting of different stages (treatment stage, convalescence stage, and rehabilitation stage) and different types (mild, moderate, and serious) from 16 patients. The second is the COVID-19 islet dataset which is from a study of the SARS-CoV-2 infected islets. This dataset only has six samples which are divided into three groups: normal cultured group, infected group, and *Remdesivir* treated group. The single-cell dataset used as the reference is from Baron et al.[35] (GEO accession number: GSE84133) which has 14 labeled cell types in pancreas tissue. Instead of using all the cells in the dataset, we only selected endocrine cells: alpha cell, beta-cell, delta cell, gamma cell, and epsilon cell to constitute the reference dataset.

In the final analysis of tissue-adaptive GEPs, we introduced an HIV PBMC dataset from the GEO database with accession number GSE115449. This dataset has PBMC data collected from 92 HIV patients. Half of them have developed BNab and the others do not have BNab. Furthermore, when we used ssGSEA to analyze cellular function changes in PBMC across different viruses' infections, we used an HCV-infected bulk RNA-seq dataset of PBMC (GEO database, accession number: GSE119117). This dataset is also from a longitudinal study of patients. RNA-seq data were collected from individuals before, during, and after acute HCV infection. See more details on the GEO database. Another virus-infected PBMC dataset is the COVID-19 PBMC dataset which has been mentioned before.

Note that, the datasets involved in this study might use different ways to represent genes. To maintain the concordance, we processed all the different representations into gene names through BioMart[45].

### The TAPE framework

**Simulation of pseudo-bulk data from a single-cell dataset**. Usually, deep learning models need a large amount of training data to optimize its loss function and learn its parameters. So, it is crucial to generate pseudo-bulk data from a single-cell dataset to train the model. Single-cell expression data with cell type fractions are used to generate pseudo-bulk data. By definition, pseudo-bulk expression data are the sum of single-cell expression data from a subset of cells. So, to generate pseudo-bulk data, cells should be sampled with a given cell type proportion (ground truth) and total cell number like the stratified sampling.

Typically, cell-type fractions could be generated using dirichlet distribution when users have some prior information about cell-type fractions in a specific tissue. The cell-type fractions were first generated using the *dirichlet()* function from the *numpy.random* package[46] and users could define the prior cell fractions by setting the parameters in the *dirichlet()* function. If they do not have prior knowledge, the prior weight of each cell type will be the same (normal samples). Following Scaden, half of the generated samples' corresponding cell type fractions contain zeros (sparse samples). Because, in our practice, deconvolution performance will be improved by training with both normal and sparse samples. Next, we multiply the total cell number with the generated cell fractions for each sample to acquire the exact sampling number for each cell type. After that, we use a stratified sampling method to sample cells of each cell type with the given number. Finally, the pseudo-bulk expression profile is created by summing the expression values of the randomly selected single-cell expression profiles for each sample.

Additionally, if users want to predict tissue-adaptive GEPs and investigate the relative gene expression value (output GEP value is between 0 and 1), they need to consider the data shift between different sequencing methods. For example, counts data from the 10X sequencing platform represent the real expression value while counts data from smart-seq[47] need to be further normalized using a method like TPM or FPKM to show the real expression value. Here we provide a simple function *counts2FPKM()* (or TPM) to transform raw counts to FPKM (or TPM). Due to the original information loss of the processed single-cell expression profile, we only normalized raw counts of a certain gene with its maximum transcripts length obtained from BioMart[45]. So, we recommend users prepare a suitable single-cell profile in advance to avoid information loss.

According to the previous study from Scaden[14], different sampling distributions and single-cell datasets with a heavy bias of different cell types do not affect deconvolution performance notably. So, we think the simulation procedure is reasonable in our settings.

**Problem definition**. To illustrate our model more clearly, it is necessary to define the problem in advance. All of the symbols defined in this section are consistent throughout the article. Intuitively, we expect

GEPs from bulk RNA-seq would be a linear combination of each cell's GEPs from single-cell RNA-seq. Furthermore, if cells belonging to one kind of cell type have the same gene expression pattern, we could use the signature gene expression pattern and the number of cells for each type to reconstruct the GEP of a bulk RNA-seq data. So, given the number of $k$ cell types, $m$ genes, and $n$ samples in bulk RNA-seq data, an ideal mathematical model could be defined as:

$$\mathbf{X} \cdot \mathbf{S} = \mathbf{B}, \tag{1}$$

where $\mathbf{B}$ is an $n \times m$ matrix representing GEPs of bulk RNA-seq; $\mathbf{S}$ is a $k \times m$ signature matrix; $\mathbf{X}$ is an $n \times k$ matrix representing cell-type fractions in each sample.

**Model set-up.** Given the well-defined problem, we just need to modify the equation to accommodate deep learning:

$$
\begin{aligned}
f_\phi(\mathbf{B}) &= \tilde{\mathbf{X}}, \\
f_\psi(\tilde{\mathbf{X}}) &= \tilde{\mathbf{X}} \cdot \mathbf{S}, \\
f_\psi(f_\phi(\mathbf{B})) &= \tilde{\mathbf{B}}.
\end{aligned}
\tag{2}
$$

Here, $f_\phi$ and $f_\psi$ represent two coordinated deep neural networks, symbols with tilde like $\tilde{\mathbf{B}}$ refers to the output of the model; and $\mathbf{S}$ refers to the explicit matrix form of $f_\psi$. Usually, $f_\phi$ and $f_\psi$ are called encoder and decoder respectively in the classical architecture of AE. $f_\phi$ is a regression model which is responsible for mapping the high dimensional bulk gene expression data to a low dimensional representation of cell compositions. In contrast, $f_\psi$ is the inverse function of $f_\phi$ which is expected to reconstruct bulk data based on the cell fractions. Obviously, $f_\psi$ functions like the signature matrix discussed in previous sections. Therefore, we want to make it have an explicit matrix-form to enforce the interpretability of $f_\phi$. To achieve the progress in the interpretability of deep model, $f_\psi$ was designed without activation layers or biases, which is only the regularized value of dot product of five weight matrices. Thus, the signature matrix is visible in the deep model:

$$f_\psi = \mathbf{S} = \mathrm{ReLU}(\mathbf{W}_1 \cdot \mathbf{W}_2 \cdot \mathbf{W}_3 \cdot \mathbf{W}_4 \cdot \mathbf{W}_5), \tag{3}$$

where $\mathrm{ReLU}(x) = (x)^+ = \max(0, x)$. The reason to design such an equation to represent $\mathbf{S}$ rather than a single matrix is that more parameters could enable the model to learn a good signature matrix more quickly and easily, and the ReLU($\cdot$) function is used to ensure the biological meaning of the signature matrix. Of note, the decoder matrix is expected to represent a meaningful signature matrix only after the training with simulated data.

We need to stress that, it may seem that our model assumes that cell proportions could be inferred from the bulk data directly through the function $f_\phi$ without the signature matrix. However, if we consider $f_\psi$, we will find that parameters of $f_\phi$ is affected by $f_\psi$ during optimization. Just like other statistical methods computing the pseudo-inverse of the signature matrix in the fitting process, we also use the inverse relationship between $f_\phi$ and $f_\psi$ in the training stage. Therefore, compared with the previous machine learning methods using a single function to predict fractions without regularization from the signature matrix, this architecture makes more sense. More specifically, although Variational AutoEncoder (VAE) has become a powerful tool to model single-cell data recently[48–50], we do not use VAE because the encoded latent variable is probabilistic, not deterministic. This is the reason why VAE is very suitable for generative tasks, while not suitable for the cell-type deconvolution.

**Input data preprocessing.** Although the input datasets varied between platforms and protocols, we utilized the same processing approach to prepare them for deep-learning models and alleviate the effect of the dimensionality curse. As for the bulk data (real or simulated), it is first

transformed to the $\mathrm{Log}_2$ space with a pre-added one to avoid null value. Then we need to filter some genes with low variance both in the training data and test data[10,14]. This step is very important, because TAPE will fail in predicting test bulk data proportions properly without proper filtering (Supplementary Table 2 and Supplementary Fig. 11)[14]. In our experiments, we control the filtering threshold to keep about 10,000 genes as reported by Scaden[14]. If the less variable genes are not filtered out, TAPE can not predict a good result because of the noises (Supplementary Table 2 and Supplementary Fig. 11). Further signature gene selection methods[24,51] may help TAPE improve its performance. Next, to maintain the meaningful signature matrix, we decide to use the *MinMaxScaler*() function provided by scikit-learn[52] to scale data into the range between 0 and 1. This function is described below:

$$\mathbf{B}_{i,j} = \frac{\mathbf{B}_{i,j} - \min(\mathbf{B}_i)}{\max(\mathbf{B}_i) - \min(\mathbf{B}_i)}, \quad j = 1, 2, 3, \ldots, m. \tag{4}$$

**Training method.** As previously stated, there are two stages of training in TAPE. The first is the training stage, where we use about 5000 pseudo-bulk samples for training. we use MAE between prediction and ground truth to optimize the parameters of encoder and MAE between the reconstructed input and the original input to optimize both the decoder and the encoder. The loss functions are defined as:

$$
\begin{aligned}
\mathrm{MAE}(\mathbf{X}, \tilde{\mathbf{X}}) &= \frac{\sum_{i,j} |\mathbf{X}_{i,j} - \tilde{\mathbf{X}}_{i,j}|}{n \times k}, \\
\mathrm{MAE}(\mathbf{B}, \tilde{\mathbf{B}}) &= \frac{\sum_{i,j} |\mathbf{B}_{i,j} - \tilde{\mathbf{B}}_{i,j}|}{n \times k},
\end{aligned}
\tag{5}
$$

where symbols with tilde represent learned/predicted data in the training stage.

Usually, we found that $\mathrm{MAE}(\mathbf{X}, \tilde{\mathbf{X}})$ is stable after 5000 iterations with batch size 128, so we stopped training to avoid overfitting.

In the adaptive stage, we aim to train the parameters to adapt to new data rather than predicting cell fractions with the same parameters in all situations. To achieve this goal, we design a greedily iterative optimizing method: step 1. optimize the decoder with loss function $\mathrm{MAE}(\mathbf{B}, \tilde{\mathbf{B}}) + \mathrm{MAE}(\tilde{\mathbf{S}}, \tilde{\mathbf{S}}_0)$ until $\mathrm{MAE}(\mathbf{B}, \tilde{\mathbf{B}})$ does not decrease; step 2. optimize the encoder with loss function $\mathrm{MAE}(\mathbf{B}, \tilde{\mathbf{B}}) + \mathrm{MAE}(\tilde{\mathbf{X}}, \tilde{\mathbf{X}}_0)$ until $\mathrm{MAE}(\mathbf{B}, \tilde{\mathbf{B}})$ does not decrease. Here, $\tilde{\mathbf{X}}_0$ and $\tilde{\mathbf{S}}_0$ refer to the results of cell type fractions and cell-type-specific gene expression matrix after initial training. The intuition is that we want the decoder (signature matrix) to adapt to the bulk data first because each new bulk sample has a different signature matrix. Then we want the encoder to adapt to the bulk data to predict a slightly different cell fraction. Since this iterative method is not guaranteed to converge, we have to make the adapted parameters as close as possible to the original parameters. For the same reason, if we train both encoder and decoder simultaneously, it would be hard to guarantee the model could converge on our expectation, so we train them separately to make more sense. Usually, repeating step 1 and step 2 several times would make the parameters of TAPE adapt to the new data. In our experiments, after the adaptive stage, the prediction of cell fractions will improve a little, and it always outputs an adaptive signature matrix. The adaptive training stage is more like the fine-tuning step in deep learning rather than being re-trained with new single-cell data. The adaptive-training time is 3 s per sample with GPU acceleration.

**Predict tissue-adaptive cell-type-specific GEP in different modes.** Generally, there are two different ways to analyze cell-type-specific GEPs: (1) The "overall" mode: using all the samples at once to capture an overall cell-type-specific GEPs in a certain condition. (2) The "high-resolution" mode: predicting all the samples one by one to maintain the differences between each sample. Certainly, the latter will consume more time than the former one. Usually, it takes 3 s to deconvolve cell-type-specific GEPs for each sample. The choice of different

modes mainly depends on users' demands. For example, if users want to discover the differentially expressed genes at a cell-type level, they should choose to predict GEPs in the "high-resolution" mode; thus they could calculate the *p* value. On the other hand, if users only need to investigate the highly expressed genes in different cell types or have an overall look at the cell-type level, they should choose to predict GEPs in the "overall" mode.

It is worth noting that, the value of the GEP predicted by TAPE is between 0 and 1, which represents the relative expression value within a single sample due to the Min–Max scaling. Using this GEP may encounter some problems when users need to analyze the foldchange of a certain gene due to the information loss induced by the nonlinear scaling function.

**Architecture and hyperparameters.** TAPE's encoder and decoder are both made up of five fully connected layers with the same weight size in the corresponding position. For example, the first layer of the encoder and the last layer of the decoder each have 512 nodes. More specifically, the number of nodes in each encoder layer is 512, 256, 128, 64, and the number of cell types is in sequential order. Before each of the first four fully connected layers, there is a dropout function with a probability of 0.5; after each layer, each has a nonlinear activation function CELU($\cdot$), defined as CELU($x$) = max(0,$x$) + min(0,$e^{x-1}$). Decoder, on the other hand, does not contain any bias in the fully connected layers or nonlinear functions except the ReLU($\cdot$) function, as we have mentioned before. During the training stage, we use Adam with a learning rate of $1 \times 10^{-4}$ to optimize parameters. Other parameters of Adam are set as default in PyTorch. We train the network for 5000 iterations with batch size 128. These training hyperparameters are succeeded from Scaden. While in the adaptive stage, we use Adam with the same learning rate $1 \times 10^{-4}$ to fine-tune the parameters on the new data. We train both the encoder and the decoder for 300 steps within each iteration. The max iteration number is flexible for users, and we recommend users set it to at least 2 to make it output a well-adapted signature matrix.

**Model interpretation.** This deep learning model is very similar to the middle-size model in Scaden. The performance improvement of this model is caused by the dropout layer, the decoder and the selection of activation function. We add the dropout layer in front of the first linear layer. This setting is rare in deep learning because it would be hard for the model to learn if there is only half of the input features. This empirical knowledge also works for TAPE, but we introduce the decoder layer to avoid the performance drop. In contrast, since the decoder could stabilize the encoder, the dropout layer in the encoder could help the model recognize which set of features is crucial to the result, and this dropout layer will enhance the performance. As for the decoder, it is actually a single matrix with constraint (all elements $\geq 0$). The reason why we use dot product of five matrices is to improve the speed of convergence. In practice, the results have shown that the more parameters the model has, the faster the convergence speed will be. Moreover, we need to stress that the activation function will affect the performance. Previous deep learning framework, Scaden, uses Softmax() as the final activation function to guarantee the prediction is meaningful. But its drawback is that the training process will be less stable. Specifically, when the training data contain zero proportions for some cell types, and the model is forced to predict a zero during the training process, it would be really hard for a Softmax() function to predict a zero, and the last layers' features should be very negative values which are harmful to the numerical stability. The similar problem is raised in the image classification task, and researchers usually use label smoothing to avoid this problem[53,54]. However, since label smoothing is not appropriate for the regression task, when predicting cell fractions, we finally use ReLU() and a scale function to guarantee the summation of cell fractions is 1.

## Performance evaluation

Within the main text above, we combined mean absolute error (MAE) with Lin's concordance correlation coefficient (CCC)[19] to evaluate different algorithms' performance because it is hard to assess performance reasonably in all situations with only one metric. For instance, suppose there are only two kinds of cell types in tissue, and one type's fraction ranges from 80–90% in the ground truth. If the model predicts this cell type fraction is 100%, then the CCC value may imply a satisfying performance, but the MAE value may indicate the opposite. So, to avoid the situation of discarding fractions of minor cell types, it is necessary to combine MAE with CCC. Generally, a higher CCC value and a lower MAE suggest a better deconvolution performance. These metrics are defined as follows:

$$\text{MAE}(\mathbf{X},\tilde{\mathbf{X}}) = \frac{\sum_{i,j}|\mathbf{X}_{i,j}-\tilde{\mathbf{X}}_{i,j}|}{n \times k},$$
$$\text{CCC}(x,\tilde{x}) = \frac{2 \times cov(x,\tilde{x})}{\sigma_x^2 + \sigma_{\tilde{x}}^2 + (\mu_x - \mu_{\tilde{x}})^2}, \tag{6}$$

where $cov(x,\tilde{x})$ stands for the covariance between these two vectors.

Notably, these two metrics are applied to all data points of the predicted matrix $\tilde{\mathbf{X}}$ and the ground truth matrix $\mathbf{X}$. More specifically, for the CCC value, we reshape the matrix into a vector and then calculate the total CCC between two vectors. This calculation pattern usually results in a higher CCC value than computing the average CCC value for each cell type.

## Statistics and reproducibility

Determining the sample size for the deconvolution problem is a challenging problem with no existing method, so we chose the sample size according to previously published datasets. Datasets were chosen in order to show the functionality and performance of our method. No data were excluded from the analyses. Replication and randomization are not applicable since we did not collect any experimental data. Hypothesis testing methods are explained in each figure legend. Any group allocations were determined by previously published dataset, we did not modify the group information. To reproduce the results, please find the Source Data file we provided.

## Software comparison and settings

To evaluate the performance of TAPE compared with other methods, we selected several representative methods for comparison. Except for Scaden, other methods were tested following the instruction and tutorials provided by each package.

For deconvolution performance on the pseudo-bulk and real bulk data with ground truth, we benchmark Scaden, RNAsieve, CIBERSORTx, DWLS, MuSiC and Bisque[8–12,14]. We will describe the details of the benchmarking procedure below. Hyperparameter tuning file is available in Supplementary Tables 3–7.

For Scaden, we tested its performance in both platform: Keras-based Scaden (provided by Scaden's authors) and PyTorch-based Scaden (implemented by ourselves). The training hyperparameters of PyTorch-based Scaden were set following the instruction of the original article and the source code. Though we tried our best to make it the same as the original Scaden, it still had some different behaviors, for example, loss plot and deconvolution performance on the SDY67 dataset were different from reported data. These differences were probably caused by the different deep learning backends (Keras or PyTorch). In general, since the differences were not huge, the implementation of Scaden is acceptable. The results shown in the Results part is mainly based on the Keras-based Scaden. Only the detailed performance comparison with different random seeds is produced by PyTorch-based Scaden.

For RNA-Sieve, it does not have a detailed documentation, so we ran it following its example code. In practice, we can produce the same result as its example code but the results of RNA-Sieve in pseudo and

real bulk test are not as good as previously reported. We first validated its performance on pseudo-bulk data and the original data provided by the authors. The results showed that it could perform well on the simulated data but it could not reproduce the same results as they reported on Newman's dataset and Monaco's dataset. The benchmarking code is available in the supplementary file.

For CIBERSORTx (CSx), we used the web-based application to test. We first used a single-cell profile to generate a signature matrix and then we deconvolved the corresponding bulk data with S mode batch-correction. Other settings were default. When we need to predict the differentially expressed genes in different cell types for HIV PBMC dataset, we first selected the signature matrix and added the RAB11FIP5 gene to the list of genes of interest. Then we ran CSx to infer the gene expression profiles at high-resolution mode. All the procedures are following the tutorial given on the website

For DWLS, we used the core functions and packages written in R programming language to generate signature matrices and therefore deconvolving the targeted pseudo-bulk data and the real ones. To guarantee the rationality of our implementation, we carefully followed the example of the intestine stem cell provided by the manual of DWLS. Since the deconvolution function provided by DWLS only deconvolves one sample at a time, a for-loop is brought in because we need to deal with some large samples. For a better performance of DWLS, in the step of generating signature matrix, we used the *Seurat* flavor for pseudo-bulk test and the MAST[24] flavor for real bulk test. Furthermore, to ensure the stability and usability for all the bulk samples, we used support vector regression (*nu-SVR*) to obtain the initial estimation instead of the ordinary least square regression.

For MuSiC, we installed the R package and ran it with default settings following its tutorial. Of note, MuSiC claims it can take the advantages of multi-subject single-cell profiles to improve deconvolution performance. But in our pseudo-bulk test, we do not have multi-subject single-cell data, to meet its requirement, we randomly assigned one dataset to two virtually different source. In the real bulk test, we first combined PBMC data6k and data8k as reference to deconvolve PBMC bulk dataset, but MuSiC failed to predict them properly. The CCC value was negative which has been reported by previous study[9]. Then we only used PBMC data8k as reference and assigned it to two virtually different datasets to deconvolve PBMC bulk data. With only one source single-cell profile the CCC value was normal and this result challenges the MuSiC's claim. Thus, we only displayed the one-subject single-cell reference results above.

For Bisque, we installed the R package BisqueRNA and ran it with default settings following its example provided by the author. In the default mode of the "Reference-based decomposition" mode, Bisque filters out low variance genes and uses genes left for decomposition, so we input all the genes without specifying some marker genes.

To generate all the figures, we used the following python packages: matplotlib[55], seaborn[56], and pandas[57]. Additional packages such as anndata and tqdm[58] are used to build our method.

It should be pointed out that, for all statistical methods (RNA-sieve, CSx, and DWLS), all PBMC datasets were deconvolved using a signature matrix generated from PBMC data8k dataset, reference of mouse brain dataset is generated from Chen et al.[44], and signature matrix of human brain dataset is generated from Darmanis et al.[28]

### Reporting summary
Further information on research design is available in the Nature Portfolio Reporting Summary linked to this article.

## Data availability
All the datasets we used are listed in the Method part. Only the ROS-MAP human brain dataset is not public, researchers need to download it from Synapse (ID: syn3219045) with a request. For convenience, we listed these datasets on the webpage: https://sctape.readthedocs.io/ datasets/. All other relevant data supporting the key findings of this study are available within the article and its Supplementary Information files or from the corresponding author upon reasonable request. Source data are provided with this paper.

## Code availability
The open source implementation of TAPE is available at https://github.com/poseidonchan/TAPE[59], and the experiments conducted to produce the main results of this article are also stored in this repository. The documentation of TAPE is published at https://sctape.readthedocs.io/.

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

## Acknowledgements

We thank Mengyue Sun for his help to accelerate the simulation process. Special thanks to people who helped us request for the ROSMAP dataset and the dataset creators from Religious Orders Study (ROS) or the Rush Memory and Aging Project (MAP). The work was supported by Chinese University of Hong Kong (CUHK) with the award number 4937025, 4937026, 5501517, and 5501329 and the General Research Funds 14306020 and 14305319 from the Hong Kong Research Grants Council of the Hong Kong Special Administrative Region of the People's Republic of China. Y.C. and T.F.C. are supported by University Grants Committee Area of Excellence Scheme [AoE/M-403/16], a generous donation from Mr. and Mrs. Sunny Yang; and the Innovation and Technology Commission, Hong Kong Special Administrative Region Government to the State Key Laboratory of Agrobiotechnology (CUHK). Y.Y.W. and T.F.C. are supported by a Project Impact Enhancement Fund (PIEF) and Science Faculty's Collaborative Research Impact Matching Scheme (CRIMS).

## Author contributions

Y.L., Y.S.C., and Y.X.W. conceived of and designed the computational method; Y.S.C. implemented the main algorithm; Y.S.C. and Y.X.W. did experiments and interpreted the results; Y.S.C. and Y.Q.C. analyzed and interpreted the clinical analysis results; Y.L.C. analyzed and interpreted the ssGSEA results; Y.S.C., Y.X.W., and Y.L.C. wrote the manuscript; Y.M.W., Y.X.L., J.W., Y.Y.W., T.F.C., and Y.L. revised the manuscript. T.F.C. and Y.L. supervised the project. All authors read and approved the final paper.

## Competing interests

The authors declare no competing interests.
