## [Peer Review File · Nature Communications]

Deep autoencoder for interpretable tissue-adaptive deconvolution and cell-type-specific gene analysisREVIEWER COMMENTS

Reviewer #1 (Remarks to the Author):

This manuscript proposes TAPE, a method consisting of a deep autoencoder to improve cell deconvolution. They propose to improve over Scaden, an ensemble of deep neural networks for cell deconvolution. The method uses reconstruction loss on real bulk samples and mean absolute error on simulated data to learn cellular fractions. The authors design the training paradigm to learn in two steps and their intuition is to adapt to the tissue being tested on. Additionally, the method is extended to predict cell-type specific gene profiles (csGEPs) both per-group/overall and per-sample/high-resolution. Comparisons with previous deconvolution approaches, CibersortX, RNAsieve and DWLS are given on simulated and real RNAseq datasets.

On the task of estimating cell proportions, TAPE generally seems to perform better than the methods compared, however, there are some ambiguities and experiments where more information is necessary. On the task of estimating cell-type specific gene expression profiles, the evidence in general seems weak.

The manuscript is well-written and easy to follow.

Major comments:

To mimic real data, the authors mention that the simulated data includes gaussian noise and dropouts, however, no further information is given. It is necessary to know the levels of that noise considered.

The evidence provided to support the improvement in TAPE over existing algorithms is given over all cell-types present in a dataset. While this is a good comparison on evaluating a method overall, it does not provide any information about how TAPE performs on different cell-types. What happens if certain cell types are missing from the bulk or single cell data?

Since TAPE uses reconstruction loss on real bulk to aid in learning, how does the sample size of the tissue bulk data affect the performance of TAPE?

It is unclear how the method behaves on cell-types that are highly related. Any evidence supporting that will be helpful. Similarly, how does TAPE perform on cell fractions below 5% and 2%? One of the main unresolved issues is good performance on rare cell types.

The authors note that concordance between estimated relative gene expression of monocytes and corresponding ground-truth is not good. The source of this error is assumed to be individual and platform differences. Is it possible to provide more experiments to confirm that, possibly adding a new experiment with known ground-truth?

A comparison with existing methods that can estimate csGEPs is necessary, in particular BLADE [1] and CibersortX, wherever the requirement of meeting the criteria of having a certain number of bulk samples is fulfilled.

Minor comments and suggestions:

[1] Andrade Barbosa, B., van Asten, S.D., Oh, J.W. et al. Bayesian log-normal deconvolution for enhanced in silico microdissection of bulk gene expression data. Nat Commun 12, 6106 (2021). <https://doi.org/10.1038/s41467-021-26328-2>

Reviewer #2 (Remarks to the Author):

Chen et al developed a deep learning method TAPE (Tissue-AdaPtive autoEncoder) for precise deconvolution of bulk RNA-seq data in a short time. The authors claimed that TAPE can predict cell-type-specific gene expression tissue-adaptively in a fast and sensitive way, and TAPE is capable to provide biological significance when analyzing clinical data.

The deconvolution of bulk RNA-seq data for in-depth analysis is of great importance in biological research, especially tools that can be applied to analyze the huge amount of the existing clinical bulk RNA-seq data in a precise and fast way. The strength of TAPE relies on the deep learning algorithm and excellent performance (if truly as the authors claimed) compare with other state-of-the-art methods. The description of TAPE's capability to analyze clinical bulk RNA-seq profiles with biological significance is the shortcoming of this manuscript in its current stage. Overall, I found this study somewhat interesting, but premature and needs to be substantially revised before considering for publication.

Specific comments are as follows:

1. The authors apparently acknowledged that there are a series of methods like CIBERSORT, MuSiC, CIBERSORTx, Bisque, DWLS, RNA-Sieve and etc. that have been developed to deconvolute bulk RNA-seq data, but why only a subset of these tools were included in the benchmark comparison? The authors claimed "we compared the performance of TAPE to that of four representative deconvolution methods (published on famous journals with high performance)". It looks to me that both MuSiC and Bisque were published in Nature Communications, which I believe is a very decent journal with high reputation, but why did the authors exclude them?
2. When performing benchmark comparison, it is critical to compare the performance of every method in its optimized state, otherwise one has reasons to suspect that the advantages of TAPE over the other existing tools may not be caused by the algorithm itself, but could be caused by the author's incorrect use of the other algorithms, such as not entering the correct dataset (for instance, a pre-selected cell-type-specific gene expression profile) according to the algorithm tutorial, not optimize the options so that each method is best performed and etc. Without these details, I can hardly be convinced that TAPE is the best performed method. As far as I know, CIBERSORTx is a very good performed tool, but way poorly performed in the authors benchmark analysis (Fig 2a).
3. Following the above question, could it be possible that TAPE works better in a certain dataset but not the others. Since there are so many published datasets with paired single cell profiles as references, the authors should compare TAPE on a much larger data cohort, where 5 datasets is way below the expectation to prove the advantages of TAPE in cell type deconvolution with statistical significance.
4. I feel it is a bit over claim of TAPE's advantage to delineate the biological significance of the clinical data, by simply showing the tendency of certain cell types. Specifically, the authors claimed that "only TAPE could correctly predict proportions of neurons or microglia cells ranging from 0.32-0.55 and 0.06-0.12 respectively" (Fig. 3a), there are multiple cell types in these studies, and how about the prediction (of cell proportions) of other cell types? Are they all correctly predicted? Are there any experimental evidences as gold standard to support these predictions? Similarly, the prediction of MLR tendency (Fig. 3b), where the authors show only the "monocytes fraction", I wonder how about the other cell types, such as CD4+ T cell, CD8+ T cell, or B cell? Does TAPE also correctly predict the proportion and changes of these cell types in the mild, moderate and serious (should be severe) COVID-19 patient? Fig. 3c has the same problem. The logic here is, if TAPE, but no other tools (based on the assumption that these tools were corrected used), is capable of correctly predict the proportion and changes of cell subtypes, it should work for most (if not all) cell subtypes. By showing only one or two, and in this case not even a consistent cell type, in Fig. 3 is misleading, since without detailed analysis with statistical power, these results can simply be "cherry picking".
5. The capability of TAPE to "predict cell-type-specific gene expression at high-resolution" is what I

believe a unique advantage of TAPE over the other methods, however results showing in Fig. 4 greatly limited my overall enthusiasm. The authors predicted gene RAB11FIP5 as a DEG in NK cells (Fig. 4e), again, this is only one gene. What about the other known DEGs? What percentage and to what precision can TAPE predict the expressions of the known DEGs? Predict one gene within expectation can be shown as an example, but is way less enough to prove that TAPE is able to “predict cell-type-specific gene expression at high-resolution”. The authors should set up a gold standard where ground truth is known, and then evaluate the predictive power of TAPE on whatever indexes the authors interested to measure.

6. As we know, major cell types, such and T cells, B cells and etc. can be further divided into cell subtypes, I am curious to what extent can TAPE reach to, i.e. what is TAPE’s limitation to predict the gene expression in smaller cell subtype, which can be referred to “high-resolution”.

7. Fig. 5 is yet another very rough and premature result. Without solid evidence of the predictive power of TAPE, any TAPE prediction with biological implications can be wrong. Even if predictions from TAPE is right, what are the enriched pathways shown in Fig. 5a? What does this mean biologically (related to virus infection) if these pathways are enriched? How could people know whether these predictions were right? Are there any positive or negative controls? Without these detailed information, how can I be convinced?

8. I am curious, if TAPE is applied to analyze clinical bulk RNA-seq data, does the model needs to be re-trained? In other words, if the authors trained TAPE with normal bulk/single-cell RNA-seq data, can TAPE be applied to predict the proportion of cell types or the expression of genes in disease state? Because I suspect genes are usually differentially expressed in disease samples, even in the same cell subtypes, and wonder if the model needs to be re-trained based on disease data. If “Yes”, then the application of TAPE to “enable and accelerate the precise analysis of high-throughput clinical data in a wide range” will be limited.

9. Fig. 4b is problematic and needs to be further clarified. The author claimed “Interestingly, comparing the relative NRG1 expression value in bulk GEP, single-cell GEPs, and predicted GEPs (Fig 4b), we found that TAPE can successfully predict a high expression value of NRG1 in neurons while a low expression value of NRG1 in endothelial cells.” I am not able to draw this conclusion by looking at this figure. Besides, why OPC, Oligodendrocytes and Unknown have either the blue or red bar, instead of both? What does “unknown” mean in this figure? Fig. 4e, very seldom people will use $-\log_{10}(P\text{-value})$ in a box-plot to show the significance, and it should not be $p=xx$ in the box-plot, otherwise $p=0.00$ is considered as “very significant”.

We are very grateful to the reviewers for their thoughtful and thorough comments, which definitely helped us improve our paper significantly. We have revised the manuscript following all the comments. Below please find the point-by-point response to all the reviewers' comments.

Reviewer #1.

This manuscript proposes TAPE, a method consisting of a deep autoencoder to improve cell deconvolution. They propose to improve over Scaden, an ensemble of deep neural networks for cell deconvolution. The method uses reconstruction loss on real bulk samples and mean absolute error on simulated data to learn cellular fractions. The authors design the training paradigm to learn in two steps and their intuition is to adapt to the tissue being tested on. Additionally, the method is extended to predict cell-type specific gene profiles (csGEPs) both per-group/overall and per-sample/high-resolution. Comparisons with previous deconvolution approaches, CibersortX, RNAsieve and DWLS are given on simulated and real RNAseq datasets.

On the task of estimating cell proportions, TAPE generally seems to perform better than the methods compared, however, there are some ambiguities and experiments where more information is necessary. On the task of estimating cell-type specific gene expression profiles, the evidence in general seems weak.

The manuscript is well-written and easy to follow.

Answer:

Thank you very much for the excellent summary and the positive comments! As you have mentioned, our tool is competitive with the existing state-of-the-art methods, and it can serve as a useful tool for the community to perform precise deconvolution in a short time. In this revision, we have further improved the manuscript based on your comments.

1. For your concerns about the ambiguous experiments of deconvolution, we re-designed the pseudo-bulk experiments thoroughly based on your suggestions. Specifically, we present all our results about deconvolution using box plots for all the cell types rather than the overall metrics for each dataset to provide more details about TAPE's performance.

2. In addition, following your suggestions, we add more experiments and detailed analyses about cell-type-specific gene expression inference to benchmark TAPE more thoroughly.

We truly appreciated your insightful suggestions and comments, which have indeed helped us improve the quality of our manuscript significantly! Below is the point-by-point response to your suggestions and comments.

(1) To mimic real data, the authors mention that the simulated data includes gaussian noise and dropouts, however, no further information is given. It is necessary to know the levels of that noise considered.

Answer: Thank you very much for pointing that out! Yes, you are right. More detailed information should be provided to make the manuscript more readable.

As we know, the relationship between the real bulk data and single-cell data is very complex and highly non-linear. Thus, it is almost impossible to truly simulate the real bulk data by simply adding many single-cell profiles.

Previously, when generating the pseudo-bulk data, people usually only considered the linear combination of single-cell profiles without adding noises. The task will be too simple and far from the real case.

Here, we add noise to the simulated data because we want to make this pseudo-bulk test more difficult and closer to the real cases, instead of toy simulations. If we simply use the summation of single-cell profiles as pseudo-bulk data and deconvolve it using single-cell profiles as the reference, this task would probably degenerate into a linear regression task. To avoid the trivial situation, we add artificial noise to make this pseudo-bulk deconvolution task as difficult as real ones.

In this paper, we add Gaussian noise and dropouts. Specifically, We used 0.01 times random value generated from a Gaussian distribution with gene expression mean and variance for each gene as the Gaussian noise and randomly masked 20% genes for each pseudo-bulk sample. The reason why we choose these two kinds of noises is that both of them can represent the real-world sequencing noises. Gaussian noise is very common in different sequencing technologies and dropout is very common in single-cell RNA-seq data. Adding dropout into the pseudo-bulk RNA samples means that we increase the dropout rate in single-cell profiles since the pseudo-bulk RNA-seq data is the sum of many single-cell profiles. For example, we check the pseudo-bulk samples, it usually contains above 20,000 genes but 5,000 of them are zero since they are zero in the single-cell data. After adding dropout genes, it contains 8,000 zero-value genes. So, this noise means that we simulate a worse scenario where the single-cell profiles contain more dropouts. We have added the details into the revision accordingly, following your suggestions:

(section 2.2) “To avoid this task degenerating into a simple linear regression task, we added Gaussian noise (0.01 times random value generated from a Gaussian distribution with gene expression mean and variance for each gene) and randomly masked 20% genes for each pseudo-bulk sample.”

(2) The evidence provided to support the improvement in TAPE over existing algorithms is given over all cell-types present in a dataset. While this is a good comparison on evaluating a method overall, it does not provide any information about how TAPE performs on different cell-types. What happens if certain cell types are missing from the bulk or single cell data?

Answer:

Thanks for pointing out that and asking the related question!. We value your suggestions and have revised our manuscripts following your comments. Your concerns can be divided into three parts. Please see the detailed reply for each one below:

a) *it does not provide any information about how TAPE performs on different cell-types*

We noticed that the performance may vary among different cell types, and the overall indices can not provide detailed information. So, following your question, in this revision, we measure the performance for each cell type and use box plots to display the performance. Due to the limited space, we only show the detailed performance of each cell type in the

Monaco dataset here in the table below. For other datasets, please refer to the box plots in Figure R1-4.

Table R1. Detailed performance on each cell type, source data of Figure R1 “monaco”

Dataset	CellType	Method	CCC	L1error
monaco	Bcells	TAPE	0.774372	0.016203
monaco	CD4Tcells	TAPE	0.484331	0.035803
monaco	CD8Tcells	TAPE	0.45154	0.037436
monaco	Monocytes	TAPE	0.320453	0.087987
monaco	NK	TAPE	0.290329	0.035385
monaco	Bcells	Scaden	0.90498	0.011847
monaco	CD4Tcells	Scaden	0.242023	0.071814
monaco	CD8Tcells	Scaden	0.455439	0.044727
monaco	Monocytes	Scaden	0.19598	0.167292
monaco	NK	Scaden	0.410949	0.03824
monaco	Bcells	RNASieve	0.108525	0.052283
monaco	CD4Tcells	RNASieve	NA	0.281521
monaco	CD8Tcells	RNASieve	0.047161	0.161824
monaco	Monocytes	RNASieve	0.336347	0.090341
monaco	NK	RNASieve	0.004297	0.388941
monaco	Bcells	MuSiC	NA	0.106905
monaco	CD4Tcells	MuSiC	NA	0.281521
monaco	CD8Tcells	MuSiC	NA	0.213033
monaco	Monocytes	MuSiC	0.535987	0.06806
monaco	NK	MuSiC	0.003785	0.665789
monaco	Bcells	DWLS	0.16424	0.063149
monaco	CD4Tcells	DWLS	0.214363	0.144211
monaco	CD8Tcells	DWLS	0.445641	0.07766
monaco	Monocytes	DWLS	0.185611	0.200044
monaco	NK	DWLS	0.347201	0.049529
monaco	Bcells	CIBERSORTx	0.120926	0.068475
monaco	CD4Tcells	CIBERSORTx	0.550437	0.051943
monaco	CD8Tcells	CIBERSORTx	0.407114	0.075094
monaco	Monocytes	CIBERSORTx	0.675171	0.065061
monaco	NK	CIBERSORTx	0.244958	0.060843
monaco	Bcells	Bisque	0.169557	0.094422
monaco	CD4Tcells	Bisque	0.059738	0.303526
monaco	CD8Tcells	Bisque	-0.072055	0.218607
monaco	Monocytes	Bisque	0.349341	0.146178
monaco	NK	Bisque	-0.093793	0.225798

Despite that the performance may vary across different cell types, TAPE is the best measured by MAE. In addition, TAPE shows a relatively small variance of CCC, which

indicates TAPE's performance for all cell types is similar and robust, although it is not always the best method measured by CCC across different cell types. We replaced the original performance figure (Figure 2a in the original manuscript) with new figures (Figure R1 c,d. Figure R2 R3 R4) in the revision, as shown below.

Figure R1. Figure 2c & d in the manuscript

Figure R2. Appendix Figure 1 in the manuscript

Figure R3. Appendix Figure 2 in the manuscript

Figure R4. Appendix Figure 3 in the manuscript

b) *What happens if certain cell types are missing from the bulk data?*

That's a very insightful question! If a certain type of cell is missing in the bulk or single-cell data, the case would be more difficult than the normal deconvolution. We also noticed that and tested the case. When certain cell types are missing from the bulk data, whose ground truth value is considered as 0, usually, the predicted values of these cell types from TAPE will be small values close to 0. Following the setting of Scaden (Menden et al. *Science Advances*, 2020), we excluded the missing cell types and scaled the rest cell types (divide by the sum of the rest cell types) to calculate CCC values and MAE values with ground truth. For instance, the ROSMAP dataset only contains five cell types (Astrocytes, Endothelial, Microglia, Neurons, and Oligodendrocytes), but its human brain single-cell RNA-seq reference contains seven defined cell types (Astrocytes, Endothelial, ExNeurons, InNeurons, Microglia, OPC and Oligodendrocytes). When evaluating the performance of all the methods, we excluded OPC, combined ExNeurons with InNeurons to represent "Neurons", and scaled the results for the performance evaluation. This situation is also suitable for the mouse brain referenced deconvolution of the ROSMAP dataset. Following your question, to make the manuscript clearer, we have added how we process mouse brain referenced deconvolution in our manuscript:

(section 2.3) "For the "missing cell types" scenario, the ROSMAP dataset using mouse brain as reference is a good demonstration. The single-cell dataset of mouse brain has more cell types than the measured bulk ROSMAP dataset (Figure 2a). So, we directly filtered out extra cell types predicted by these methods and re-scaled the predicted fraction to make the summation is 1."

c) *What happens if certain cell types are missing from the single-cell data?*

You have pointed out a very practical question. Usually, obtaining single-cell data is more expensive than obtaining bulk data. But fortunately, platforms such as Human Cell Atlas (HCA), Single Cell Portal, and so on provide easy access to many publicly available single-cell datasets. On the other hand, this task can be difficult, and all the existing methods did not even consider it.

We find this situation interesting and explore more in the added experiments. Intuitively, we expect that the algorithm should transfer one cell type's proportion to its similar

cell types if this kind of cell is missing from the single-cell profiles. To test this situation, we design a new scenario called “similar transferring” in the pseudo-bulk test. This scenario is tested on the Marrow single-cell data from Tabular Muris as we mentioned before. In Marrow dataset, it contains two similar B cells: late-pro B cell and immature B cell. We delete one kind of B cell from the single-cell profiles and use them as a reference to deconvolve pseudo-bulk data, which includes both kinds of B cells. The results show that TAPE can transfer one kind of B cell’s weight to another if it is missing from the reference (Figure R5, “missing transferring”).

Figure R5. Figure 2c in the manuscript

We have added all the descriptions about our re-designed pseudo-bulk test into section 2.2 of the revision as the following:

“Since a real bulk dataset with its corresponding cell type fractions assessed by traditional experimental methods (e.g., flow cytometry) is rare, and it is hard to analyze how the batch effect would affect deconvolution performance, it is necessary to conduct a pseudo-bulk test. The pseudo-bulk data are generated *in silico* from single-cell GEPs with ground truth (pre-defined cell type proportions). That is, pseudo-bulk data are the summation of many single-cell profiles. To avoid this task degenerating into a simple linear regression task, we added Gaussian noise (0.01 times random value generated from a Gaussian distribution with gene expression mean and variance for each gene) and randomly masked 20% genes for each pseudo-bulk sample. The single-cell profiles are from *Tabular Muris* [17], a cell atlas for mouse with two different sequencing techniques, 10X-seq (UMI-based method) and Smart-seq (counts-based method). This cell atlas is a good resource for us to simulate the batch effect. Thus, in the following experiments, we used one protocol’s single-cell data as the reference to predict another protocol’s pseudo-bulk data. Here we only selected three tissues/organs from *Tabular Muris* because they have the largest number of shared cell types across different protocols in all the tissues/organs. Specifically, “Limb Muscle” has 6 cell types, “Marrow” has 7 cell types, and “Lung” has 9 cell types. To fully exploit the advantages of pseudo-bulk data, we defined three deconvolution scenarios: “normal”, “rare”, and “similar”. For the “normal” scenario, all the cell type proportions are randomly generated, while in the “rare” scenario, some cell types’ fractions are set below

3%. To be specific, skeletal muscle satellite cells and endothelial cells are set to be rare cell types in “Limb Muscle”; monocyte and hematopoietic precursor cells are set to be rare cell types in “Marrow”; T cells, natural killer cells and ciliated columnar cells of tracheobronchial trees are set to be rare cell types in “Lung”. In the “similar” task, we only used “Marrow” because there are two similar subtypes of B cell in it: “late-pro B cell” and “immature B cell”. Here, we expect that if we delete one kind of B cell from the single-cell reference, the predicted fraction of the other type of B cell would still be similar to the summation of the two kinds of B cell. That is, we expect the method could correctly transfer the weight of one kind of B cell to another.”

Regarding the real dataset, as we mentioned in the Method part, section 4.1, second paragraph:

“More specifically, the unknown fraction was calculated by one minus the sum of known proportions and cell types of the same kind were added together to fit cell types in training data. For example, monocytes C, monocytes I, and monocytes NC are different kinds of monocytes, so their fractions will be added together as the total fraction of monocytes.”

We actually add the similar cell type up to calculate the performance for monocyte if the single-cell profiles do not have concrete subtypes. But, in practice, if we don’t have prior knowledge about what cell types are in the bulk data, we can not exclude extra cell types in single-cell profiles, and the prediction would be wrong. Notice that this situation is less likely to happen, and currently, all the algorithms we tested can not solve this problem.

In general, for better performance, we suggest that single-cell data and bulk data should include the same cell types. Through our experiments, we have the following findings. If some cell types are missing from single-cell profiles, the deconvolution algorithm will allocate weights of missing cell types to their similar cell types; if some cell types are missing from the bulk data, it would be difficult, but our algorithm can guarantee that the relative cell type ratio of existing cell types is almost correct (since in the previous analysis, we all exclude extra cell types to calculate the existing cell type ration and the result is good). Considering these situations may happen if users are not careful, we have added a special warning on the Github page to remind users of the correct usage of deconvolution algorithms (Figure R6).

Usage

Required Files:

1. single-cell reference: txt format, indices are cell types, columns are gene names
2. bulk data: tabular format, needed to specify the separation ('\\t', ',' or others), indices are sample names, columns are gene names
3. gene length file: used to scale the expression value, columns should contain: [Gene name, Transcript start (bp), Transcript end (bp)]. This is provided in ./data/ directory.

Warning: single-cell reference and bulk samples should contain the same cell types

```
# basic example
from TAPE import Deconvolution
SignatureMatrix, CellFractionPrediction = \
    Deconvolution(sc_ref, bulkdata, sep='\\t',
                  datatype='counts', genelenfile='./GeneLength.txt',
                  mode='overall', adaptive=True,
                  save_model_name=None,
                  batch_size=128, epochs=128)
```

Figure R6. Special warnings on the Github page

(3) Since TAPE uses reconstruction loss on real bulk to aid in learning, how does the sample size of the tissue bulk data affect the performance of TAPE?

Answer:

Thank you for asking this question! Yes, you are correct. Usually, the sample size will affect the results if the tissue bulk data are included in the training process. But, regarding our algorithm, for the prediction of cell-type fractions, it would not be affected by the sample size because we use about 5,000 pseudo-bulk samples for training, and we only use the real bulk data for adaptive prediction. Normally, the model's performance shall be affected once the reconstruction loss is considered. However, we use enough pseudo-bulk data in the training process, and the "reduction" method of the loss function `torch.nn.functional.l1_loss()` is "mean" not the "sum", so the sample size term in our reconstruction loss is considered per sample and would not be affected by the sample size. The model's parameters are fixed after training with pseudo-bulk data, and we just use this model to predict cell-type fractions.

To further resolve your concern, we added an additional experiment on "Lung" (pseudo-bulk dataset) in the revision, and the results are shown in the below table.

Table R2. Sample size of real bulk does not affect the performance of TAPE

Sample size of test data	20	40	60	80	100
Overall CCC	0.42494	0.49177	0.45102	0.41475	0.47003
Overall MAE	0.09995	0.09061	0.08586	0.09911	0.09152

That is why the performance has very small changes among different sample sizes in the table (these changes should be mainly caused by the pseudo-bulk samples' differences).

As for the GEP estimation part, our reconstruction loss is also considered in a per-sample way. That's why we can train the model per sample to achieve the "high-resolution" mode.

To make the training process clearer, we added the below details to the revised manuscript.

(section 4.2.5) “As previously stated, there are two stages of training in TAPE. The first is the training stage, where we use about 5,000 pseudo-bulk samples for training. We use MAE between prediction and ground truth to optimize the parameters of the encoder and MAE between the reconstructed input and the original input to optimize both the decoder and the encoder.”

(4) *It is unclear how the method behaves on cell-types that are highly related. Any evidence supporting that will be helpful. Similarly, how does TAPE perform on cell fractions below 5% and 2%? Once of the main unresolved issues is good performance on rare cell types.*

Answer: Thank you very much for this remarkable question! Deconvolution on the highly related cell types and rare cell types is indeed very difficult. In the original manuscript, we had some attempts on the real data. But following your questions and the comments from Reviewer 2, we performed a more thorough and systematic evaluation of TAPE and other methods in these specific situations in this revision. As we have mentioned in (2), to explore the performance of TAPE in more detail, we defined three deconvolution scenarios: “normal”, “rare”, and “similar”. The “similar” and “rare” scenarios can help to explain the following questions separately.

a). *How the method behaves on cell-types that are highly related?*

Since highly related cell types are common in real-world data, we should give further exploration on such situations. In the “similar” scenario, we use “Marrow” to compare the deconvolution performance with other methods. This dataset has two types of B cells in it: “late-pro B cell” and “immature B cell”, which satisfies the settings of highly related cell types. As shown in the below figure, the calculated CCC and MAE values suggest that TAPE is the best algorithm. In detail, TAPE can not only distinguish each cell type when both kinds of the B cell are in the reference but also transfer one B cell’s proportion to another if this B cell is missing from the reference (Figure R7).

Figure R7. Figure 2c in the manuscript

Following your question, we have added the corresponding description in the revision.

(section 2.2) “As for the “similar” scenario, we investigate the performance of two kinds of B cell in the “normal” scenario and what would happen if we delete one cell type from the reference. The results show that TAPE is the most robust algorithm and can distinguish cell subtypes when both kinds of the B cell are in the reference (Figure 2c, “similar distinguishment”).”

b). How does TAPE perform on cell fractions below 5% and 2%?

Following your comments, we have added additional related experiments in the “rare” scenario, where some cell-type fractions are set below 3%. To be specific, skeletal muscle satellite cells and endothelial cells are set to be rare cell types in “Limb Muscle”; monocyte and hematopoietic precursor cells are set to be rare cell types in “Marrow”; T cells, natural killer cells and ciliated columnar cells of tracheobronchial trees are set to be rare cell types in “Lung”. As you have pointed out, deconvolution on the rare cell types is indeed the bottleneck of all the deconvolution methods. As shown in Figure R8, if we only calculate the metrics for pre-defined rare cell types, all the methods cannot predict a satisfying concordance with the ground truth. Interestingly, the CCC value is pretty low in all methods, but the MAE is comparable to the “normal” scenario, which indicates these methods can predict a value near the ground truth but are not correlated with each other. Regarding TAPE, although it is not designed to focus on rare cell types, its performance (MAE) is comparable to DWLS, which focuses on rare cell types. But indeed, the CCC values from TAPE should be further improved in this specific situation. We thank you a lot for pointing out this challenging problem. In the future, we will try to improve this situation by minimizing the relative MAE for each cell type rather than the overall MAE in further package updates.

Figure R8. Figure 2c in the manuscript

Following your question and our experiments, we also added the explanation of the “rare” scenario to the revision.

(section 2.2) “In the “rare” scenario, we only display the metrics for pre-defined rare cell types. The results show that all the methods can not result in a satisfying concordance

between prediction and ground truth in this scenario (Figure 2c). Interestingly, although the CCC values are pretty low with those methods, their MAEs are comparable to those in the “normal” scenario, which indicates that those methods can predict a value near ground truth but are not correlated with each other. Though TAPE is not the best algorithm in this scenario, its performance is comparable to DWLS, which focuses on rare cell types.”

Understandably, TAPE could be further improved, especially on the rare cell type deconvolution, which is the bottleneck of all the deconvolution methods, as you have pointed out. In the future, we will try to improve this situation by minimizing the relative MAE for each cell type (like what DWLS does) rather than the overall MAE in further package updates (Tsoucas et al. Nature Communications, 2019). Because the key feature distinguishing our current work from previous methods is inferring the high-resolution cell-type-specific GEPs, we added the rare cell type deconvolution issue in the discussion part as future work, following your insightful question.

(section 3) “Although we have shown that TAPE’s deconvolution performance is pretty good in many scenarios, we find that it would perform poorly in the “rare” scenario since it shows a low CCC value. But, in the benchmarking process (Figure 2c), the results show that other tools’ performance also drops in the “rare” scenario. This phenomenon indicates the “rare” scenario has not been solved well by current methods and needs to be addressed in future works.”

(5) The authors note that concordance between estimated relative gene expression of monocytes and corresponding ground-truth is not good. The source of this error is assumed to be individual and platform differences. Is it possible to provide more experiments to confirm that, possibly adding a new experiment with known ground-truth?

Answer: Thank you very much for the comment! Your concern is very reasonable. Additional experiments should be added to support the assumption. Following your comment, another experiment is added to confirm our assumption. We test TAPE on a simulated dataset with single-cell profiles as ground truth (Figure R9 a). We can see that our method can predict a good concordance with the pseudo-bulk data but not good for only monocytes in the real bulk data. Considering the real bulk data is not from the same source as single-cell data, and the other five kinds of cell types have a good concordance, we draw the conclusion that this discordance of monocytes is caused by individual differences, and these results also prove that our method can predict adapted results rather than simply copy the single-cell data.

Considering your comment, we have added the below figure and the description in the revision to make the discussion more solid.

Figure R9. Figure 4a,b in the manuscript **a** Concordance between the predicted relative gene expression value in simulated bulk data and the relative gene expression value from single-cell data. The relative gene expression value is the original expression value after Log_2 and $\text{MinMaxScaler}()$ transformation. **b** Concordance between the predicted relative gene expression value in real bulk data and the relative gene expression value from single-cell data.

(section 2.6) “After testing TAPE on a simulated dataset with a single-cell profile as ground truth (Figure 4a) and considering the good concordance in other five cell types, we draw the conclusion that this distortion is caused by the individual difference.”

(6) A comparison with existing methods that can estimate csGEPs is necessary, in particular BLADE [1] and CibersortX, wherever the requirement of meeting the criteria of having a certain number of bulk samples is fulfilled.

Answer: Thank you very much for this excellent comment and for mentioning the recently published work! Estimating csGEPs is an important question in this field, and only a few methods have considered this problem. In the previous version, we showed the correctness of the predicted csGEPs of TAPE by measuring the concordance between the predicted gene expression value of each cell type and the original gene expression value obtained from single-cell RNA. We should also compare TAPE with more methods designed for this. Following your suggestion, we added additional experiments as described below.

(a) Thank you for mentioning the excellent work, BLADE! For BLADE, we ran it with 1,000 selected signature genes that are produced by the CSx. Usually, CSx would select about 3,000 genes as signature genes. But it cannot be all used because BLADE has higher time complexity. For efficiency, we only randomly selected 1,000 genes from all the signature genes. Unfortunately, in the pseudo-bulk test, BLADE requires half a day to deconvolve 100 samples. During the experiment, BLADE is running on a 48-core platform using the full computational resource. Due to the limited recourse and time, we would not be able to run it on the larger datasets, so we have to exclude it from the comparison. For your information, the time consumption of BLADE and comparison among BLADE, TAPE, and other methods we considered is listed in the below table.

```

May  1 11:49 Limb_Muscle_counts2umi_pred.csv
Apr 29 18:37 Limb_Muscle_counts_sigm.txt
May  1 10:15 Limb_Muscle_umi2counts_pred.csv
Apr 29 18:39 Limb_Muscle_umi_sigm.txt
May  2 13:26 Lung_counts2umi_pred.csv
Apr 29 18:33 Lung_counts_sigm.txt
May  2 09:56 Lung_umi2counts_pred.csv
Apr 29 18:42 Lung_umi_sigm.txt
May  3 02:40 Marrow_counts2umi_pred.csv
Apr 29 18:35 Marrow_counts_sigm.txt
May  2 21:13 Marrow_umi2counts_pred.csv
Apr 29 18:43 Marrow_umi_sigm.txt

```

Figure R10. Screenshot of the time interval for BLADE to do all the tasks, it finishes 5 tasks (each one has 100 samples) from 11:49 May 1 to 2:40 May 3.

Table R3. Time consumption on the test of “Lung umi2counts” task for each method, unit is second

tested on Lung umi2counts	100	200	300	400	500	600	700	800
TAPE	112.72	107.58	110.45	106.56	114.79	146.63	157.50	122.82
Scaden	340.16	352.71	358.82	353.40	347.19	351.96	393.23	342.95
RNA sieve	120.20	158.52	198.18	231.03	275.40	315.85	348.13	398.68
CIBERSORTx (web)	821.68	1664.05	2595.24	NA	NA	NA	NA	NA
DWLS	267.85	530.96	839.19	1064.65	1314.96	1581.76	1913.18	2133.69
MuSiC	14.74	17.21	21.20	26.13	28.73	34.15	39.23	44.02
Bisque	12.39	12.42	12.81	13.45	13.87	14.67	14.99	15.14
BLADE (Lung umi2counts)	79620	NA	NA	NA	NA	NA	NA	NA

Although we did not show the results of BLADE eventually, we still thank the reviewer for mentioning BLADE as an interesting work on the deconvolution task. BLADE is an interpretable Bayesian method, and it can solve all the tasks we expected. We have mentioned it in the manuscript for the reader’s interest.

(section 2.7) “Of note, the recently published method, BLADE [13], can do this task too, but we did not benchmark BLADE in our experiments, considering its high time complexity.”

(b) As for CIBERSORTx, we now added it to estimate the cell-type-specific GEPs in comparison to TAPE. To fully evaluate the ability to estimate differentially expressed genes (DEGs) of these two methods, we added a new experiment to test their classification ability. Following the settings in CIBERSORTx, we created a series of pseudo-bulk datasets with foldchange gradients and cell proportion gradients. After obtaining the GEP of the CD8 T cell, we used a two-sided t-test to detect DEGs ($p < 0.05$), and we chose the area under the receiver operating characteristic curve (AUROC) as the criterion. The results in (Figure R11 e,f) show that TAPE can successfully predict cell-type-specific DEGs correctly (with good

sensitivity) and selectively (with good specificity), while CIBERSORTx almost fails on this task. The reason why it is hard for CIBERSORTx to infer the 100 DEGs properly is that CIBERSORTx usually focuses on signature genes that have bigger statistical power and are easily detectable if they are differentially expressed, but in this task, we randomly select 100 genes that are probably insignificant genes. Furthermore, we also tested whether CIBERSORTx can predict RAB11FIP5 as DEG in NK cells rather than other cell types. The results in (Figure R11 h) show that CIBERSORTx sensitively predicts the differential expression of RAB11FIP5 in NK cells, while there is not sufficient statistical power for it to estimate the expression value of RAB11FIP5 in other cell types (expression value of RAB11FIP5 is NaN in other cell types).

Figure R11. Figure 4 in the manuscript

Following your suggestion, we have added the same figures to Figure 4 and the description in the revision accordingly.

(section 2.7) "Following the settings in CIBERSORTx [9], we selected 100 cells across four cell types (CD8 T cell, Natural Killer (NK) cell, B cell, and Monocyte) from PBMC single-cell

data and another 10 cells from human brain single-cell dataset as noise to compose the pseudo-bulk data. Then we randomly selected 100 genes among 10,000 genes in CD8 T cells as up-regulated genes to adjust their expression. Each pseudo-bulk dataset contains 50 pseudo-bulk samples and half of them are composed of up-regulated CD8 T cells. The cell proportion of CD8 T cells in pseudo-bulk data ranges from 5% to 30%, and the foldchange of up-regulated genes ranges from 1.5 to 5. In total, we created a series of pseudo-bulk datasets with foldchange gradients and cell proportion gradients. After obtaining the GEP of CD8 T cells, we used a two-sided t-test to detect DEGs ($p_{\text{adj}} < 0.05$). So, this task is essentially a binary classification task, and we naturally chose area under receiver operating characteristic curve (AUROC) as the criterion. The results show that (Figure 4f), TAPE can successfully predict cell-type-specific DEGs correctly (with good sensitivity) and selectively (with good specificity) while CIBERSORTx almost fails on this task. The overall trend is that algorithms can easily recognize DEGs in one cell type if the proportion of this cell type or the foldchange of DEGs is high. Interestingly, using DEGs in bulk as the reference, we can see that TAPE can even predict DEGs not shown up in bulk samples but in CD8 T cells (the maximum AUROC of TAPE is higher than the maximum AUROC of bulk samples in Figure 4e,f). In the original article [9], CIBERSORTx has also demonstrated its great ability in DEGs prediction, the reason why it failed in this task is that CIBERSORTx usually focuses on signature genes which have bigger statistical power and are easily detectable if they are differentially expressed, but in this task, we randomly selected 100 genes which are probably insignificant genes; therefore, it is hard for CIBERSORTx to infer the 100 DEGs properly. In contrast, TAPE has shown its ability in predicting DEGs even when they are not significant genes, which means TAPE has a broader application potential than CIBERSORTx. Of note, the recently published method, BLADE [13], can do this task too, but we did not benchmark BLADE in our experiments, considering its high time complexity.”

(section 2.7) “So, we used TAPE and CIBERSORTx to tissue-adaptively deconvolve the HIV PBMC data [35]. To avoid batch effects and harmful effects caused by the low-quality single-cell data, we combined data6k, data8k, and data10k PBMC single-cell data [30, 37, 38] as the reference. After obtaining the predicted GEPs for each sample at high resolution, we calculated the adjusted p-value and fold change for each cell type (Fig 4h). The results show that both TAPE and CIBERSORTx successfully predict that RAB11FIP5 is differentially expressed in NK cells.”

Reviewer #2.

Chen et al developed a deep learning method TAPE (Tissue-AdaPtive autoEncoder) for precise deconvolution of bulk RNA-seq data in a short time. The authors claimed that TAPE can predict cell-type-specific gene expression tissue-adaptively in a fast and sensitive way, and TAPE is capable to provide biological significance when analyzing clinical data.

The deconvolution of bulk RNA-seq data for in-depth analysis is of great importance in biological research, especially tools that can be applied to analyze the huge amount of the existing clinical bulk RNA-seq data in a precise and fast way. The strength of TAPE relies on the deep learning algorithm and excellent performance (if truly as the authors claimed) compare with other state-of-the-art methods. The description of TAPE's capability to analyze clinical bulk RNA-seq profiles with biological significance is the shortcoming of this manuscript in its current stage. Overall, I found this study somewhat interesting, but premature and needs to be substantially revised before considering for publication.

Specific comments are as follows:

Answer: We truly appreciated your thorough and constructive review of the paper and for taking the time to go through many of the details!

1. For your concerns about the performance compared with other methods, we have added more thorough experiments with hyperparameter tuning results and evaluated performance for each cell type rather than the previous overall index. Moreover, we further compared our method with CIBERSORTx on the "high-resolution" GEPs estimation task. On the simulated data, we proved that our method is more sensitive than CIBERSORTx, and on the real data, both of the methods can discover the DEGs accurately.

2. In addition, for your questions about the clinical data analysis results, we have added more details to explain the background and avoid misleading results. Specifically, following your suggestions, we examined each cell type's proportion change in section 2.5 (question 4) and added the statistically significant results. In our GEPs' analysis (section 2.7, question 5), we really thank you for your helpful suggestions. We designed a new simulated-data test to evaluate our method's performance and limitations on the DEGs detection task. As for the ssGSEA analysis (section 2.8, question 7), we added more discussion and specific results to show that our method can be useful in discovering the biologically meaningful pathways.

Based on your comments, our manuscript has been largely revised and improved. Below we respond to all of your major concerns point by point.

(1) The authors apparently acknowledged that there are a series of methods like CIBERSORT, MuSiC, CIBERSORTx, Bisque, DWLS, RNA-Sieve and etc. that have been developed to deconvolute bulk RNA-seq data, but why only a subset of these tools were included in the benchmark comparison? The authors claimed "we compared the performance of TAPE to that of four representative deconvolution methods (published on famous journals with high performance)". It looks to me that both MuSic and Bisque were published in Nature Communications, which I believe is a very decent journal with high reputation, but why did the authors exclude them?

Answer: We thank the reviewer very much for pointing it out. We have revised our manuscript accordingly and added the comparison with the two methods. We did not include them in the original manuscript because their deconvolution performances were not so good in the previous reports (Erdmann-Pham et al. *Genome Research*, 2021, Menden et al. *Science Advances*, 2020). And in the original manuscript, we compared TAPE with the previous state-of-the-art methods. But your concern is very reasonable. We should perform a more thorough comparison with all these methods. Based on your suggestions, we have added additional experiments in this revision.

The detailed comparison of TAPE and all the methods considered is shown in the figure below (Figure R12c & d). Figure R12c shows the deconvolution results on simulated data. Each box contains metric values for all the cell types considered in all the tissues. Different color refers to different methods. We defined three deconvolution scenarios: “normal”, “rare”, and “similar”. As for the “normal” scenario, all the cell type proportions are randomly generated, while in the “rare” scenario, some cell types’ fraction is set below 3%. In the “similar” task, there are two situations, “similar distinguishment” and “similar transferring”. The results show that TAPE is the best algorithm and can distinguish cell subtypes when both kinds of the B cell are in the reference (“similar distinguishment”), and TAPE can transfer one B cell’s proportion to another if this kind of B cell is missing from the reference (“similar transferring”). Figure R12d shows the deconvolution results on real data. Of note, the hyperparameters for all the methods have been optimized in these experiments. Details could be referred to answer to the next question.

Figure R12. Figure 2c & d in the manuscript

Following your question, we have added the detailed statements of setting for running the two new methods in the manuscript section 4.4.

“For MuSiC, we installed the R package and ran it with default settings following its tutorial. Of note, MuSiC claims it can take advantage of multi-subject single-cell profiles to improve deconvolution performance. But in our pseudo-bulk test, we do not have multi-subject single-cell data. To meet its requirement, we randomly assigned one dataset to two virtually different sources. In the real bulk test, we first combined PBMC data6k and data8k as the reference to deconvolve PBMC bulk dataset, but MuSiC failed to predict them properly. The CCC value was negative, which was reported by a previous study [9]. Then we only used PBMC data8k as the reference and assigned it to two virtually different datasets to deconvolve PBMC bulk data. With only one source single-cell profile, the CCC value was normal. Thus, we only displayed the one-subject single-cell reference results above.”

“For Bisque, we installed the R package BisqueRNA and ran it with default settings following the example provided by the author. In the default mode of the 'Reference-based decomposition' mode, Bisque filters low-variance genes first and uses the left genes for decomposition, so we input all the genes without specifying some marker genes.”

(2) When performing benchmark comparison, it is critical to compare the performance of every method in its optimized state, otherwise one has reasons to suspect that the advantages of TAPE over the other existing tools may not be caused by the algorithm itself, but could be caused by the author's incorrect use of the other algorithms, such as not entering the correct dataset (for instance, a pre-selected cell-type-specific gene expression profile) according to the algorithm tutorial, not optimize the options so that each method is best performed and etc. Without these details, I can hardly be convinced that TAPE is the best performed method. As far as I know, CIBERSORTx is a very good performed tool, but way poorly performed in the authors benchmark analysis (Fig 2a).

Answer: Thanks for this comment! We totally understand your concerns. Although we did not show it in the original manuscript due to the limited space, we had done the hyperparameter searching previously. But following your comments, we have done a more thorough optimization for all the compared methods in the revision. The details are shown below.

In the benchmarking settings, we compared TAPE with six methods. Besides carefully following their tutorial, we also added hyperparameter fine-tuning experiments for all the methods on the Monaco's dataset to explore whether those parameters matter in the outcomes. Each method's optimized state is described below.

For Bisque, we ran it following the example Jupyter Notebook file on its GitHub repository. By default, Bisque is designed to deal with all genes (after filtering out zero variance genes), so we input all the genes without specifying some marker genes. Since Bisque uses non-negative least-squares regression (NNLS) to estimate cell proportions from the bulk RNA-seq data, there are no hyperparameters left to be tuned (no other parameters in the main function, see https://github.com/cozygene/bisque/blob/master/R/reference_based.R). Thus, we assume that Bisque is in the optimized state.

For Scaden: This is a deep learning method that uses almost all genes as input (because it has a variance cutoff, we use the default settings). The simulation step is processed according to its recommended settings, 500 cells per pseudo-bulk sample and 5000 samples in total. The training step has hyperparameters like learning rate, batch size, and training steps for users to tune. We tested these hyperparameters' effects, as shown in the table below. We also tried to tune network architecture, but the authors did not open the interface, so we tuned the above hyperparameters. We found that, following its default settings, Scaden can achieve a good performance. It's not a surprise because, in the original paper (Menden et al. *Science Advances*, 2020), the authors have searched these parameters before. So, we just used the default settings following the tutorials (<https://scaden.readthedocs.io/en/latest/usage.html>).

Table R4. Hyperparameters tuning for Scaden

parameters	batch size	128	64	64	64	64	128	128	128
	learning rate	1.00E-04	1.00E-04	1.00E-04	1.00E-05	1.00E-05	1.00E-04	1.00E-05	1.00E-05
	steps	5000	2000	5000	2000	5000	2000	2000	5000
metrics	CCC_overall	0.49	0.47	0.48	0.56	0.51	0.49	0.51	0.53
	MAE_overall	0.07	0.07	0.08	0.06	0.07	0.07	0.07	0.07

	CCC_average	0.37	0.32	0.32	0.33	0.31	0.31	0.33	0.31
	MAE_average	0.07	0.07	0.08	0.06	0.07	0.07	0.07	0.07

For RNAsieve, it does not have detailed documentation, so we ran it following its example code. In practice, we can produce the same result as its example code, but the results of RNAsieve in pseudo and the real bulk test are not good. The benchmarking code is available in the GitHub repository (<https://github.com/poseidonchan/TAPE>). We adjust the values of the two parameters (trim_percent & gene_thresh) in the core function, “model_from_raw_counts”, within a certain range around the default values (highlighted in yellow), and the test results shown in the table below indicate that the outcome of RNAsieve can not be significantly improved by just tuning the hyperparameters. Therefore, we followed the default setting from the author to derive other results.

Table R5. Hyperparameters tuning for RNAsieve

parameters	trim_percent	0.02	0.10	0.05	0.01	0.02	0.10	0.05	0.01	0.02	0.10	0.05	0.01
	gene_thresh	0.20	0.20	0.20	0.20	0.10	0.10	0.10	0.10	0.30	0.30	0.30	0.30
metrics	CCC_overall	-0.10	-0.08	-0.08	-0.10	-0.06	-0.07	-0.08	-0.06	-0.11	-0.10	-0.11	-0.10
	MAE_overall	0.19	0.17	0.17	0.19	0.18	0.20	0.20	0.18	0.18	0.18	0.18	0.18
	CCC_average	nan	nan	nan	nan	nan	nan	nan	nan	nan	nan	nan	nan
	MAE_average	0.19	0.17	0.17	0.19	0.18	0.20	0.20	0.18	0.18	0.18	0.18	0.18

For MuSiC, it claims that it can take advantage of multi-subject single-cell profiles and find the informative genes automatically. Considering the MuSiC’s claim, which can automatically allocate weights to informative genes, we did not input a marker gene list. Besides the maker genes, it also has parameters for users to change, including “nu”, “centered”, and “normalized”. We found that these parameters don’t affect the performance significantly. MuSiC still can not predict a proper CCC for some cell types (it predicts zero for some cell types), leading to the NaN value for the average CCC.

Table R6. Hyperparameters tuning for MuSiC

parameters	nu	0.0001	0.0000	0.0010	0.0100	0.1000	0.0001	0.0000	0.0010	0.0100	0.1000	0.0001	0.0010	0.0100	0.1000	0.0000	0.0001	0.0000	0.0010	0.0100	0.1000	
	centered	FALSE	FALSE	FALSE	FALSE	TRUE	TRUE	TRUE	TRUE	TRUE	FALSE	FALSE	FALSE	FALSE	FALSE	TRUE	TRUE	TRUE	TRUE	TRUE	TRUE	TRUE
	normalized	FALSE	FALSE	FALSE	FALSE	FALSE	FALSE	FALSE	FALSE	FALSE	FALSE	TRUE	TRUE	TRUE	TRUE	TRUE	TRUE	TRUE	TRUE	TRUE	TRUE	TRUE
Metrics	CCC_overall	-0.0275	0.0249	-0.0262	0.1967	0.2373	0.0986	0.0821	0.1222	0.2324	0.2219	-0.0165	-0.0650	-0.0924	-0.0898	0.0044	-0.0165	0.0280	0.0491	0.0302	-0.0162	
	MAE_overall	0.1796	0.1769	0.1683	0.1435	0.1731	0.1885	0.1954	0.1696	0.1496	0.1901	0.1739	0.1741	0.1771	0.1756	0.1712	0.1739	0.1821	0.1739	0.1786	0.1911	
	CCC_average	NaN	NaN	NaN	NaN	NaN	NaN	NaN	NaN	NaN	NaN	NaN	NaN	NaN	NaN	NaN	NaN	NaN	NaN	NaN	NaN	
	MAE_average	0.1796	0.1769	0.1683	0.1435	0.1731	0.1885	0.1954	0.1696	0.1496	0.1901	0.1739	0.1741	0.1771	0.1756	0.1712	0.1739	0.1821	0.1739	0.1786	0.1911	

For DWLS, the first step is to choose the signature genes. It provides two flavors, “MAST” and “Seurat” for us to use. In practice, we found that for the pseudo-bulk test, the “Seurat” flavor can lead to a better result, while the “MAST” flavor can lead to a better performance in real datasets. So, we used the “Seurat” flavor for the pseudo-bulk test and the “MAST” flavor for the real bulk dataset. The p-value cutoff is 0.01, and the foldchange cutoff is 0.5. After selecting the signature genes, we used them to deconvolve bulk RNA-seq samples one by one. Of note, for the stability, we used nu-SVR to obtain an initial estimation instead of OLS (the solve.QP function in R usually has many problems). We tested the performance for the

two different initial estimation methods, and the performance is the same in the pseudo-bulk test. We think this is mainly due to the nature of the iterative method: an initial estimation in a proper range can lead to the same result when the algorithm is converged. Considering the marker gene selection process may affect the performance significantly, we used different parameters to test DWLS' performance on the real PBMC datasets from (Monaco et al. *Cell Reports*, 2019) According to the results below, we found the "p-value cutoff" and "diff cutoff" do not affect the performance significantly, so we use the default settings.

Table R7. Hyperparameters tuning for DWLS

parameters	p-value cutoff	0.0100	0.0100	0.0500	0.0500
	diff cutoff	0.5	0.5	0.5	0.5
	flavor	MAST	Seurat	MAST	Seurat
Metrics					
	CCC_overall	0.4056	0.4358	0.4056	0.1539
	MAE_overall	0.0979	0.1037	0.0979	0.2040
	CCC_average	0.2894	0.1515	0.2894	0.0554
	MAE_average	0.0979	0.1037	0.0979	0.2040

For CIBERSORTx, we ran it following the tutorial on the website. We notice that it has some potential hyperparameters at each step. When we generate the signature matrix, it has hyperparameters like the maximum condition number, the q-value cutoff, sampling fraction, and cell type fractions. When we impute cell fractions, it has hyperparameters like batch correction mode, quantile normalization, and permutation. When we impute cell-type-specific GEPs at high resolution, it has hyperparameters like batch correction and quantile normalization. According to our prior knowledge, we exclude some hyperparameters from the fine-tuning test. In the original paper, the success of CIBERSORTx mainly relies on the success of the heuristical batch correction method. So, we all use S-mode for deconvolution (since we only used single-cell data as reference). As for the quantile normalization, the website has shown that it is recommended to be disabled for the RNA-seq data, and we follow the instruction in all the tests. Considering the q-value cutoff is conventional in statistics, so we tested the "conditional number" and "quantile normalization" in the following table.

Table R8. Hyperparameters tuning for CIBERSORTx

parameters	Kappa	14.63	8.98	14.63	8.98	14.63	8.98	14.63	8.98	5	2
	quantile normalization in generating signature matrix	FALSE	TRUE	FALSE	TRUE	FALSE	TRUE	FALSE	TRUE	FALSE	FALSE
	S-mode correction	TRUE	TRUE	TRUE	TRUE	FALSE	FALSE	FALSE	FALSE	TRUE	TRUE
	quantile normalization in deconvolution	FALSE	FALSE	TRUE	TRUE	FALSE	FALSE	TRUE	TRUE	FALSE	FALSE
Metrics	CCC_overall	0.6167	0.4955	0.6108	0.5517	0.2260	0.0953	0.2389	0.1061	0.6817	0.5788

	MAE_overall	0.0683	0.0910	0.0659	0.0753	0.0866	0.0996	0.0875	0.0978	0.0603	0.0772
	CCC_average	0.3235	0.2254	0.3189	0.2542	0.3258	0.2504	0.3282	0.2594	0.3477	0.0772
	MAE_average	0.0683	0.0910	0.0659	0.0753	0.0866	0.0996	0.0875	0.0978	0.0603	0.0772

These results show that the default settings are quite near the optimized state. With the “Kappa” (condition number) restricted below 5, the deconvolution performance achieves the best. If the condition number is restricted below 2, the performance will drop. So, this indicates the condition number of the signature matrix is really important to CSx. Since the condition number below 5 can show the best performance. We updated all the results using the same “Kappa” settings in Figure R13 accordingly.

Figure R13. Figure 2c & d in the manuscript

After we use CSx with a proper “Kappa” value, it can achieve satisfactory performance in many situations.

We want to thank the reviewer again for this comment! An even more thorough hyperparameter search would indeed further improve the performance of a few compared methods. The reason why changing the parameters seems to have little influence on many other algorithms or lead to worse performance, according to the above additional experiments, is that the hyperparameters are already fine-tuned by their original developers.

Moreover, it is natural that the quality of the dataset would affect the algorithms' performance. Usually, a better dataset will lead to better performance. But since the hyperparameters in dataset pre-processing should be consistent across different algorithms, which would have an impact across different algorithms in the same direction, we have already normalized them and made them consistent across different methods in all of our experiments, ruling out those hyperparameter's effects. Thus, in the above experiments, we only focused on the hyperparameters of each algorithm along, not the dataset choice and pre-processing hyperparameters.

All of these results have already been incorporated into the manuscript. Following your question, we have also put the hyperparameter searching results into the supplemental materials. Your question has indeed made our experiments more thorough and solid.

(3) Following the above question, could it be possible that TAPE works better in a certain dataset but not the others. Since there are so many published datasets with paired single cell profiles as references, the authors should compare TAPE on a much larger data cohort, where 5 datasets is way below the expectation to prove the advantages of TAPE in cell type deconvolution with statistical significance.

Answer: Thanks for the comment! We fully understand your concern because we know that a well-performed machine learning algorithm may perform poorly on another dataset if the assumption of the method is not satisfied. This problem is critical but is hard to be solved perfectly as every method has its own assumption and suitable application scenarios. Following your comments, we have performed much more thorough experiments on more datasets in the revision. Also, we performed additional experiments to investigate the suitable application scenarios of our methods, guiding the users when they can use our method confidently.

The ideal scenario is that we can find enough real bulk RNA-seq data with its cell type fractions measured by flow cytometry or immunohistochemistry method. But the fact is that we cannot find so many datasets with ground truth, that's why we additionally do experiments on pseudo-bulk data and real bulk data with clinical information (thus we can test performance with prior clinical knowledge without cell fractions). These experimental settings are inspired by previous work and these datasets are widely used in previous work (Erdmann-Pham et al. *Genome Research*, 2021, Menden et al. *Science Advances*, 2020, Jew et al. *Nature Communications*, 2020). Specifically, for the pseudo-test experiments, the settings are similar to the RNAsieve (Erdmann-Pham et al. *Genome Research*, 2021), the source datasets are also *Tabular Muris*. We also add some additional experiments following the reviewers' comments in this revision. Three deconvolution scenarios: "normal", "rare", and "similar" are defined to fully exploit the advantages of pseudo-bulk data. For the real bulk test, all the five datasets are the same as the Scaden (Menden et al. *Science Advances*, 2020). For the clinical data deconvolution, we test the ROSMAP datasets following the original paper of Bisque (Jew et al. *Nature Communications*, 2020). Additionally, we investigate the MLR tendency in the COVID-19 PBMC dataset and the decrease of beta cell proportion after COVID-19 infection in the cultured islet dataset.

Another problem is that, although we can define cell type and calculate cell type fractions in single-cell data, these cell type fractions are highly biased and can not be used as ground truth to evaluate cell type fractions in bulk RNA-seq data (see the original paper of CIBERSORTx, Fig. 2f, Newman et al. *Nature Biotechnology*, 2019). Considering this problem, in this revision, we expand experiments to pseudo-bulk data and evaluate the performance for each cell type (Figure R14, Table R9), thus we can test the performance in detail with more data points to generate some statistical meaningful results.

Figure R14. Figure 2c & d in the manuscript, we show the results by box plots, each box contains all the cell types' performance in one dataset.

Table R9. Detailed performance on each cell type, source data of Figure R14 d "monaco"

Dataset	CellType	Method	CCC	L1error
monaco	Bcells	TAPE	0.774372	0.016203
monaco	CD4Tcells	TAPE	0.484331	0.035803
monaco	CD8Tcells	TAPE	0.45154	0.037436
monaco	Monocytes	TAPE	0.320453	0.087987
monaco	NK	TAPE	0.290329	0.035385
monaco	Bcells	Scaden	0.90498	0.011847
monaco	CD4Tcells	Scaden	0.242023	0.071814

monaco	CD8Tcells	Scaden	0.455439	0.044727
monaco	Monocytes	Scaden	0.19598	0.167292
monaco	NK	Scaden	0.410949	0.03824
monaco	Bcells	RNASieve	0.108525	0.052283
monaco	CD4Tcells	RNASieve	NA	0.281521
monaco	CD8Tcells	RNASieve	0.047161	0.161824
monaco	Monocytes	RNASieve	0.336347	0.090341
monaco	NK	RNASieve	0.004297	0.388941
monaco	Bcells	MuSiC	NA	0.106905
monaco	CD4Tcells	MuSiC	NA	0.281521
monaco	CD8Tcells	MuSiC	NA	0.213033
monaco	Monocytes	MuSiC	0.535987	0.06806
monaco	NK	MuSiC	0.003785	0.665789
monaco	Bcells	DWLS	0.16424	0.063149
monaco	CD4Tcells	DWLS	0.214363	0.144211
monaco	CD8Tcells	DWLS	0.445641	0.07766
monaco	Monocytes	DWLS	0.185611	0.200044
monaco	NK	DWLS	0.347201	0.049529
monaco	Bcells	CIBERSORTx	0.120926	0.068475
monaco	CD4Tcells	CIBERSORTx	0.550437	0.051943
monaco	CD8Tcells	CIBERSORTx	0.407114	0.075094
monaco	Monocytes	CIBERSORTx	0.675171	0.065061
monaco	NK	CIBERSORTx	0.244958	0.060843
monaco	Bcells	Bisque	0.169557	0.094422
monaco	CD4Tcells	Bisque	0.059738	0.303526
monaco	CD8Tcells	Bisque	-0.072055	0.218607
monaco	Monocytes	Bisque	0.349341	0.146178
monaco	NK	Bisque	-0.093793	0.225798

Through our experiments, we find the setting of the variance cutoff fraction of the input data matters in TAPE's deconvolution performance. We made the assumption that TAPE works well on datasets that have a proper number of highly variable genes, which has also been reported by (Menden et al. *Science Advances*, 2020). Usually, this number is around 10,000. To verify this, we further investigate the changes in both the overall CCC value and MAE value among different variance cutoff settings on "Lung" (pseudo-bulk dataset, using UMI-based single-cell reference to predict pseudo-bulk constructed from count-based single-cell data) and add the related experiments.

Table R10. TAPE's performance is affected by variance cut off

fractions of genes left after variance cutoff	0.99	0.80	0.60	0.50	0.40	0.20	0.05
used genes number	16599	16529	13200	10718	8076	3489	739
overall CCC	0.28	0.28	0.42	0.43	0.59	0.56	0.33

overall MAE	0.12	0.12	0.10	0.10	0.07	0.07	0.09
-------------	-------------	-------------	-------------	-------------	-------------	-------------

From the table, we can tell that TAPE performs better with proper variance cutoff. We also draw the data distribution of different variance cutoffs in Figure R15 a. When the distribution does not have too many near-zero noises, the performance gets better. When the distribution contains many low variance genes, which means the noise is very high, the performance of TAPE is not so good. For your information, we also draw the data distribution of the real datasets we considered in Figure R13 d, which may help to explain why TAPE is not the top method on part of the datasets. From Table R9 and Figure R13, we can see that TAPE is robust enough to deal with two different distributions after filtering low variance genes but can not deal with distributions with too many near-zero noises (when we keep too many genes with low variance), which confirms our assumption.

Figure R15. Data distribution. a. Data distribution of the Lung dataset with different variance cutoff. b. Data distribution of the real datasets.

To guide the user to use our method better, following your comments, we also added this assumption and stressed the variance cutoff procedure in our methods part.

(section 4.2.4) “Then we need to filter some genes with low variance both in the training data and test data [10, 14]. This step is very important because TAPE will fail in predicting test bulk data proportions properly without proper filtering (Supplementary Table 1 and Supplementary Figure 7) [14]. In our experiments, we control the filtering threshold to keep about 10,000 genes as reported by Scaden [14]. If the less variable genes are not filtered out, TAPE can not predict a good result because of the noises (Supplementary Table 1 and Supplementary Figure 7).”

(4) I feel it is a bit over claim of TAPE’s advantage to delineate the biological significance of the clinical data, by simply showing the tendency of certain cell types. Specifically, the authors claimed that “only TAPE could correctly predict proportions of neurons or microglia cells ranging from 0.32-0.55 and 0.06-0.12 respectively” (Fig. 3a), there are multiple cell types in these studies, and how about the prediction (of cell proportions) of other cell types?

Are they all correctly predicted? Are there any experimental evidences as gold standard to support these predictions? Similarly, the prediction of MLR tendency (Fig. 3b), where the authors show only the “monocytes fraction”, I wonder how about the other cell types, such as CD4+ T cell, CD8+ T cell, or B cell? Does TAPE also correctly predict the proportion and changes of these cell types in the mild, moderate and serious (should be severe) COVID-19 patient? Fig. 3c has the same problem. The logic here is, if TAPE, but no other tools (based on the assumption that these tools were corrected used), is capable of correctly predict the proportion and changes of cell subtypes, it should work for most (if not all) cell subtypes. By showing only one or two, and in this case not even a consistent cell type, in Fig. 3 is misleading, since without detailed analysis with statistical power, these results can simply be “cherry picking”.

Answer: Thank you for raising the question about the biological significant deconvolution of TAPE. We understand your concerns because, in our practice, each method’s performance will vary across cell types, and the results may be biased. Your concerns can be divided into three parts, and we explain them below.

a) For Alzheimer’s Disease (AD), we did these experiments following the previous deconvolution methods Bisque (Jew et al. *Nature Communications*, 2020). In the original manuscript, we explained the changing tendency of neurons and microglia cells with references because neurons and microglia cells are of particular focus to AD. We also tried to find other cell types’ changes during the development of AD, but there are not enough previous studies. Therefore, we didn’t draw conclusions about other cell types.

Following your comments, in Figure R16, we show the proportions of all the cell types in the ROSMAP dataset rather than just including neurons and microglia cells. Before explaining the results, we need to clarify two things:

- The gold standard of these AD patients’ cell type proportions was cited from a previous study that uses the immunohistochemistry method to measure the cell type proportions. Actually, these fractions are the same as the ground truth in the real bulk test, but the difference is that the ground truth only measures 41 patients included in 6 braak stage, but here we study about 532 patients, so we simplified the question and accept an assumption that the cell type proportion range of the 41 measured patients is the same as all the patients. We have added this assumption to section 2.5 in the manuscript:

“Impressively, if we accept the assumption that the cell type proportions’ ranges of the 41 patients are the same as those of the 532 patients, only TAPE could predict proportions in this range, which shows the remarkable accuracy of TAPE’s prediction.”

- The 0 and 6 braak stages of the ground truth only have one sample, so we use braak stages 1-5 as references to demonstrate whether the predicted results of TAPE have both the correct range and trend.

For astrocytes cells, both TAPE and Scaden can predict the stable trend with changes in the braak stage. Oligodendrocytes and endothelial cells show a slightly increasing trend which is reasonable since the neuronal loss of AD, and TAPE can predict the trend and range similar to the ground truth.

Figure R16. Each cell type's fraction from immunohistochemistry (IHC) analysis (left column) and predicted fraction from each method. IHC data only measures 41 samples from a total of 532 samples.

b) Sorry for the unclear statement in the text. Actually, the monocytes-to-lymphocytes ratio (MLR) is calculated by the fraction of monocytes to the summation of fractions of CD8 T cell, CD4 T cell, and B cell. So, the MLR index has included these cell types in the PBMC dataset.

Following your comments, to make things clear, we have added this detail into the text (section 2.5):

“The MLR is calculated by the fraction of monocytes divided by the sum of fractions of CD4 T cell, CD8 T cell, and B cell.”

Moreover, for your information, to show our method has statistical power, we re-plot the bar plot with error bars (Figure R17) and test the increasing tendency with more data points from the convalescence and rehabilitation stage, not only from the treatment stage. We found that the data points from the convalescence and rehabilitation stage also show similar characteristics to data points from the treatment stage thus we delete the sentence from our manuscript.

~~“More specifically, we only considered the treatment stage data because patients in the convalescence or rehabilitation stage do not represent the same pathology characteristics as the real infected circumstances.”~~

In these results (Figure R17), we can see that Scaden, CIBERSORTx, DWLS, and TAPE show an increasing tendency, so we test whether the MLR of serious patients is higher than the MLR of moderate patients or mild patients. The results show that we can only find the statistical significance from TAPE that the MLR of serious patients is higher than the MLR of mild patients. Other methods don't have this statistical significance. Of note, MLR of moderate patients is not significantly higher than MLR of mild patients and MLR of serious patients is not significantly higher than MLR of moderate patients. To clearly show the statistical significance value, we add a table below:

Table R11. Statistical significance of Figure 3c

Methods	mild vs. moderate	moderate vs. serious	mild vs. serious
TAPE	0.1262	0.2578	0.0238
Scaden	0.1349	0.4291	0.0522
DWLS	0.0938	0.4214	0.081
CIBERSORTx	0.2008	0.3862	0.2512

Figure R17. Figure 3 c in the manuscript

Following your comments and the experiments, we have revised the manuscript accordingly.

(section 2.5) “Although Scaden, CIBERSORTx, DWLS, and TAPE predict an increasing tendency correctly, after hypothesis tests, only TAPE predicts the increasing tendency of MLR value with statistical significance, and the value range is suitable for the clinical report (0.29-0.88) [15].”

c) Similarly, the correlation between the decrease of beta cells' proportion and COVID infection is reported before (Müller et al. *Nature Metabolism*, 2021). As for other cell types in islet, we also want to find some evidence but there are insufficient experimental results. So, we only used this report as an example to show the sensitivity of TAPE. To address your concerns about the results, we did a one-sided t-test to test whether the virus-affected beta-cell proportion is lower than the normal and medicine-treated ones. Both p-values are lower than 0.05 which shows that our prediction has some statistical power (Figure R18). We updated the t-test results in the figure and revised the corresponding description in the text:

(section 2.5) “Though Scaden and TAPE can predict both beta cell loss and restoration in this experiment among the three conditions, after one-sided t-test, only TAPE’s predictions show a statistical significance. The accurate deconvolution results of these controlled experiments demonstrate that TAPE is sensitive to the biological changes in the bulk RNA-seq data and can produce biologically significant results, which are consistent with the previous research and reports. All the clinical deconvolution results show that TAPE’s prediction is stable, with potential clinical applications for disease early screening and treatment outcome prediction.”

Figure R18. Figure 3 d in the manuscript

To sum up, we used three examples to show TAPE’s application power in clinical data and made relevant analysis according to the previous experimental evidence, rather than only showing the positive cell type proportion results for TAPE. Following the reviewer’s comments, we have made the analysis more comprehensive, covering more cell types, and shown the stability and potential usage of the proposed method with statistical power. But of course, during real-life usage, most of the analysis will be focused on the biomarker gene and cell types. We recommend that users select the cell types they want to analyze further from the TAPE output based on existing experimental evidence.

Following the reviewer’s insightful comments, which lead us to make the above comprehensive analysis and discussion, we have added the following discussion into the revision.

(section 3) “In the scenario of clinical data prediction, TAPE is capable of predicting the ratio change for most cell types in clinical cases stably with statistical power, whose results are consistent with the previous related clinical studies [15, 25, 28, 29, 31, 32]. During real-life

usage, to make the study more focused, we recommend that users select the cell types they want to analyze further from the TAPE output based on the existing experimental evidence.”

(5) The capability of TAPE to “predict cell-type-specific gene expression at high-resolution” is what I believe a unique advantage of TAPE over the other methods, however results showing in Fig. 4 greatly limited my overall enthusiasm. The authors predicted gene RAB11FIP5 as a DEG in NK cells (Fig. 4e), again, this is only one gene. What about the other known DEGs? What percentage and to what precision can TAPE predict the expressions of the known DEGs? Predict one gene within expectation can be shown as an example, but is way less enough to prove that TAPE is able to “predict cell-type-specific gene expression at high-resolution”. The authors should set up a gold standard where ground truth is known, and then evaluate the predictive power of TAPE on whatever indexes the authors interested to measure.

Answer: Really thanks for this insightful comment with detailed suggestions! Yes, we totally agree with you. And we also think that this is the most interesting part of TAPE and should be investigated more thoroughly.

Before introducing the newly added experiment, we want to clarify that RAB11FIP5 is the only DEG between the two conditions. In the original paper (see Figure 1, S1 from Bradley et al. *Cell*, 2018), the researchers found about 300 DEGs ($p_{\text{adj}} < 0.01$) in bulk data, but those DEGs were filtered out because they were identified as non-related genes for BNab development. Those genes are probably differentially expressed because of age, sex, country, auto-antibody status, and virus load. We understand your concern that if we only study one DEG, the result may be “cherry-picking”, so we tried to reproduce the DEG analysis procedure of the non-filter data using DESeq-2 (following the original study) and found about 270 DEGs in the original bulk data without filtering. Then, to verify whether these genes are DEGs detected by our method, we test the intersection of the 270 DEGs and DEGs detected by TAPE ($p_{\text{adj}} < 0.05$) for each cell type. The results show that the intersection is none for CD4 T cell, CD8 T cell, B cell, and Monocyte. Only two genes are shared by the DEGs of NK cells and bulk samples. One is NOP2, and the other is RAB11FIP5 (Figure R19, NK from TAPE). So, this means that though there are many pseudo-DEGs in bulk samples, our method can automatically filter them out and reduce the possible DEGs number from 270 to 2. We think that this reduction of DEGs number and the accurate DEGs’ source detection (distinguish DEG from which cell type) will accelerate the biological discovery.

Following your comments, we have updated the manuscript with this detailed description to clarify the experiment in the revision.

(section 2.7) “Considering that there are about 270 pseudo-DEGs in bulk samples, we further validated whether TAPE can distinguish them as pseudo-DEGs by checking the DEGs in each cell type. The results show that TAPE only predicts NOP2 and RAB11FIP5 as DEGs in NK cell and no DEGs for other cell types (Supplementary Figure 5). We can see that the prediction is not perfect, but our method can correctly predict that NK cells have DEGs rather than other cell types and reduce the number of possible DEGs (including pseudo-DEGs if there is not any filter) from 270 to 2. All the results displayed prove that our

methods can be applied to the real-life scenario and accelerate biological discoveries by identifying which cell type has DEGs and reducing the number of possible DEGs.”

Figure R19. Volcano plots of DEGs were calculated from bulk GEPs and inferred GEPs. The orange stars refer to the RAB11FIP5 gene. The upper right plot of HIV is produced from the DESeq2 package (Love et al. Genome Biology, 2014) with original RNA-seq counts as input (pseudo-DEGs are not filtered out). Other DEGs of inferred GEPs are calculated by a two-sided t-test.

In addition to that, following your awesome suggestions, in this revision, we added a new series of simulated data with foldchange gradients and cell proportion gradients to test the CIBERSORTx and our method’s ability to detect DEGs in bulk samples. We chose 100 cells among CD8 T cell, Natural Killer, B cell, and Monocyte from PBMC single-cell data and select 10 cells from the human brain single-cell dataset as noise. After that, we chose 100 up-regulated genes randomly from 10,000 genes in CD8 T cells to modify their expression. There are 50 samples in each pseudo-bulk dataset, half of which are composed of CD8 T cells that have been up-regulated. Then, we used two-sided t-test to detect DEGs ($p < 0.05$) after obtaining the GEP of the CD8 T cell. The results show that TAPE can successfully predict cell-type-specific DEGs correctly (with good sensitivity) and selectively (with good specificity) when DEGs in one cell type if both the proportion of this cell type and foldchange of DEGs are high, while CIBERSORTx almost fails (see Figure R20 e&f). For instance, the AUROC of the bulk reference at foldchange=4 and cell proportion of CD8 T cells=0.2 is 0.8134 and TAPE’s AUROC value of this setting is 0.8274, while CSx only have an AUROC of 0.503. CIBERSORTx shows its great ability in DEGs prediction when signature genes have bigger statistical power and are easily detectable if they are differentially expressed. However, when randomly selecting 100 genes that may contain insignificant genes, it is hard

for CIBERSORTx to infer the 100 DEGs properly. This shows TAPE's broader application potential than CIBERSORTx for its ability in predicting DEGs even though they are not significant genes.

Following your great comment, we have added the additional experiments and the following paragraph to section 2.7 in the manuscript to discuss the property of TAPE more comprehensively.

“Since TAPE has shown its ability to predict cell-type-specific GEPs correctly and selectively given a group of bulk samples, we continued to use TAPE to predict cell-type-specific GEP per sample at high-resolution. To test TAPE's capability under “high-resolution” mode, we synthesized a series of pseudo-bulk samples with known differentially expressed genes (DEGs) (Figure 4e). Following the settings in CIBERSORTx [9], we selected 100 cells across four cell types (CD8 T cell, Natural Killer (NK) cell, B cell, and Monocyte) from PBMC single-cell data and another 10 cells from human brain single-cell dataset as noise to compose the pseudo-bulk data. Then we randomly selected 100 genes among 10,000 genes in CD8 T cells as up-regulated genes to adjust their expression. Each pseudo-bulk dataset contains 50 pseudo-bulk samples and half of them are composed of up-regulated CD8 T cells. The cell proportion of CD8 T cells in pseudo-bulk data ranges from 5% to 30%, and the foldchange of up-regulated genes ranges from 1.5 to 5. In total, we created a series of pseudo-bulk datasets with foldchange gradients and cell proportion gradients. After obtaining the GEP of CD8 T cells, we used a two-sided t-test to detect DEGs ($p_{\text{adj}} < 0.05$). So, this task is essentially a binary classification task, and we naturally chose area under receiver operating characteristic curve (AUROC) as the criterion. The results show that (Figure 4f), TAPE can successfully predict cell-type-specific DEGs correctly (with good sensitivity) and selectively (with good specificity) while CIBERSORTx almost fails on this task. The overall trend is that algorithms can easily recognize DEGs in one cell type if the proportion of this cell type or the foldchange of DEGs is high. Interestingly, using DEGs in bulk as the reference, we can see that TAPE can even predict DEGs not shown up in bulk samples but in CD8 T cells (the maximum AUROC of TAPE is higher than the maximum AUROC of bulk samples in Figure 4e,f). In the original article [9], CIBERSORTx has also demonstrated its great ability in DEGs prediction, the reason why it failed in this task is that CIBERSORTx usually focuses on signature genes which have bigger statistical power and are easily detectable if they are differentially expressed, but in this task, we randomly selected 100 genes which are probably insignificant genes; therefore, it is hard for CIBERSORTx to infer the 100 DEGs properly. In contrast, TAPE has shown its ability in predicting DEGs even when they are not significant genes, which means TAPE has a broader application potential than CIBERSORTx.”

Figure R20. Figure 4 in the manuscript

(6) As we know, major cell types, such as T cells, B cells and etc. can be further divided into cell subtypes, I am curious to what extent can TAPE reach to, i.e. what is TAPE's limitation to predict the gene expression in smaller cell subtype, which can be referred to "high-resolution".

Answer: Thank you for asking this excellent question! This question is really interesting, and we must do the related experiments since no one else has done it before.

Before examining the ability to distinguish similar cell types' GEPs, we first want to show the ability to distinguish the similar cell type's fractions. Previously, we have done some preliminary experiments to test TAPE's performance on the subtype deconvolution. For example, in PBMC data, we evaluated the subtypes of T cells (CD4 T cells and CD8 T cells) separately. The CCC and MAE results (Figure R21) show that TAPE is the best algorithm on these datasets.

Figure R21. show the ability to predict cell type fractions of similar cell types in real datasets.

In this revision, we performed more thorough experiments for this scenario. We carry out further analysis of the highly-related cell types in the “similar” scenario of pseudo bulk data. In the “similar” scenario, we use “Marrow” to compare the deconvolution performance with other methods. This dataset has two types of B cell in it: “late-pro B cell” and “immature B cell”, which satisfies the setting of highly related cell types. As expected, the calculated CCC and MAE values show that TAPE is the best algorithm. In detail, TAPE can not only distinguish each cell type when both kinds of the B cell are in the reference but also transfer one B cell's proportion to another if this B cell is missing from the reference (Figure R22). And we added the corresponding statement accordingly:

(section 2.2) “As for the “similar” scenario, we investigate the performance of two kinds of B cell in the “normal” scenario and what would happen if we delete one cell type from the reference. The results show that TAPE is the best algorithm and can distinguish cell subtypes when both kinds of the B cell are in the reference (Figure 2c, “similar distinguishment”). Moreover, TAPE can transfer one B cell's proportion to another if this kind of B cell is missing from the reference (Figure 2c, “similar transferring”).”

Figure R22. Figure 2c in the manuscript, “similar distinguishment” and “similar transferring” show the ability to predict cell type fractions of similar cell types.

Here, we also use simulated data to test whether TAPE can distinguish similar cell types’ GEPs. The simulated process is similar to the process we mentioned before in question 5. But here, the four cell types are CD8 T cells, CD4 T cells, Monocytes, and NK cells. Similar cell types are CD4 T cell and CD8 T cell. We also randomly select 100 genes from CD8 T cells and up-regulate them with different foldchanges. Then we simulated bulk samples with different CD8 T cell’s fractions to compose a gradient. So, we totally create 30 tests and each test has 50 bulk samples of which half are up-regulated and half are controls. We also use CIBERSORTx and our method to test their ability to detect DEGs. Since this can be seen as a classification task, we use the index AUROC as we mentioned before.

The results show that both CIBERSORTx and TAPE can not distinguish the cell subtypes’ GEPs well. But TAPE still has better predictive power than CIBERSORTx. Also, we find that TAPE can distinguish some DEGs from CD8 T cells as well but CIBERSORTx always detects DEGs from CD4 T cells rather than CD8 T cells (Figure R23). Specifically, for CIBERSORTx and TAPE, the highest AUROC value in CD4 T cells are 0.75 and 0.86 respectively while the highest AUROC value in CD8 T cells are 0.53 and 0.84 respectively. So, we conclude that TAPE has better predictive power than CIBERSORTx and both methods will wrongly detect DEGs from similar cell types. Despite these methods having this limitation, we also think these methods are useful because they can successfully distinguish DEGs from T cells and exclude unrelated cell types like monocytes and NK cells. This can reduce the potential candidates and accelerate biological research.

Figure R23. DEG detection ability of CIBERSORTx and TAPE when there are similar cell types in bulk samples. The red color means the DEGs are detected well and the blue color means the DEGs are not well detected. In this scenario, both CIBERSORTx and TAPE will detect DEGs from CD4 T cell rather than CD8 T cell.

We really appreciate your constructive comments because we can not clearly find our methods' limitations without your suggestions. So, in the manuscript, we also add this experiment in section 2.7 and discuss this issue as our main limitation in the discussion section:

(section 2.7) "In addition to the positive results that TAPE has really good performance in predicting cell-type-specific GEPs, we also found that TAPE has its limitation in predicting cell subtypes' GEPs. In detail, we set up a series of simulated bulk data to detect DEGs as we mentioned before. But we used similar cell types in this test. Specifically, we used similar cell types like CD4 T cells and CD8 T cells together with two other cell types, namely monocytes and NK cells. We also benchmarked CIBERSORTx and TAPE to see whether they can distinguish DEGs from CD8 T cells and DEGs from CD4 T cells. The results show that (Supplementary Figure 6) both methods can not correctly distinguish DEGs but TAPE still shows a better performance than CIBERSORTx because TAPE can distinguish some DEGs from CD8 T cells correctly, but CIBERSORTx always detects DEGs as from CD4 T cells rather than from CD8 T cells. Furthermore, the AUROC value of TAPE is still higher than CIBERSORTx. These results clearly show the shortage of both methods, which will be examined in more detail in the Discussion part."

(section 3) "Secondly, we notice that both CIBERSORTx and our method can not distinguish DEGs from cell subtypes correctly (Supplementary Figure 6) which means that their resolution is still limited. Though our method cannot precisely predict DEGs from cell subtypes, it still reduces the potential candidates by excluding unrelated cell types. So, our method is still useful and can be applied in the real-life scenarios to accelerate biological research."

(7) Fig. 5 is yet another very rough and premature result. Without solid evidence of the predictive power of TAPE, any TAPE prediction with biological implications can be wrong. Even if predictions from TAPE is right, what are the enriched pathways shown in Fig. 5a? What does this mean biologically (related to virus infection) if these pathways are enriched? How could people know whether these predictions were right? Are there any positive or negative controls? Without these detailed information, how can I be convinced?

Answer: Thank you very much for the comment! We fully understand your concerns. Indeed, in the original manuscript, our discussion about Fig. 5 was too short, which could lead to misunderstanding. In the revision, following your comments, we have discussed it in much more detail, clarifying what you were concerned about. Below are the details.

Essentially, in that part, what we show is a potential application of TAPE. That is, by combining it with ssGSEA, we can study which pathway may be enriched in different cell types rather than the whole bulk sample. In other words, with the help of TAPE, we can

study the biological pathways in a cell-specific resolution, which other methods have not achieved. For example, CIBERSORTx is usually restricted to predict a small number of significant genes in cell-type-specific GEPs.

Your concern about the positive and negative controls is very reasonable. In the bulk sample pathway analysis, we usually need to have the controls. But as we are doing the pathway analysis in the single-sample condition, that traditional analysis is unsuitable. Thus, we combined TAPE with single-sample gene set enrichment analysis (ssGSEA) (Barbie et al., *Nature*, 2009). Unlike GSEA, ssGSEA can analyze a single sample by evaluating the gene expression rank and allows one to define an enrichment score that represents the degree of absolute enrichment of a gene set in each sample within a given dataset (see the original article, methods online, ***signature projection method***, Barbie et al. *Nature*, 2009). It eliminates the necessity of controls, which is suitable for our case. Therefore, we choose ssGSEA to obtain the gene set enrichment score. Of note, in all the results, we only consider the statistically significant results, and the p-value of ssGSEA is calculated by testing the hypothesis that the Spearman correlation between the enrichment score of gene set of interest was greater than zero.

To clarify that part, following your comments, we have added the description to the manuscript.

(section 2.8) “Besides the differential expressed genes, we also investigated the functions of each cell type by incorporating cell-specific GEPs and ssGSEA [16]. The ssGSEA algorithm only needs the gene rank, which our cell-type-specific GEPs could provide, we could predict the activities of each function pathway for each sample without positive or negative controls.”

It also makes perfect sense that we should try to verify the prediction and detected enriched pathways. To clarify, usually, people can verify their results by directly checking whether the pathway function description is related to the experimental results (Grimes & Grimes. *Journal of Molecular and Cellular Cardiology*, 2020, cited by 125). The pathway description can be found on the MSigDB website (<https://www.gsea-msigdb.org/gsea/msigdb/genesets.jsp>). And the experimental results usually demonstrate that one key protein in a pathway is inhibited/activated by some molecules. If the key protein exists in the pathway and is inhibited/activated under the virus infection condition, and the gene set analysis results also show that this pathway is inhibited/activated, people can conclude that this gene set enrichment analysis is correct. The same analysis procedure can be found in this article (Grimes & Grimes. *Journal of Molecular and Cellular Cardiology*, 2020, cited by 125). The results from those papers are usually considered as ground truth. Here, what we have done is to compare our results, which were obtained using the more efficient algorithm proposed by us, with the ground truth from those works.

Regarding the pathways detected by our method as well as their biological meanings, in Figure 5, the subfigure a shows the overview of our results and we can not list or investigate all of these pathways since it is too complex to be clearly understood. Therefore, in the subfigure b, we provide an easy way to interpret the statistically significant pathways ($p_{\text{adj}} < 0.05$) by finding some special pathways in one virus-infected dataset but not in others. This is a useful analysis method when we have to face 200 pathways in 200 samples. In this way,

we could find some uniquely activated pathways, and naturally, these pathways would have interesting biological meaning by reflecting the different effects caused by different viruses.

Specifically, when we compared the enriched functional pathways in each cell type in different virus infections, we found that most of the commonly activated pathways within the three infections in the B cell are general immune response pathways, including BIOCARTA_IL2_PATHWAY, BIOCARTA_IL4_PATHWAY, BIOCARTA_IL6_PATHWAY, and BIOCARTA_IL7_PATHWAY, representing the typical immune response in the three infections. Also, we identified some specific activated functional pathways in B cell with SARS-CoV-2 infections: BIOCARTA_MAPK_PATHWAY and BIOCARTA_LONGEVITY_PATHWAY. BIOCARTA_MAPK_PATHWAY (MAPKinase Signaling Pathway) activation has proved to cause an overwhelming inflammatory response in SARS-CoV-2 infection (Grimes & Grimes. *Journal of Molecular and Cellular Cardiology*, 2020, cited by 125). The blockage of the BIOCARTA_LONGEVITY_PATHWAY (The IGF-1 Receptor and Longevity) has also been reported to mitigate lung injury and decrease the risk of death in patients with SARS-CoV-2 (Winn. *Medical Hypotheses*, 2020, cited by 20). Recent studies also found the coronavirus would induce cell cycle arrest, which did not exist in other kinds of virus infection, which was also discovered by our algorithm (Su et al. *Frontiers in Veterinary Science*, 2020, cited by 25). These specific activated pathways in SARS-CoV-2 samples were consistent with previous studies, representing the capability of interpreting cell-type-specific dysfunctional pathways by the GEPs.

Following your questions and comments, we have added this more detailed discussion and analysis into the revision.

(section 2.8) “Combining with prior knowledge, some pathways we found are highly correlated with these diseases. For instance, most of the commonly activated pathways within the three infections in the B cell are general immune response pathways, including BIOCARTA_IL2_PATHWAY, BIOCARTA_IL4_PATHWAY, BIOCARTA_IL6_PATHWAY, and BIOCARTA_IL7_PATHWAY. Interestingly, we identified some specific activated functional pathways in SARS-CoV-2 infections: BIOCARTA_MAPK_PATHWAY, BIOCARTA_LONGEVITY_PATHWAY, and BIOCARTA_CELLCYCLE_PATHWAY which have been reported before. BIOCARTA_MAPK_PATHWAY (MAPKinase Signaling Pathway) activation has proved to cause an overwhelming inflammatory response in SARS-CoV-2 infection [38]. The blockage of the BIOCARTA_LONGEVITY_PATHWAY (The IGF-1 Receptor and Longevity) has also been reported to mitigate lung injury and decrease the risk of death in patients with SARS-CoV-2 [39]. Recent studies also found the coronavirus would induce the cell cycle arrest, which did not exist in other kinds of virus infection, which was also discovered by our algorithm [40]. Generally, these examples show that the combination of TAPE and ssGSEA can indeed discover some significant pathways as clues for further experimental validation.”

In summary, we have shown that TAPE can be used to study the pathways in high resolution. Although TAPE’s prediction is not perfect, our results have been partially proved by the related publications based on experiments. Moreover, since more and more biologists start using the GSEA results as a guide to design their experiments, combining ssGSEA with our method is helpful for biologists to make some *ex-ante* analyses and can help them make biological discoveries in an efficient way.

We understand our previous discussion is indeed premature and unclear, as you have pointed out. Following your insightful comments, we have added the following discussion into the revision to clarify the potential usage of our method.

(section 3) “Benefited from the predicted cell-type-specific GEPs of the TAPE in the “high-resolution” mode, we could identify specific activated functional pathways in each cell type for each sample, which could be another potential advantage of our algorithm. According to the results above, we could identify cell types involved in the dysfunctional pathways. Combining ssGSEA and TAPE could help identify the specific dysfunctional pathways in particular cell types using the bulk RNA-seq data, which will essentially make use of previous population transcriptome datasets.”

(8) I am curious, if TAPE is applied to analyze clinical bulk RNA-seq data, does the model needs to be re-trained? In other words, if the authors trained TAPE with normal bulk/single-cell RNA-seq data, can TAPE be applied to predict the proportion of cell types or the expression of genes in disease state? Because I suspect genes are usually differentially expressed in disease samples, even in the same cell subtypes, and wonder if the model needs to be re-trained based on disease data. If “Yes”, then the application of TAPE to “enable and accelerate the precise analysis of high-throughput clinical data in a wide range” will be limited.

Answer: Thanks for asking about the re-training and speed! In fact, the training process and speed are advantages of TAPE over the other methods. In the previous version, we showed in section 2.5 that TAPE has a good performance in predicting the cell-type proportions in disease state and we also mentioned that “TAPE only needs simulated data from healthy samples to train, but it can predict the cell-type-specific gene expression in pathological conditions if the corresponding bulk RNA-seq data is given.” This task is relatively difficult and only a few methods have considered it before. But we realized that we didn’t clarify the re-training process in the two tasks enough. Following your suggestion, we explain it in detail:

(section 4.2.5) “The adaptive training stage is more like the fine-tuning step in deep learning rather than being re-trained with new single-cell data. The adaptive-training time is 3 seconds per sample with GPU acceleration.”

When predicting the proportion of cell types, TAPE does not need to be re-trained. Being trained with normal single-cell RNA-seq data, TAPE can be applied to predict the proportion of cell types of disease tissues directly and robustly. Although we do expect there to exist some genes that are differentially expressed between disease states, the proportion of differentially expressed genes is usually small in real applications. Thus, we can expect the performance of TAPE will not be affected much, and indeed TAPE shows a good performance in predicting the cellular compositions, which is often the key signal to many diseases. For example, TAPE can predict the tendency of neuron loss and have a good prediction of microglia activation and deactivation among 532 samples with clinical information, see more details in Figure R24 and response point 4.

Figure R24. Figure 3 in the manuscript

Even when it is applied to predict the gene expression profiles in a disease state, TAPE will just be further trained in an adaptive way (using real bulk data to train the model in an unsupervised manner, like the fine-tuning step in deep learning) rather than being re-trained with single-cell data from the disease state. The adaptive-training time is 3 seconds per sample. What is more, in section 2.7 and response point 5, we have proved that this adaptive stage is useful for predicting differentially expressed genes (also see Figure R25).

Figure R25. Figure 4 in the manuscript

Furthermore, even if the users want to re-train everything from scratch for their own completely different in-house applications, it is not a time-consuming task to do so. Below, we show the training time of TAPE.

Table R12. source data of Figure 2b in our manuscript. Time consumption on the test of “Lung umi2counts” task for each method, unit is second

tested on Lung umi2counts, sample size	100	200	300	400	500	600	700	800
TAPE	112.72	107.58	110.45	106.56	114.79	146.63	157.50	122.82
Scaden	340.16	352.71	358.82	353.40	347.19	351.96	393.23	342.95
RNA sieve	120.20	158.52	198.18	231.03	275.40	315.85	348.13	398.68
CIBERSORTx (web)	821.68	1664.05	2595.24	NA	NA	NA	NA	NA
DWLS	267.85	530.96	839.19	1064.65	1314.96	1581.76	1913.18	2133.69
MuSiC	14.74	17.21	21.20	26.13	28.73	34.15	39.23	44.02
Bisque	12.39	12.42	12.81	13.45	13.87	14.67	14.99	15.14

We can see that, including the process to generate 5000 pseudobulk training data, it only takes about 2 mins to train TAPE from scratch. This light method could be potentially useful to users without plenty of computational resources.

Following your comments, we have added the following description and figure about our training time in the revision.

(section 2.4) “Among all the methods tested (Figure 2b), Bisque is the fastest algorithm, and it can deconvolve 800 samples in 15 seconds. For TAPE, it takes about 120 seconds in total to construct the training data and train the deep learning model for 5,000 iterations. But its inference speed is very fast and its time complexity is $O(n)$ with a very small coefficient due to the inherent advantage of using deep learning. Thus, TAPE’s time consumption would not increase markedly with a larger cohort size. Besides the time complexity, TAPE only needs about 1900MB GPU memory during the training stage. When deconvolving new bulk samples, the memory consumption will increase along with the number of samples, but this increment is really small in practice. Comparing with Scaden, another deep learning method, TAPE is faster because of its highly optimized training data simulation procedure and a smaller model size. Of note, the deconvolution step of DWLS is not slow, but the step of constructing signature matrix using *MAST* is really time- and memory-consuming. As for CIBERSORTx, its slow prediction speed is not justified because its speed is limited by the web server. We would expect a much better performance if users can acquire the source program from the developers. Generally, within the test settings, algorithms that do not require complicated preprocessing steps (Bisque and MuSiC) achieve a better performance on speed.”

Figure R26. Figure 2b in the manuscript, Log time consumption of each method. Source data is Table R12.

(9) Fig. 4b is problematic and needs to be further clarified. The author claimed “Interestingly, comparing the relative NRG1 expression value in bulk GEP, single-cell GEPs, and predicted GEPs (Fig 4b), we found that TAPE can successfully predict a high expression value of NRG1 in neurons while a low expression value of NRG1 in endothelial cells.” I am not able to draw this conclusion by looking at this figure. Besides, why OPC, Oligodendrocytes and Unknown have either the blue or red bar, instead of both? What does “unknown” mean in this figure? Fig. 4e, very seldom people will use $-\log_{10}(P\text{-value})$ in a box-plot to show the significance, and it should not be $p=xx$ in the box-plot, otherwise $p=0.00$ is considered as “very significant”.

Answer: We appreciate the reviewer for pointing out the blurring of Figure 4 in the manuscript and other detailed comments! Following your suggestions, to make things clearer, in this revision, we have updated the subfigures of Figure 4 and changed the structure of Figure 4.

(a) In the original figure and the revised figure (see Figure R27 below), the blue columns represent the NRG1 expression value predicted by our model. In contrast, the red columns represent the NRG1 expression value from healthy human brain single-cell profiles. Here, we want to say that, for the predicted value (blue columns), compared to the high-level gene expression column in ExNeurons, InNeurons, and Astrocytes, the gene expression in Endothelial is low. In contrast, for the healthy single-cell profiles (red columns), expression values of NRG1 in these four cell types don't have such big differences. Thus, we draw the conclusion that the predicted gene expression value from our model can selectively allocate gene expression to its corresponding cell types rather than simply allocating the same gene expression value to all the cell types.

To avoid confusion from the readers, following your question, we have revised the manuscript as below.

(section 2.6) “Interestingly, for the predicted values (Fig 4g, blue columns), the gene expression value in Endothelial is low compared with the high-level gene expression values in ExNeurons, InNeurons, and Astrocytes. In contrast, for the healthy single-cell profiles (Fig 4g, red columns), expression values of NRG1 in these four cell types don't have such big

differences. Thus, TAPE can successfully predict a high expression value of NRG1 in neurons while a low expression value of NRG1 in endothelial cells.”

(b) For your information, we changed the label of the original Figure 4b to Figure 4g in the revised manuscript. The reason why “OPC, Oligodendrocytes and Unknown have either the blue or red bar, instead of both” is that when a certain bar is missing, it shows that the corresponding gene expression value is equal to zero. And “unknown” represent cell types that have no specific definition in the single-cell data. To make it clear, we remove “unknown” from the results and change the appearance of Figure 4b (Figure 4g (Figure R27) in the revised version). Also, to avoid causing misunderstanding and confusion, following your question, we have revised the manuscript as below.

(caption Figure 4g) “The relative gene expression value of NRG1 from different sources. The dashed line represents the total relative NRG1 expression value in the AD patients’ brain tissue. The missing blue or red column means the relative gene expression value of prediction or single-cell data is zero.”

Figure R27. Figure 4g in the manuscript

(c) In the original manuscript, we used $-\log_{10}(P\text{-value})$ following (Fury et al. 2006 International Conference of the IEEE Engineering in Medicine and Biology Society, 2006). Following your suggestion, we replace $-\log_{10}(P\text{-value})$ with P-value and the revised figure is below.

Figure R28. Figure 4h in the manuscript

References

- Barbie, D. A., Tamayo, P., Boehm, J. S., Kim, S. Y., Moody, S. E., Dunn, I. F., Schinzel, A. C., Sandy, P., Meylan, E., Scholl, C., Fröhling, S., Chan, E. M., Sos, M. L., Michel, K., Mermel, C., Silver, S. J., Weir, B. A., Reiling, J. H., Sheng, Q., ... Hahn, W. C. (2009). Systematic RNA interference reveals that oncogenic KRAS-driven cancers require TBK1. *Nature*, *462*(7269), 108–112. <https://doi.org/10.1038/nature08460>
- Bradley, T., Peppas, D., Pedroza-Pacheco, I., Li, D., Cain, D. W., Henao, R., Venkat, V., Hora, B., Chen, Y., Vandergrift, N. A., Overman, R. G., Edwards, R. W., Woods, C. W., Tomaras, G. D., Ferrari, G., Ginsburg, G. S., Connors, M., Cohen, M. S., Moody, M. A., ... Haynes, B. F. (2018). RAB11FIP5 expression and altered natural killer cell function are associated with induction of HIV broadly neutralizing antibody responses. *Cell*, *175*(2), 387-399.e17. <https://doi.org/10.1016/j.cell.2018.08.064>
- Erdmann-Pham, D. D., Fischer, J., Hong, J., & Song, Y. S. (2021). Likelihood-based deconvolution of bulk gene expression data using single-cell references. *Genome Research*, *31*(10), 1794–1806. <https://doi.org/10.1101/gr.272344.120>
- Fury, W., Batliwalla, F., Gregersen, P. K., & Li, W. (2006, August). Overlapping probabilities of top ranking gene lists, hypergeometric distribution, and stringency of gene selection criterion. *2006 International Conference of the IEEE Engineering in Medicine and Biology Society*. <http://dx.doi.org/10.1109/iembs.2006.260828>
- Grimes, J. M., & Grimes, K. V. (2020). p38 MAPK inhibition: A promising therapeutic approach for COVID-19. *Journal of Molecular and Cellular Cardiology*, *144*, 63–65. <https://doi.org/10.1016/j.yjmcc.2020.05.007>
- Jew, B., Alvarez, M., Rahmani, E., Miao, Z., Ko, A., Garske, K. M., Sul, J. H., Pietiläinen, K. H., Pajukanta, P., & Halperin, E. (2020). Accurate estimation of cell composition in bulk expression through robust integration of single-cell information. *Nature*

Communications, 11(1). <https://doi.org/10.1038/s41467-020-15816-6>

Love, M. I., Huber, W., & Anders, S. (2014). Moderated estimation of fold change and dispersion for RNA-seq data with DESeq2. *Genome Biology*, 15(12).

<https://doi.org/10.1186/s13059-014-0550-8>

Menden, K., Marouf, M., Oller, S., Dalmia, A., Magruder, D. S., Kloiber, K., Heutink, P., & Bonn, S. (2020). Deep learning–based cell composition analysis from tissue expression profiles. *Science Advances*, 6(30). <https://doi.org/10.1126/sciadv.aba2619>

Monaco, G., Lee, B., Xu, W., Mustafah, S., Hwang, Y. Y., Carré, C., Burdin, N., Visan, L., Ceccarelli, M., Poidinger, M., Zippelius, A., Pedro de Magalhães, J., & Larbi, A. (2019). RNA-Seq Signatures Normalized by mRNA Abundance Allow Absolute Deconvolution of Human Immune Cell Types. *Cell Reports*, 26(6), 1627-1640.e7. <https://doi.org/10.1016/j.celrep.2019.01.041>

Müller, J. A., Groß, R., Conzelmann, C., Krüger, J., Merle, U., Steinhart, J., Weil, T., Koepke, L., Bozzo, C. P., Read, C., Fois, G., Eiseler, T., Gehrmann, J., van Vuuren, J., Wessbecher, I. M., Frick, M., Costa, I. G., Breunig, M., Grüner, B., ... Kleger, A. (2021). SARS-CoV-2 infects and replicates in cells of the human endocrine and exocrine pancreas. *Nature Metabolism*, 3(2), 149–165.

<https://doi.org/10.1038/s42255-021-00347-1>

Newman, A. M., Steen, C. B., Liu, C. L., Gentles, A. J., Chaudhuri, A. A., Scherer, F., Khodadoust, M. S., Esfahani, M. S., Luca, B. A., Steiner, D., Diehn, M., & Alizadeh, A. A. (2019). Determining cell type abundance and expression from bulk tissues with digital cytometry. *Nature Biotechnology*, 37(7), 773–782.

<https://doi.org/10.1038/s41587-019-0114-2>

Su, M., Chen, Y., Qi, S., Shi, D., Feng, L., & Sun, D. (2020). A mini-review on cell cycle regulation of coronavirus infection. *Frontiers in Veterinary Science*, 7.

<https://doi.org/10.3389/fvets.2020.586826>

Tsoucas, D., Dong, R., Chen, H., Zhu, Q., Guo, G., & Yuan, G.-C. (2019). Accurate estimation of cell-type composition from gene expression data. *Nature*

Communications, 10(1). <https://doi.org/10.1038/s41467-019-10802-z>

Winn, B. J. (2020). Is there a role for insulin-like growth factor inhibition in the treatment of COVID-19-related adult respiratory distress syndrome? *Medical Hypotheses*, 144, 110167. <https://doi.org/10.1016/j.mehy.2020.110167>

REVIEWER COMMENTS

Reviewer #1 (Remarks to the Author):

Thanks to the authors for addressing some of my concerns. While I really like the general premise of the manuscript, there are some ambiguities in the experiments that make it hard to estimate if TAPE improves over SOTA methods in cell fraction or gene expression estimation. There are several main and minor points that I would deem important to address.

Main points:

- 1) The authors presented figure R1 (Figures 2c and 2d in the updated manuscript) to demonstrate performance per cell type. From this, it is difficult to say that TAPE improves over Scaden for prediction of cell type composition on real bulk datasets. Monaco is the only real bulk dataset there where the median CCC of tape seems higher than Scaden.
- 2) Similar is the case for figure R3, where it is really hard to compare TAPE and Scaden. On umi2counts, TAPE definitely seems a lot better in MAE but substantially worse in CCC, indicating poor correlation or predictions and ground truth.
- 3) Given points 1 & 2, it seems that there are many overstatements of TAPE's performance compared to SOTA methods. If the author cannot show significant improvement of existing methods, they need to tone down many of their claims throughout the manuscript.
- 4) For Figure R4, I would have appreciated going deeper into cellular subsets rather than evaluating performance on just CD4 and CD8 T cells, which are very broad T cell subtypes. To provide useful cell type deconvolution to biologists (or immunologists) it would be nice to know if TAPE can predict naive and memory subsets well, for instance.
- 5) Since TAPE doesn't show clear improvement in the estimation of cell proportions, the novelty of TAPE lies in the fact that it can estimate gene expression at scale (i.e. of all genes present in the single-cell data). However experiments presented by authors still seem cherry-picked. If TAPE can predict all genes, what is the need to select only 100 genes, especially on simulated data where TAPE has access to the ground truth?
- 6) Thank you for adding an experiment in estimation of gene expression (Figure R9). I would be interested in knowing if the correlation of relative expression at gene level is also good.
- 7) By design CibersortX only imputes genes that are likely to be significant in at least one cell type. It is therefore improper to compare TAPE and CibersortX on genes that are insignificant as authors did for simulated dataset. Could you please compare using significant genes. In addition, the evaluation of CibersortX on AD brains is missing.
- 8) To evaluate gene expression estimation, it would be necessary to know how cell type-specific gene expression from highly resolved samples cluster? Are similar cell types closer to each other?
- 9) Thank you for the clear explanation of the reasoning behind usage of Gaussian noise. The authors added "Here, we add noise to the simulated data because we want to make this pseudo-bulk test more difficult and closer to the real cases, instead of toy simulations." As far as I know adding Gaussian noise may not necessarily mimic real bulk data better. Further explanation/experiment is necessary to show that Gaussian noise indeed makes simulations closer to real bulks.

Minor points:

- 10) The authors responded with "In this paper, we add Gaussian noise and dropouts." However, it is unclear how adding dropouts to single-cell data makes simulations better or closer to the real bulks. If

anything, real bulk contains more non-zero genes.

11) Without looking at the source code, it is difficult to follow the training procedure of TAPE. It would be nice if the authors explain it more clearly.

12) Since the output of the encoder does not sum to 1 and can be negative, the authors apply ReLU activation and normalize the result to sum to 1 using a scaling function.

However, from the code in `train.py` and `model.py` in <https://github.com/poseidonchan/TAPE/tree/main/TAPE>, it seems that ReLU and scaling functions are only used during prediction (and adaptive stage). Am I wrong? Are ReLU and scaling functions always included in the model (i.e. in forward propagations)? If not, does it not violate entirely the assumption that the signature matrix is visible in the decoder since the proportions have to sum to 1 for $XS=B$ to be valid? Or is the decoder meant to represent the signature matrix only at the adaptive stage? This should be clearly stated in the manuscript.

Best of luck and kind regards,
Stefan Bonn

Reviewer #2 (Remarks to the Author):

All my concerns have been fully addressed in an awesome way.

Reviewer #1. (Remarks to the Author):

Thanks to the authors for addressing some of my concerns. While I really like the general premise of the manuscript, there are experiments that make it hard to estimate if TAPE improves over SOTA methods in cell fraction or gene expression estimation. There are several main and minor points that I would deem important to address.

It's nice to hear your supportive comments! We truly appreciate your thorough and constructive review of the paper and for taking the time to go through many of the details! In this revision, following your comments, we mainly made the following three revisions to further improve the clarity of our manuscript.:

1. We evaluate different methods more comprehensively and elaborate on their advantages and shortcomings under various scenarios. We summarize all the results in Table R1 and R3 in this reply letter. For your convenience, we also put them below. Understandably, no method, including TAPE, could outperform all the other methods constantly across all the datasets. For example, Scaden shows good CCC performance on microarray dataset. Meanwhile, TAPE has satisfactory performance on monaco's and ROSMAP_human datasets (accurate and robust with a small interquartile range). To make the comparison clear, we vote for the method that performs the best in most scenarios. To sum up, TAPE has the most frequent occurrences as the top3 methods under all the scenarios (as shown in Table R1 and R2). Furthermore, it is the best method for 16 out of 20 evaluation scenarios on the real-bulk datasets.
2. We design a set of comprehensive experiments to test the DEG detection ability of TAPE and CIBERSORTx. We show that each method has its own advantages and shortages in different scenarios, as shown in below Table R3. To sum up, TAPE performs better than CIBERSORTx when DEGs are randomly selected, which is the real scenario.
3. In the writing part, for readability, we use a pseudo-code table to explain how we train the model in the adaptive stage. For other misleading/unclear claims, we rewrite and explain them more comprehensively.

Please find below for the point-to-point response to your comments and concerns.

Table R1. Performance summary of TAPE and SOTA methods. Here, we list the **top3** methods in order for different scenarios and datasets. The performance comparison between box plots is evaluated by two-sided t-test. The order is based on the p-value, a small p-value represents better performance. In some scenarios with only two data points, we only compute the average value for comparison.

Datatype	Scenario	Dataset	Metrics				
			CCC		MAE		
			Overall	P-value (used for comparing the box plot)	Overall	P-value (used for comparing the box plot)	
Real-bulk		sdyl67	TAPE, Scaden, CSx	DWLS, Scaden, TAPE	TAPE, Scaden, MuSiC	TAPE, Scaden, CSx	
		monaco	TAPE, CSx, Scaden	TAPE, CSx, Scaden	TAPE, CSx, Scaden	TAPE, CSx, Scaden	
		microarray	Scaden, TAPE, DWLS	Scaden, CSx, TAPE	TAPE, Scaden, CSx	TAPE, Scaden, CSx	
		rosmap_h	TAPE, Scaden, RNAsieve	TAPE, Scaden, MuSiC	TAPE, Scaden, MuSiC	TAPE, Scaden, MuSiC	
		rosmap_m	TAPE, CSx, DWLS	RNAsieve, DWLS, TAPE	TAPE, DWLS, Scaden	TAPE, DWLS, Scaden	
Pseudo-bulk	umi2counts	normal	Limb_Muscle	DWLS, CIBERSORTx, Scaden	DWLS, CIBERSORTx, Scaden	DWLS, Scaden, CIBERSORTx	Scaden, DWLS, CIBERSORTx
			Lung	DWLS, Bisque, TAPE	DWLS, Bisque, TAPE	DWLS, Scaden, TAPE	DWLS, Scaden, TAPE
			Marrow	DWLS, TAPE, Scaden	DWLS, TAPE, CIBERSORTx	TAPE, DWLS, Scaden	TAPE, DWLS, CIBERSORTx
		rare*	Limb_Muscle	not applicable	Scaden, DWLS, TAPE	not applicable	TAPE, DWLS, Scaden
			Lung	not applicable	MuSiC, CIBERSORTx, Scaden	not applicable	MuSiC, DWLS, RNAsieve
			Marrow	not applicable	DWLS, Scaden, TAPE	not applicable	DWLS, TAPE, MuSiC
	similar distinguishment*	Marrow	not applicable	TAPE, DWLS, CIBERSORTx	not applicable	TAPE, CIBERSORTx, Scaden	
	similar transferring*	Marrow	not applicable	DWLS, TAPE, Scaden	not applicable	DWLS, TAPE, Scaden	
	counts2umi	normal	Limb_Muscle	TAPE, DWLS, Scaden	DWLS, TAPE, MuSiC	TAPE, Scaden, DWLS	TAPE, DWLS, Scaden
			Lung	MuSiC, TAPE, Scaden	MuSiC, DWLS, Bisque	MuSiC, TAPE, Scaden	MuSiC, TAPE, Scaden
			Marrow	MuSiC, TAPE, Scaden	MuSiC, TAPE, Scaden	TAPE, Scaden, MuSiC	TAPE, MuSiC, Scaden
		rare*	Limb_Muscle	not applicable	MuSiC, DWLS, TAPE	not applicable	MuSiC, TAPE, Scaden
			Lung	not applicable	DWLS, MuSiC, CIBERSORTx	not applicable	CIBERSORTx, DWLS, MuSiC
			Marrow	not applicable	MuSiC, CIBERSORTx, Scaden	not applicable	Scaden, DWLS, MuSiC
similar distinguishment*		Marrow	not applicable	MuSiC, TAPE, CIBERSORTx	not applicable	TAPE, MuSiC, CIBERSORTx	
similar transferring*		Marrow	not applicable	MuSiC, TAPE, Scaden	not applicable	MuSiC, TAPE, Scaden	

*only compute the average performance of all available data points because of the small number of data points

Table R2. Statistical table of the number of occurrences in Table R1.

		Occurrences	Top1 Occurrences	Top2 Occurrences	Top3 Occurrences
All	TAPE	53	27	16	10
	Scaden	48	5	16	27
	DWLS	35	16	16	3
	MuSiC	26	14	3	9
	CIBERSORTx	24	1	11	12
	RNAsieve	3	1	0	2
	Bisque	3	0	2	1
Real	TAPE	20	16	1	3
	Scaden	18	2	10	6
	CIBERSORTx	10	0	6	4
	DWLS	6	1	3	2
	MuSiC	4	0	0	4
	RNAsieve	2	1	0	1
	Bisque	0	0	0	0
Pseudo	TAPE	33	11	15	7
	Scaden	30	3	6	21
	DWLS	29	15	13	1
	MuSiC	22	14	3	5
	CIBERSORTx	14	1	5	8
	Bisque	3	0	2	1
	RNAsieve	1	0	0	1

Table R3. Performance summary of TAPE and CIBERSORTx on the DEG detection task. The performance is evaluated by the average AUROC in CD8 T cells. Since DEGs are only associated with different conditions, which are not related to cell types' signature genes, we usually care about the case when DEGs are randomly selected. In real cases, the number of DEGs is usually smaller than 1000.

Scenario	DEG type	Similar cell type	DEG number	Performance
1	random	no	100	TAPE > CIBERSORTx
			1000	TAPE > CIBERSORTx
			5000	TAPE ~ CIBERSORTx
2	signature	no	about 200	TAPE ~ CIBERSORTx
3	random	yes	100	both failed, TAPE > CIBERSORTx
			1000	both failed, TAPE > CIBERSORTx
4	signature	yes	about 150	CIBERSORTx > TAPE

Main points:

1) The authors presented figure R1 (Figures 2c and 2d in the updated manuscript) to demonstrate performance per cell type. From this, it is difficult to say that TAPE improves over Scaden for prediction of cell type composition on real bulk datasets. Monaco is the only real bulk dataset there where the median CCC of tape seems higher than Scaden.

Answer: We thank the reviewer for this comment! We totally understand your concern about the performance comparison. To make a clearer and more comprehensive comparison in this revision, we summarized all the results and showed the top3 methods in Table R1 for each evaluation scenario. Table R2 summarizes the numbers of occurrences in Table R1, in which TAPE appears most frequently not only when taking all the scenarios into consideration but also when considering real-bulk and pseudo-bulk separately. Additionally, TAPE occurs most frequently as the best method for real-bulk datasets.

Regarding your comments on specific datasets and experiments with specific evaluation criteria, in the beginning, we followed the performance evaluation metrics (overall CCC) used in Scaden (Menden et al., 2020) to test each method's performance. In our first version manuscript, from our experiments, TAPE has the best overall performance on 4 of 5 real

datasets and is also among the top methods evaluated on the cell-type level. Following the evaluation criteria in SOTA methods, we calculated the overall CCC and MAE values in real-bulk datasets. As shown in Figure R1, TAPE raises up to 48.9% of the CCC and reduces up to 40.6% of the MAE in comparison with Scaden.

Figure R1. Supplementary figure 5. Overall performance of all tested methods on five real datasets. The overall performance is calculated by all the data points of a dataset.

But in the last revision, following your suggestions, we realized that overall CCC is not as informative as CCC for each cell type, and we should go through the details of performance for each cell type. Therefore, we changed the evaluation criteria to cell-type level in the last revision. But since the previous SOTA method Scaden uses the overall performance as evaluation metrics, we still maintain the overall performance in our supplementary files (Supplementary figure 5). At the cell type level, our results show that TAPE achieves the best MAE and the smallest variance. As for the CCC metric, even though the median CCC of TAPE isn't ranked first in sdy67 and microarray datasets, TAPE still shows comparable performance with relatively small variance, indicating the performance of TAPE is robust for all the cell types (Figure R2d). To be specific, TAPE has a median CCC of 0.275 (not as good as Scaden's 0.348), while TAPE's interquartile range on sdy67 is 0.169 (smaller than 0.205 of Scaden on sdy67). Also, we admit that TAPE's median CCC performance (0.386)

on microarray is below the 0.441 of Scaden. However, TAPE's interquartile range of 0.089 shows the robustness of TAPE's prediction among different cell types in comparison to Scaden (0.209). For monaco and ROSMAP_human dataset, not only the median CCC of TAPE is higher than Scaden's (TAPE's median CCC: 0.386 and 0.140; Scaden's median CCC: 0.326 and 0.121), but the interquartile range is also smaller (TAPE's interquartile range: 0.178 and 0.113; Scaden's interquartile range: 0.237 and 0.155). TAPE and Scaden show comparable performance on ROSMAP_mouse dataset. It is worth noticing that TAPE's MAE performance is remarkable in comparison to SOTA methods.

Figure R2. Figure 2c&d in the manuscript. *c* Deconvolution results on simulated data. CCC represents the Lin's concordance correlation coefficient, measuring the concordance between the predicted

fraction and the ground truth. MAE represents mean absolute error, measuring the accuracy of prediction. Higher CCC and lower MAE are better. Each box contains metric values for all the cell types considered in all the tissues. Different color refers to different methods. d Deconvolution results on real data. The columns' labels refer to the datasets. CCC and MAE are used as metrics.

Considering your suggestion, we modify our claims in the abstract of TAPE as:

(Abstract) “Compared with popular methods on several datasets, TAPE has the best overall performance and comparable accuracy at cell type level. Additionally, it is more robust among different cell types, faster, and sensitive to provide biologically meaningful predictions.”

and we added more detailed description about the performance in the manuscript:

(section 2.3) “To be specific, for ROSMAP_human dataset, the median CCC of TAPE is the best (0.140). While Scaden achieves the best median CCC of 0.326 and 0.202 on SDY67 dataset and ROSMAP_mouse dataset, and CIBERSORTx achieves the best median CCC on Monaco’s PBMC dataset and microarray PBMC dataset. Though TAPE’s median CCC on these four datasets is not the highest, the values are comparable with the difference smaller than 0.07. Considering the interquartile range, we can see that the performance of DWLS is close to the best on SDY67 dataset and ROSMAP_mouse dataset. Detailed comparison results are available in the Supplementary Table 8.”

2) Similar is the case for figure R3, where it is really hard to compare TAPE and Scaden. On umi2counts, TAPE definitely seems a lot better in MAE but substantially worse in CCC, indicating poor correlation or predictions and ground truth.

Answer: We thank the reviewer again for the comments related to performance. The overall performance comparison has been discussed extensively in other questions and comments. Regarding this specific experiment, we are sorry for the misunderstanding caused. As this experiment was added during the first revision, perhaps we did not make the logic behind this experiment clear enough. We further clarify it below and have revised the manuscript to increase the readability of the paper.

In Figure R3, we carried out an additional experiment to explore the cell-type level performance of TAPE and other popular methods in the “rare” scenario. Since we do not force TAPE to specifically overfit the pseudo-bulk data but design it to solve practical problems, we admit that TAPE is not the best method in this scenario. Unfortunately, all the methods can not achieve an appealing prediction power for rare cell types on Limb Muscle, Lung, and Marrow datasets as well. Since we only consider rare cell types in this scenario and the data points are not enough for systematic comparison using a t-test, we calculated the average CCC and MAE among different cell types. We found that the top methods evaluated by CCC on umi2counts datatype are Scaden (0.215 on Limb_Muscle), Music (0.238 on Lung), and DWLS (0.150 on Marrow). And the top methods evaluated by MAE on umi2counts datatype are TAPE (0.013 and 0.047 on Limb_Muscle and Marrow) and Music (0.011 on Lung) (Table R1). We admit that TAPE is worse in CCC in comparison with Scaden on umi2counts, which indicates that “TAPE has an obvious limitation in predicting a good correlation for rare cell types” as we mentioned in the caption of Supplementary Figure 2. However, a CCC value below 0.3 seems inadequate to prove to solve the deconvolution problem in the “rare” scenario. Also, we need to mention that even though TAPE shows a relatively good MAE performance in the “rare” scenario, we don't think we resolve this problem. This statement is consistent with what we have mentioned in the last revision:

(section 2.2) “In the ‘rare’ scenario, we only display the metrics for pre-defined rare cell types. The results show that all the methods can not result in a satisfying concordance between prediction and ground truth in this scenario (Figure 2c). Interestingly, although the CCC values are pretty low with those methods, their MAEs are comparable to those in the ‘normal’ scenario, which indicates that those methods can predict a value near ground truth but are not correlated with each other.”

Therefore, the intuition behind this part is not for performance comparison, but to **demonstrate a situation remaining to be tackled by TAPE and all SOTA methods.**

For clearer illustration, we revised the following statement to the caption of Supplementary

Figure 2:

“All the methods can not achieve an appealing prediction power for rare cell types. Even though TAPE can achieve a relatively good performance on MAE (for example on umi2counts datatype, TAPE has the smallest average MAE on Limb_Muscle(0.013) and Marrow(0.047)), TAPE has worse performance on CCC in comparison with other methods, which indicates that TAPE needs further improvement in predicting a good correlation for rare cell types. Meanwhile, we have to point out that, for all the methods, a CCC value below 0.3 seems inadequate to show the deconvolution problem being resolved in the ‘rare’ scenario.”

Figure R3. Appendix Figure 2 in the manuscript

3) Given points 1 & 2, it seems that there are many overstatements of TAPE’s performance compared to SOTA methods. If the author cannot show significant improvement of existing methods, they need to tone down many of their claims throughout the manuscript.

Answer: Thank you for the comments! We totally understand your concern related to performance comparison. And we will clarify it.

Essentially, in the manuscript, we evaluate different methods comprehensively and elaborate on their advantages and shortcomings under **various scenarios**. Understandably, no method, including TAPE, could outperform all the other methods constantly across all the datasets. For your information, we sum up all the deconvolution experiments carried out in this work, hoping to demonstrate TAPE's performance.

1. Evaluation on the pseudo bulk datasets. To fully exploit the advantages of pseudo-bulk data, we defined three deconvolution scenarios: "normal", "rare", and "similar".
 - a. "normal": DWLS achieves the best performance on both metrics and TAPE is comparable to DWLS.
 - b. "rare": The results show that all the methods can not result in a satisfying concordance between prediction and ground truth in this scenario. Though TAPE is not the best algorithm in this scenario, its performance is comparable to DWLS, which focuses on rare cell types.
 - c. "similar": TAPE's performance is the best. It has the best median CCC and median MAE with small interquartile range.
2. Evaluation on the real bulk datasets. TAPE's MAE is the smallest on all the datasets and it achieves the best median CCC value on ROSMAP_human dataset. For other datasets, Scaden is the best on SDY67 and ROSMAP_mouse and CIBERSORTx is the best on Monaco's PBMC and microarray dataset. In detail, the median CCC value of TAPE on ROSMAP_mouse and Monaco's PBMC datasets are very close to the best. Specifically, the difference between TAPE and Scaden is only 0.0001 on ROSMAP_mouse dataset, and the difference between TAPE and CIBERSORTx is 0.02 on Monaco's dataset.

Please kindly refer to the following Table R1 and R2. Essentially, TAPE has the most frequent occurrences as the top3 methods under all the scenarios. More importantly, it is the best method for 16 out of 20 evaluation scenarios on the real-bulk datasets.

*Table R1. Performance summary of TAPE and SOTA methods. Here, we list the **top3** methods in order in different scenarios and datasets. The performance comparison between box plots is evaluated by two-sided t-test. The order is based on the p-value, a small p-value represents higher performance. In some scenarios with only two data points, we only compute the average value for comparison.*

Datatype	Scenario	Dataset	Metrics						
			CCC		MAE				
			Overall	P-value (used for comparing the box plot)	Overall	P-value (used for comparing the box plot)			
Real-bulk		sdv67	TAPE, Scaden, CSx	DWLS, Scaden, TAPE	TAPE, Scaden, MuSiC	TAPE, Scaden, CSx			
		monaco	TAPE, CSx, Scaden	TAPE, CSx, Scaden	TAPE, CSx, Scaden	TAPE, CSx, Scaden			
		microarray	Scaden, TAPE, DWLS	Scaden, CSx, TAPE	TAPE, Scaden, CSx	TAPE, Scaden, CSx			
		rosmap_h	TAPE, Scaden, RNAsieve	TAPE, Scaden, MuSiC	TAPE, Scaden, MuSiC	TAPE, Scaden, MuSiC			
		rosmap_m	TAPE, CSx, DWLS	RNAsieve, DWLS, TAPE	TAPE, DWLS, Scaden	TAPE, DWLS, Scaden			
Pseudo-bulk	umi2counts	normal	Limb_Muscle	DWLS, CIBERSORTx, Scaden	DWLS, CIBERSORTx, Scaden	DWLS, Scaden, CIBERSORTx	Scaden, DWLS, CIBERSORTx		
			Lung	DWLS, Bisque, TAPE	DWLS, Bisque, TAPE	DWLS, Scaden, TAPE	DWLS, Scaden, TAPE		
			Marrow	DWLS, TAPE, Scaden	DWLS, TAPE, CIBERSORTx	TAPE, DWLS, Scaden	TAPE, DWLS, CIBERSORTx		
			Limb_Muscle	not applicable	Scaden, DWLS, TAPE	not applicable	TAPE, DWLS, Scaden		
			Lung	not applicable	MuSiC, CIBERSORTx, Scaden	not applicable	MuSiC, DWLS, RNAsieve		
		rare*	Marrow	not applicable	DWLS, Scaden, TAPE	not applicable	DWLS, TAPE, MuSiC		
			similar distinguishment*	Marrow	not applicable	TAPE, DWLS, CIBERSORTx	not applicable	TAPE, CIBERSORTx, Scaden	
			similar transferring*	Marrow	not applicable	DWLS, TAPE, Scaden	not applicable	DWLS, TAPE, Scaden	
			counts2umi	normal	Limb_Muscle	TAPE, DWLS, Scaden	DWLS, TAPE, MuSiC	TAPE, Scaden, DWLS	TAPE, DWLS, Scaden
					Lung	MuSiC, TAPE, Scaden	MuSiC, DWLS, Bisque	MuSiC, TAPE, Scaden	MuSiC, TAPE, Scaden
	Marrow	MuSiC, TAPE, Scaden			MuSiC, TAPE, Scaden	TAPE, Scaden, MuSiC	TAPE, MuSiC, Scaden		
	Limb_Muscle	not applicable			MuSiC, DWLS, TAPE	not applicable	MuSiC, TAPE, Scaden		
	Lung	not applicable			DWLS, MuSiC, CIBERSORTx	not applicable	CIBERSORTx, DWLS, MuSiC		
	rare*	Marrow		not applicable	MuSiC, CIBERSORTx, Scaden	not applicable	Scaden, DWLS, MuSiC		
		similar distinguishment*		Marrow	not applicable	MuSiC, TAPE, CIBERSORTx	not applicable	TAPE, MuSiC, CIBERSORTx	
		similar transferring*		Marrow	not applicable	MuSiC, TAPE, Scaden	not applicable	MuSiC, TAPE, Scaden	

*only compute the average performance of all available data points because of the small number of data points

Table R2. Statistical table of the number of occurrences in Table R1.

		Occurrences	Top1 Occurrences	Top2 Occurrences	Top3 Occurrences
All	TAPE	53	27	16	10
	Scaden	48	5	16	27
	DWLS	35	16	16	3
	MuSiC	26	14	3	9
	CIBERSORTx	24	1	11	12
	RNAsieve	3	1	0	2
	Bisque	3	0	2	1
Real	TAPE	20	16	1	3
	Scaden	18	2	10	6
	CIBERSORTx	10	0	6	4
	DWLS	6	1	3	2
	MuSiC	4	0	0	4
	RNAsieve	2	1	0	1
	Bisque	0	0	0	0
Pseudo	TAPE	33	11	15	7
	Scaden	30	3	6	21
	DWLS	29	15	13	1
	MuSiC	22	14	3	5
	CIBERSORTx	14	1	5	8
	Bisque	3	0	2	1
	RNAsieve	1	0	0	1

We totally agree with your suggestion that we should not overclaim the manuscript, which may cause reader's abuse and incorrect usage of the proposed method. We toned down in the last revision when we found that TAPE's CCC performance is not superior at the cell-type level. Since the last revision, following your suggestions, we have rewritten the

results section and we describe the results with a fair description. Though TAPE's overall CCC performance is better than other methods, we have admitted TAPE's medium CCC value on cell-type level is not always the best on all real datasets as (section 2.3) "For the CCC metric, although other methods like DWLS and Scaden surpass TAPE,"

However, it is still worth noticing that TAPE shows comparable performance with relatively small variance, indicating that

(section 2.3) "the prediction performance of TAPE is similar for all the cell types and hence robust".

Additionally, TAPE's clinical performance is what we attached great importance to. Experiments demonstrate TAPE's sensitivity to biological changes as well as its ability to predict cell-type-specific GEP tissue-adaptively.

We toned down our claims on the cell-type level performance of TAPE in this revision:

(section 1) "Empirically, our method could achieve a better overall performance than previous state-of-the-art methods. When evaluated on cell-type level, TAPE has the best performance of MAE, and comparable CCC with relatively small variance on real datasets."

4) For Figure R4, I would have appreciated going deeper into cellular subsets rather than evaluating performance on just CD4 and CD8 T cells, which are very broad T cell subtypes. To provide useful cell type deconvolution to biologists (or immunologists) it would be nice to know if TAPE can predict naive and memory subsets well, for instance.

Answer: Thank you for raising the question about the problem of going deeper into cellular subsets of TAPE. According to the datasets we used, we find that Monaco's PBMC datasets are appropriate for the subset deconvolution task since it contains many subtypes in the flow cytometer file. To annotate the single-cell data with subtypes labels, we use the newly published immune cells annotator CellTypist (Domínguez Conde et al., 2022) to label the cell subtypes for data8k dataset. The reason why we choose CellTypist is that the training data contains nearly one million hand-labeled immune cells with clear cell subtypes, and in practice, it can achieve very good results on many datasets. Moreover, in order to obtain a plausible annotation, we only select the cells with a confidence score greater than 0.8. According to the results of CellTypist, we obtain the annotated data8k datasets with 17

immune cell types (Figure R4 a). Combining the cell types in Monaco's dataset and cell types in the single-cell dataset, we finally define 13 cell types. The relations are displayed in the following table:

Table R4. Relations between defined cell types and existing cell types in original datasets. Notice that we merge some cell types to make the categories identical.

Defined cell types	data8k cell types	monaco's cell types
NK	CD16+ NK cells NK cells	NK
Monocytes	Classical monocytes Non-classical monocytes	Monocytes C Monocytes N Monocytes L
mDC	DC1 DC2	mDCs
pDC	pDC	pDCs
Naïve B	Naïve B cells	B Naïve B Exhausted
Memory B	Memory B cells	B SM B NSM
MAIT	MAIT cells	MAIT
Naïve CD8 T	Tcm/Naïve cytotoxic T cells	T CD8 Naïve
Naïve CD4 T	Tcm/Naïve helper T cells	T CD4 Naïve
non-Naïve CD4 T	Tem/Effector helper T cells	Tfh Th1 Th1/Th17 Th17 Th2 T CD4 TE
non-Naïve CD8 T	Tem/Temra cytotoxic T cells Tem/Trm cytotoxic T cells	T CD8 CM T CD8 EM T CD8 TE
Treg	Regulatory T cells	Tregs
Unknown	HSC/MPP	Progenitors Plasmablasts T gd Vd2 Tdg non-Vd2 Neutrophils LD Basophiles LD

After this preparation, we test the deconvolution performance for each method. The results show that all the methods can not deal with these detailed cell type deconvolution tasks since the CCC is pretty low and some cell types' CCC is negative. The probable reason

contains two points: 1. many cell types' ratio is pretty low and we have shown that all the methods can not deal with the “rare” scenario well; 2. there exist many similar cell types and the signal (signature gene) / noise (cross-protocol sequencing error) ratio may decrease and leads to failure. From Figure R4, we can tell that TAPE has higher CCC and lower MAE than SOTA methods. However, we would not claim that TAPE can resolve the problem of predicting cell subtypes due to the low absolute CCC values.

Figure R4. Deconvolution of immune cell subsets. a. The annotated cell types of data8k datasets. This is produced from the pipeline of CellTypist. Only cells with a confidence score greater than 0.8 were selected. b,c. Deconvolution performance of current methods on Monaco's dataset. Only TAPE predicts positive CCC for all cell types.

Following your insightful comment, we have added this phenomenon to the manuscript and point out it as a common limitation of current methods:

(section 2.3) “Moreover, we test all the methods' ability of deconvolving immune cell subtypes. With 13 defined cell subtypes, all the methods can not achieve satisfying results (Supplementary Figure 4, median CCC < 0.1), which clearly shows the common limitation of current methods.”

(section 4.1) “When we used it to test whether deconvolution methods can achieve good performance with immune cell subtypes, we merged all 30 cell types into 13 cell subtypes (Supplementary Table 1). The similar subtypes are defined as ‘mDC’ and ‘pDC’, ‘naive CD4 T cell’ and ‘non-naive CD4 T cell’, ‘naive CD8 T cell’ and ‘non-naive CD8 T cell’, and ‘naive B cell’ and memory B cell’.”

5) Since TAPE doesn't show clear improvement in the estimation of cell proportions, the novelty of TAPE lies in the fact that it can estimate gene expression at scale (i.e. of all genes present in the single-cell data). However experiments presented by authors still seem cherry-picked. If TAPE can predict all genes, what is the need to select only 100 genes, especially on simulated data where TAPE has access to the ground truth?

Answer: Thank you for your comment. Firstly, we would like to clarify TAPE's performance in the estimation of cell proportions. To make things clearer, please refer to the following table for a comprehensive comparison.

Table R1. Performance summary of TAPE and SOTA methods. Here we list the **top3** methods in order in different scenarios and datasets. The performance comparison between box plots is evaluated by two-sided t-test. The initial assumption is TAPE's performance is better than other methods. If $p > 0.5$, TAPE's performance is not better than other methods. The order is based on the p-value, a small p-value represents higher performance. In some scenarios with only two data points, we only compute the average value for comparison.

Datatype	Scenario	Dataset	Metrics				
			CCC		MAE		
			Overall	P-value (used for comparing the box plot)	Overall	P-value (used for comparing the box plot)	
Real-bulk		sdyl67	TAPE, Scaden, CSx	DWLS, Scaden, TAPE	TAPE, Scaden, MuSIC	TAPE, Scaden, CSx	
		monaco	TAPE, CSx, Scaden	TAPE, CSx, Scaden	TAPE, CSx, Scaden	TAPE, CSx, Scaden	
		microarray	Scaden, TAPE, DWLS	Scaden, CSx, TAPE	TAPE, Scaden, CSx	TAPE, Scaden, CSx	
		rosmap_h	TAPE, Scaden, RNAsieve	TAPE, Scaden, MuSIC	TAPE, Scaden, MuSIC	TAPE, Scaden, MuSIC	
		rosmap_m	TAPE, CSx, DWLS	RNAsieve, DWLS, TAPE	TAPE, DWLS, Scaden	TAPE, DWLS, Scaden	
Pseudo-bulk	umi2counts	normal	Limb_Muscle	DWLS, CIBERSORTx, Scaden	DWLS, CIBERSORTx, Scaden	DWLS, Scaden, CIBERSORTx	Scaden, DWLS, CIBERSORTx
			Lung	DWLS, Bisque, TAPE	DWLS, Bisque, TAPE	DWLS, Scaden, TAPE	DWLS, Scaden, TAPE
			Marrow	DWLS, TAPE, Scaden	DWLS, TAPE, CIBERSORTx	TAPE, DWLS, Scaden	TAPE, DWLS, CIBERSORTx
		rare*	Limb_Muscle	not applicable	Scaden, DWLS, TAPE	not applicable	TAPE, DWLS, Scaden
			Lung	not applicable	MuSIC, CIBERSORTx, Scaden	not applicable	MuSIC, DWLS, RNAsieve
			Marrow	not applicable	DWLS, Scaden, TAPE	not applicable	DWLS, TAPE, MuSIC
	similar distinguishment*	Marrow	not applicable	TAPE, DWLS, CIBERSORTx	not applicable	TAPE, CIBERSORTx, Scaden	
	similar transferring*	Marrow	not applicable	DWLS, TAPE, Scaden	not applicable	DWLS, TAPE, Scaden	
	counts2umi	normal	Limb_Muscle	TAPE, DWLS, Scaden	DWLS, TAPE, MuSIC	TAPE, Scaden, DWLS	TAPE, DWLS, Scaden
			Lung	MuSIC, TAPE, Scaden	MuSIC, DWLS, Bisque	MuSIC, TAPE, Scaden	MuSIC, TAPE, Scaden
			Marrow	MuSIC, TAPE, Scaden	MuSIC, TAPE, Scaden	TAPE, Scaden, MuSIC	TAPE, MuSIC, Scaden
		rare*	Limb_Muscle	not applicable	MuSIC, DWLS, TAPE	not applicable	MuSIC, TAPE, Scaden
			Lung	not applicable	DWLS, MuSIC, CIBERSORTx	not applicable	CIBERSORTx, DWLS, MuSIC
			Marrow	not applicable	MuSIC, CIBERSORTx, Scaden	not applicable	Scaden, DWLS, MuSIC
		similar distinguishment*	Marrow	not applicable	MuSIC, TAPE, CIBERSORTx	not applicable	TAPE, MuSIC, CIBERSORTx
similar transferring*		Marrow	not applicable	MuSIC, TAPE, Scaden	not applicable	MuSIC, TAPE, Scaden	

Your concern about the selection of genes is reasonable. There are two main reasons for

selecting 100 genes.

Firstly, by randomly selecting 100 genes among 10,000 genes in CD8 T cells as up-regulated genes, the picked genes are unbiased. And in many **real cases**, the number of DEGs is around 100 (data from Table2 of Zhao et al., 2018, where the max number of detected DEGs is 833 and the average number of DEGs is around 100).

Secondly, in the ROC graph, the horizontal axis FPR is only concerned with negative samples; the vertical axis TPR is only concerned with positive samples. Therefore, the horizontal and vertical axes are not affected by the proportion of positive and negative samples, and the integration, which is AUC (Area Under The Curve) ROC (Receiver Operating Characteristics) curve, is certainly not affected by them. Thus, AUROC is insensitive to the prevalence. Hence, we assume that our setting is reasonable.

In this revision, we consider all your related questions (including Q7 and Q8) about the DEG detection task and design a comprehensive test with four scenarios to test both methods' ability to detect DEGs. In Figure R5, the four scenarios are "randomly selected DEGs without similar cell type", "randomly selected DEGs with similar cell type", "signature genes as DEGs without similar cell type" and "signature genes as DEGs with similar cell type". In this question, since you mainly concern that testing on only 100 genes may be biased, following your suggestion, we carried out additional experiments in the settings with more genes selected as DEGs. In this revision, we mainly test the scenario with 1,000 DEGs instead of using all the genes as DEGs because the number of DEGs is **usually under 1000** (data from Table2 of Zhao et al., 2018, where the max number of detected DEGs is 833 and the average DEGs is around 100). For your interest, we also design an experiment with 5,000 DEGs as an extreme case to show the prediction ability of TAPE and CIBERSORTx. In Figure R5, we notice that when the number of DEGs is under 1,000, TAPE has better predictive power than CIBERSORTx. The average AUROC in CD8 T cell for CSx and TAPE are 0.5578 and 0.6538, respectively. But in the extreme case, when there are 5,000 DEGs, the performance of CIBERSORTx increases and is better than TAPE. The main reason why

TAPE's performance drops in the extreme case is that TAPE intends to predict normal genes as DEG when the up-regulated gene ratio increases. Since the extreme case is not expected to appear in usual studies, we still draw the conclusion that TAPE's performance is better than CIBERSORTx when the DEGs are randomly selected, and there are not any similar cell types.

Following your comments, we have revised the below section in the manuscript.

(section 2.7) "In addition to the normal scenario that there are only 100 randomly selected DEGs with four non-similar cell types in simulated bulk samples, we designed comprehensive tests with four scenarios to benchmark TAPE and CIBERSORTx's performances. The four scenarios are: 'randomly selected DEGs without similar cell type', 'randomly selected DEGs with similar cell type', 'signature genes as DEGs without similar cell type', and 'signature genes as DEGs with similar cell type'. In detail, we set up a series of simulated bulk data to detect DEGs as we mentioned before. However, we used similar cell types or changed the number of randomly selected genes, or used signature genes as DEGs in this test. Specifically, for the 'similar' scenario, we used similar cell types like CD4 T cells and CD8 T cells together with two other cell types, namely monocytes and NK cells. In the scenarios where DEGs are randomly selected, the number of DEGs ranges from 100 to 5,000. For the 'signature genes as DEGs' scenario, we up-regulated the signature genes of CD8 T cells produced by CIBERSORTx in the simulated bulk samples. From the results (Supplementary Figure 10), we can have four conclusions: 1. TAPE's predictive power is better than CIBERSORTx when the randomly selected DEGs are less than 1,000; 2. both methods can achieve good performance when the DEGs are signature genes and there are not any similar cell types; 3. both methods can not distinguish DEGs from CD8 T cell rather than CD4 T cell if the DEGs are randomly selected; 4. CIBERSORTx is better than TAPE if the DEGs are signature genes and there exist similar cell types. Interestingly, from points 2 and 4, it seems that TAPE can learn the signature genes between distinguished cell types but not exactly enough to distinguish similar cell types. In all, considering all the scenarios,

we display that each method has its own advantages and disadvantages and it can be seen as a guide for researchers to decide which method to use.”

We added this additional experiment to the Supplementary of the manuscript:

Figure R5. Comprehensive tests for TAPE and CIBERSORTx in four scenarios. The upper left scenario uses randomly selected DEGs and it does not contain similar cell types in single-cell profiles. The number of DEGs ranges from 1,00 to 5,000. However, the number of DEGs is usually below 1,000 (Zhao et al., 2018). The second one is the “signature genes as DEGs without similar cell types” scenario which is located in the upper right. The bottom left area is the “randomly selected DEGs with similar cell types” scenario, and the bottom right one is the “signature genes as DEGs with similar cell types” scenario. All the tests use AUROC as criteria, and the high AUROC value is expected to only appear in CD8 T cells. In the first scenario, TAPE is better than CIBERSORTx when the number of DEGs is below 5,000 (average AUROC in CD8 T cells for CSx and TAPE are 0.5578 and 0.6538

respectively). In the second scenario, both methods can achieve a good predictive power (average AUROC in CD8 T cells for CSx and TAPE are 0.7639 and 0.7611 respectively). In the third scenario, both methods can not distinguish DEGs from similar cell types well but TAPE's performance is a little better (average AUROC in CD8 T cells for CSx and TAPE are 0.5146 and 0.5249 respectively). In the last scenario, CIBERSORTx behaves better than TAPE because of the incorporation of the signature matrix (average AUROC in CD8 T cell for CSx and TAPE are 0.7466 and 0.5336 respectively).

We also summarize the experimental results below for the reader's quick reference.

Table R3. Performance summary of TAPE and CIBERSORTx on the DEG detection task. The performance is evaluated by the average AUROC in CD8 T cells. Since DEGs are only associated with different conditions which are not related to cell types' signature genes, we usually care about the case that DEGs are randomly selected.

Scenario	DEG type	Similar cell type	DEG number	Performance
1	random	no	100	TAPE > CIBERSORTx
			1000	TAPE > CIBERSORTx
			5000	TAPE ~ CIBERSORTx
2	signature	no	about 200	TAPE ~ CIBERSORTx
3	random	yes	100	both failed, TAPE > CIBERSORTx
			1000	both failed, TAPE > CIBERSORTx
4	signature	yes	about 150	CIBERSORTx > TAPE

6) Thank you for adding an experiment in estimation of gene expression (Figure R9). I would be interested in knowing if the correlation of relative expression at gene level is also good.

Answer: Thank you very much for the insightful suggestion! Following your comment, we calculated the relevant CCC values of TAPE and compare the performance with CIBERSORTx at the gene level in this revision. Though our performance in adapt2real scenario (median CCC 0.2127) is better than CIBERSORTx (median CCC 0.0627), there is still a large room for improvement. To be specific, we first added an experiment to show gene concordance of CIBERSORTx and then calculate the CCC at the cell type level and label them in Figure R6 a&b. Except for the similar result of TAPE and CIBERSORTx on monocytes, the CCC values of TAPE were significantly higher than CIBERSORTx on the other five cell types. Then, we rearrange the data from Figure R6 a&b at the gene level. That is, we calculate the correlation for each gene with the gene expression value from 6 cell types.

Here, we use an enhanced box plot to show the results (Figure R6 c). We can see that about half of the genes' CCC is lower than 0.25 and about 25% of genes' CCC are negative. The main reason is that some gene expression values are predicted as zero while they should not be zero in reality (Figure R6 c). For your information, we also use CIBERSORTx's group mode to analyze the genes' concordance between predicted gene expression value and real gene expression value from single-cell profiles. For CCC comparison at the gene level, we do not use the filtered inferred GEP of CIBERSORTx (because gene numbers will be different in different cell types). We can see that, for CIBERSORTx, many genes are predicted as zero and the CCC values at the gene level are not satisfying (Figure R6 b). We think that CIBERSORTx still shows its shortage in using single-cell profiles as references. Because in the original article (Figure 3 b,d from Newman et al., 2019), the authors of CIBERSORTx only display the value from microarray data (note the x-axis label is MAS5). We can easily tell from Figure R6 c that the correlation of relative expression of TAPE at gene level in adapt2real scenario is better than CIBERSORTx's.

Really thanks for your constructive comments, we have added the gene-level CCC results to the supplementary files and mentioned them in the manuscript.

(section 3) "Firstly, when we study the correlation at the gene level using 'overall' mode (Supplementary Figure 7), about 30% of the predicted genes have a negative correlation. Although our method's performance (median CCC 0.2127) is better than CSx (median CCC 0.0627), there is still large room for improvement."

Figure R6. Gene concordance of TAPE and CSx. **a** Figure 4b in the manuscript. Concordance between the predicted relative gene expression value in real bulk data and the relative gene expression value in single-cell data of TAPE. **b** Concordance between the predicted relative gene expression value in real bulk data and the relative gene expression value in single-cell data of CSx. **c** Gene level CCC of TAPE and CSx (median CCC of TAPE and CSx in adapt2real scenario are 0.2171 and 0.0627, respectively).

7) By design CibersortX only imputes genes that are likely to be significant in at least one cell type. It is therefore improper to compare TAPE and CibersortX on genes that are insignificant as authors did for simulated dataset. Could you please compare using significant genes. In addition, the evaluation of CibersortX on AD brains is missing.

Answer: Thanks for the comment. We totally understand your concerns and design new experiments following your awesome suggestion.

First, we want to clarify that the significant genes are **not really significant in practice** because they are the DEGs among all the cell types in single-cell datasets. When the cell types are changing in a single-cell dataset, the calculated DEGs between cell types will change. So, they should be called signature genes instead of significant genes. We apologize for using misleading words. Actually, we did not consider comparing TAPE and CIBERSORTx on signature genes because the actual DEGs might not be the signature gene of one cell type. For example, in our manuscript section 2.7, the DEG between HIV PBMC samples under two conditions is RAB11FIP5 which is not a signature gene in any cell type (calculated by CIBERSORTx). Therefore, we think that only considering the signature gene may not be actually useful in real applications. But, for your information, we also test both methods in this situation. We designed two scenarios (Figure R7), the first is “signature genes as DEGs without similar cell type”, and the second one is “signature genes as DEGs with similar cell type (CD4 T cell and CD8 T cell)”. Of note, the signature genes are the signature genes of CD8 T cells calculated by CIBERSORTx, and the number of signature genes is about 200 in the first scenario and 130 in the second scenario. In the first scenario, we can see that both methods can achieve a really good performance. Comparing this result with the scenario “randomly selected DEGs without similar cell type”, we can see that the predictive power of both methods increases a lot (with 100 randomly selected DEGs, the average AUROC value of CIBERSORTx increases from 0.53 to 0.76, the average AUROC value of TAPE increases from 0.67 to 0.76). This phenomenon indicates that although TAPE is **not designed by using signature genes** as input, the deep learning model may still learn the signature genes during the training process with simulated bulk data. However, in the second scenario, CIBERSORTx shows its superior performance over TAPE. This means that TAPE can not distinguish the signature genes between similar cell types and CIBERSORTx can handle this scenario pretty well since it is designed by using signature genes. But considering that the actual DEGs may not be the signature genes, we think that this scenario is not as common as the scenario with randomly selected DEGs, and TAPE can achieve a better performance in a common scenario.

In this revision, we have modified what we discussed in the last revision:

(section 3) Thirdly, we notice that both CIBERSORTx and our method can not distinguish DEGs from similar cell subtypes correctly if the DEGs are not signature genes (Supplementary Figure 9) which means that their resolution is still limited. But CIBERSORTx has displayed its advantages in distinguishing signature DEGs from similar cell types because of the incorporation of the signature matrix (Supplementary Figure 10). Though our method cannot precisely predict DEGs from cell subtypes or have better performance than CIBERSORTx if all signature genes are DEGs which probably does not occur in the real world, it still reduces the potential candidates by excluding unrelated cell types. So, our method is still useful and can be applied in real-life scenarios to accelerate biological research.

Thanks for your comments regarding the performance of CIBERSORTx. In our manuscript Figure 4g, we want to establish the correctness of our method firstly, that is showing our method can be adaptive, and that this adaptation is reasonable. Thus, we did not display the predicted value of CIBERSORTx. For your information, we use the “group” mode of CIBERSORTx to predict the gene expression value of each cell type (Figure R8). The results show that, though CIBERSORTx can predict a high NRG1 value in Inhibited Neurons it can not predict a high NRG1 value in excited Neurons. We have added this comparison in the manuscript and modify the caption:

(section 2.6) “In this test, we also used the ‘group’ mode of CIBERSORTx to predict the expression value of NRG1 in different cell types. The results show that although CIBERSORTx can predict high expression value of NRG1 in InNeurons, it can not predict an expected high value in ExNeurons.”

Figure R7. Comprehensive tests for TAPE and CIBERSORTx in four scenarios. The upper left scenario uses randomly selected DEGs and it does not contain similar cell types in single-cell profiles. The number of DEGs ranges from 1,00 to 5,000. However, the number of DEGs is usually below 1,000 (Zhao et al., 2018). The second one is the “signature genes as DEGs without similar cell types” scenario which is located in the upper right. The bottom left area is the “randomly selected DEGs with similar cell types” scenario and the bottom right one is the “signature genes as DEGs with similar cell types” scenario. All the tests use AUROC as criteria, and the high AUROC value is expected to only appear in CD8 T cells. In the first scenario, TAPE is better than CIBERSORTx when the number of DEGs is below 5,000 (average AUROC in CD8 T cells for CSx and TAPE are 0.5578 and 0.6538 respectively). In the second scenario, both methods can achieve a good predictive power (average AUROC in CD8 T cells for CSx and TAPE are 0.7639 and 0.7611 respectively). In the third scenario, both methods can not distinguish DEGs from similar cell types well but TAPE’s performance is a little better (average AUROC in CD8 T cells for CSx and TAPE are 0.5146 and 0.5249 respectively). In the last scenario, CIBERSORTx behaves better than TAPE because of the incorporation of the signature matrix (average AUROC in CD8 T cell for CSx and TAPE are 0.7466 and 0.5336 respectively).

Figure R8. The relative gene expression value of NRG1 from different sources. Figure 4g in the manuscript. The dashed line represents the total relative NRG1 expression value in the AD patients' brain tissue. The missing column means the relative gene expression value is zero.

8) To evaluate gene expression estimation, it would be necessary to know how cell type-specific gene expression from highly resolved samples cluster? Are similar cell types closer to each other?

Answer: Thank you for asking this excellent question! This question is really interesting. In the last revision, when we compare the DEG detection ability, we have already included a set of experiments with similar cell types CD4 T cell and CD8 T cell (section 2.7, last paragraph). The results show that both TAPE and CIBERSORTx can not distinguish DEGs from CD8 T cells rather than CD4 T cells when the DEGs are randomly selected. So, we think the predicted GEPs between similar cell types are close. In this revision, considering your questions 5 and 7, we have added more experiments to compare these two methods in the "similar" scenario. Firstly, in the scenario "randomly selected DEGs with similar cell type", we increase the number of selected DEGs to 1,000, and the results show that both methods still mix the DEGs from CD8 T cell up with CD4 T cell (Figure R9). Secondly, considering

your suggestion in question 7, we test their performance in the “signature genes as DEGs with similar cell type” scenario. As we expected, CIBERSORTx can distinguish the DEGs from CD8 T cells well, but TAPE can not (Figure R9). The reason is that CIBERSORTx was designed to use the signature genes, and the signature genes are calculated by the CIBERSORTx. We really appreciate your advice. Now, we clearly know the advantages and limitations of both methods in different scenarios. We have rewritten section 2.7 with the comprehensive experiments and discussed the limitations as well in the manuscript.

(section 2.7) In addition to the normal scenario that there are only 100 randomly selected DEGs with four non-similar cell types in simulated bulk samples, we designed comprehensive tests with four scenarios to benchmark TAPE and CIBERSORTx's performances. The four scenarios are: “randomly selected DEGs without similar cell type”, “randomly selected DEGs with similar cell type”, “signature genes as DEGs without similar cell type” and “signature genes as DEGs with similar cell type”. In detail, we set up a series of simulated bulk data to detect DEGs as we mentioned before. But we used similar cell types or changed the number of randomly selected genes or used signature genes as DEGs in this test. Specifically, for the similar scenario, we used similar cell types like CD4 T cells and CD8 T cells together with two other cell types, namely monocytes and NK cells. In the scenarios where DEGs are randomly selected, the number of DEGs ranges from 100 to 5,000. For the “signature genes as DEGs” scenarios, we up-regulated the signature genes of CD8 T cells produced by CIBERSORTx in the simulated bulk samples. From the results (Supplementary Figure 10), we can have four conclusions: 1. TAPE's predictive power is bigger than CIBERSORTx when the randomly selected DEGs are below 1,000; 2. both methods can achieve a good performance when the DEGs are signature genes and there is not any similar cell types; 3. both methods can not distinguish DEGs from CD8 T cell rather than CD4 T cell if the DEGs are randomly selected; 4. CIBERSORTx is better than TAPE if the DEGs are signature genes and there exists similar cell types. Interestingly, from point 2 and 4, it seems that TAPE can learn the signature genes between distinguished cell types but not exactly enough to distinguish similar cell types. In all, considering all the scenarios,

we display that each method has its own advantages and disadvantages and it can be seen as a guide for researchers to decide which method to use.

(section 3) Thirdly, we notice that both CIBERSORTx and our method can not distinguish DEGs from similar cell subtypes correctly if the DEGs are not signature genes (Supplementary Figure 9) which means that their resolution is still limited. But CIBERSORTx has displayed its advantages in distinguishing signature DEGs from similar cell types because of the incorporation of the signature matrix (Supplementary Figure 10). Though our method cannot precisely predict DEGs from cell subtypes or have better performance than CIBERSORTx if all signature genes are DEGs which probably does not occur in the real world, it still reduces the potential candidates by excluding unrelated cell types. So, our method is still useful and can be applied in real-life scenarios to accelerate biological research.

Figure R9. Comprehensive tests for TAPE and CIBERSORTx in four scenarios. The upper left scenario uses randomly selected DEGs, and it does not contain similar cell types in single-cell profiles. The number of DEGs ranges from 1,00 to 5,000. However, the number of DEGs is usually below 1,000 (Zhao et al., 2018). The second one is the “signature genes as DEGs without similar cell types” scenario which is located in the upper right. The bottom left area is the “randomly selected DEGs with similar cell types” scenario and the bottom right one is the “signature genes as DEGs with similar cell types” scenario. All the tests use AUROC as criteria, and the high AUROC value is expected to only appear in CD8 T cells. In the first scenario, TAPE is better than CIBERSORTx when the number of DEGs is below 5,000 (average AUROC in CD8 T cells for CSx and TAPE are 0.5578 and 0.6538 respectively). In the second scenario, both methods can achieve a good predictive power (average AUROC in CD8 T cells for CSx and TAPE are 0.7639 and 0.7611 respectively). In the third scenario, both methods can not distinguish DEGs from similar cell types well but TAPE’s performance is a little better (average AUROC in CD8 T cells for CSx and TAPE are 0.5146 and 0.5249 respectively). In the last scenario, CIBERSORTx behaves better than TAPE because of the incorporation of the signature matrix (average AUROC in CD8 T cell for CSx and TAPE are 0.7466 and 0.5336 respectively).

9) Thank you for the clear explanation of the reasoning behind usage of Gaussian noise.

The authors added “Here, we add noise to the simulated data because we want to make this pseudo-bulk test more difficult and closer to the real cases, instead of toy simulations.” As far as I know adding Gaussian noise may not necessarily mimic real bulk data better. Further explanation/experiment is necessary to show that Gaussian noise indeed makes simulations closer to real bulks.

Answer: Thanks for raising this concern. We totally agree with your opinion. The problem is that the description in our last response is ambiguous. Here, “the real case” doesn’t refer to “the real bulk data” but refers to “the methods’ performances when deconvolving real data”. We do not mean to make the pseudo-bulk gene expression pattern to be exactly like the real bulk gene expression pattern, which is almost impossible for now. What we want to do is

increase the difficulty for deconvolution methods to deal with pseudo-bulk data. If the pseudo-bulk data does not contain any noises, almost all the methods can achieve an overall CCC over 0.95, while on the real datasets, many of them can only achieve an overall CCC around 0.4. Therefore, we want to add some noise to make the pseudo-bulk data as difficult as real bulk data. That's why we said, "closer to the real cases, instead of toy simulation". Specifically, we choose Gaussian noise because it is common modeling of the batch effect of bulk gene expression. For example, the famous batch effect correction method ComBat (Johnson et al., 2006, cited by 5567) models the additive batch effect using Gaussian distribution as prior and achieves a good performance. We are sorry for the unclear response, and we really appreciate your suggestions. In this revision, we rewrite this sentence in the manuscript as:

(section 2.2) "To make this pseudo-bulk test as difficult as the real bulk test instead of trivial linear regression task, we added Gaussian noise [17] (0.01 times random value generated from a Gaussian distribution with gene expression mean and variance for each gene) and randomly masked 20% genes for each pseudo-bulk sample."

Minor points:

10) *The authors responded with "In this paper, we add Gaussian noise and dropouts."*

However, it is unclear how adding dropouts to single-cell data makes simulations better or closer to the real bulks. If anything, real bulk contains more non-zero genes.

Answer: Thanks for your comments. Principally, we use the pseudo-bulk test because we follow the original article of Scaden (Menden et al., 2020) and use the pseudo-bulk test as an initial estimation of each method's performance in different scenarios.

(section 2.2) "Since a real bulk dataset with its corresponding cell type fractions assessed by traditional experimental methods (e.g., flow cytometry) is rare, and it is hard to analyze how the batch effect would affect deconvolution performance, it is necessary to conduct a pseudo-bulk test..."

We understand your concern that adding noise does not make pseudo-bulk data closer to real bulk data. Also, we do not aim at making the expression pattern of pseudo-bulk data closer to the real bulk one. We apologize for the misleading response. In our setting, adding noise to the pseudo-bulk data makes the task more difficult and is not trivial.

As we know, the relationship between the real bulk data and single-cell data is very complex and highly non-linear. Thus, it is almost impossible to truly simulate the real bulk data by simply adding many single-cell profiles. Specifically, pseudo-bulk data has already contained many zeros. For example, the pseudo-bulk data we synthesized in the manuscript usually have 20,000 genes, but 5,000 gene expression values are zero. In contrast, as you mentioned, real bulk only contains about 2% zero-value genes (monaco's dataset). That is, it is impossible to mimic pseudo-bulk data as real-bulk data. In our experiments, after adding 20% dropouts, there are about 8000 zero-value genes in the pseudo-bulk, and we still consider it as pseudo-bulk data rather than real bulk data. Therefore, adding dropout into the pseudo-bulk RNA samples means that we **increase the dropout rate ratio in single-cell profiles** since the pseudo-bulk RNA-seq data is the sum of many single-cell profiles and this scenario is quite common when the quality of single-cell data is not good enough. Although pseudo-bulk data is still pseudo-bulk data after adding dropouts, the deconvolution performance of each method drops. What we want is **"making the difficulty of pseudo-bulk test closer to the real bulk test"**. Because most methods can achieve quite good performance without adding noise and it would be better for us to distinguish the differences between each method's performance after adding noise.

Sorry again for the misleading response. In this revision, we modify the starting of section 2.2:

"Since a real bulk dataset with its corresponding cell type fractions assessed by traditional experimental methods (e.g., flow cytometry) is rare, and it is hard to analyze how the batch effect would affect deconvolution performance, it is necessary to conduct a pseudo-bulk test for an initial estimation. The pseudo-bulk data are generated *in silico* from single-cell GEPs with ground truth (pre-defined cell type proportions). That is, pseudo-bulk data are the

summation of many single-cell profiles. To make this pseudo-bulk test as difficult as the real bulk test instead of a trivial linear regression task, we added Gaussian noise (0.01 times random value generated from a Gaussian distribution with gene expression mean and variance for each gene) and randomly masked 20% genes for each pseudo-bulk sample.”

11) Without looking at the source code, it is difficult to follow the training procedure of TAPE. It would be nice if the authors explain it more clearly.

Answer: Thank you for pointing out our unclearness in describing the training procedure of TAPE. To make things clear, we arrange the training procedure as the following pseudo-code. We also add it to section 4 of the manuscript.

Algorithm 1: Adaptive Training Procedure

input : Encoder parameters E and decoder parameters D from the initial training stage,
 GEPs of bulk RNA-seq B of size $n \times m$, step number α , max iteration β

output: signature matrix S of size $k \times m$,
 predicted fractions X of size $n \times k$,
 training loss L

```

1  $\tilde{S}_0, \tilde{X}_0 \leftarrow \text{model}(B)$ ;
2 for  $k \leftarrow 1$  to  $\beta$  do
3   for  $i \leftarrow 1$  to  $\alpha$  do
4      $\tilde{B}, X \leftarrow \text{model}(B)$ ;
5      $L \leftarrow \text{MAE}(\tilde{B}, B) + \text{MAE}(S, \tilde{S}_0)$ ;
6      $D \leftarrow D - \frac{\partial L}{\partial D}$ ;
7   end
8   for  $j \leftarrow 1$  to  $\alpha$  do
9      $\tilde{B}, S \leftarrow \text{model}(B)$ ;
10     $L \leftarrow \text{MAE}(\tilde{B}, B) + \text{MAE}(X, \tilde{X}_0)$ ;
11     $E \leftarrow E - \frac{\partial L}{\partial E}$ ;
12  end
13 end
14  $S, X \leftarrow \text{model}(B)$ ;

```

12) Since the output of the encoder does not sum to 1 and can be negative, the authors apply ReLU activation and normalize the result to sum to 1 using a scaling function.

However, from the code in `train.py` and `model.py` in

<https://github.com/poseidonchan/TAPE/tree/main/TAPE>, it seems that ReLU and scaling

functions are only used during prediction (and adaptive stage). Am I wrong? Are ReLU and

scaling functions always included in the model (i.e. in forward propagations)? If not, does it not violate entirely the assumption that the signature matrix is visible in the decoder since the proportions have to sum to 1 for $XS=B$ to be valid? Or is the decoder meant to represent the signature matrix only at the adaptive stage? This should be clearly stated in the manuscript.

Answer: Thank you very much for your detailed review! Your understanding of our code is right, ReLU and scaling function are not always included in the model. In the training stage, the summation of cell fractions of training data equals 1, and we think the model can learn the pattern if it is well-trained. But, considering the differences between training data and real bulk data, we also add the ReLU and scaling function to guarantee the prediction result is meaningful. Your concerns are right, $XS=B$ is not always valid during the training stage because we do not constrain the X. But we expect that $XS=B$ should be valid after the initial training with labeled pseudo-bulk data. Since the decoder is also learned from the training data, it can only be a signature matrix after the training stage, that is, the adaptive stage. Thanks again for pointing out the misleading part of our manuscript! We have added one more sentence in the model part:

(section 4.2.3) Of note, the decoder matrix is expected to represent a meaningful signature matrix only after the training with simulated data.

References

Domínguez Conde, C., Xu, C., Jarvis, L. B., Rainbow, D. B., Wells, S. B., Gomes, T., Howlett, S. K., Suchanek, O., Polanski, K., King, H. W., Mamanova, L., Huang, N., Szabo, P. A., Richardson, L., Bolt, L., Fasouli, E. S., Mahbubani, K. T., Prete, M., Tuck, L., ... Teichmann, S. A. (2022). Cross-tissue immune cell analysis reveals tissue-specific features in humans. *Science*, 376(6594).

<https://doi.org/10.1126/science.abl5197>

Johnson, W. E., Li, C., & Rabinovic, A. (2006). Adjusting batch effects in microarray expression data using empirical Bayes methods. *Biostatistics*, *8*(1), 118–127.

<https://doi.org/10.1093/biostatistics/kxj037>

Menden, K., Marouf, M., Oller, S., Dalmia, A., Magruder, D. S., Kloiber, K., Heutink, P., & Bonn, S. (2020). Deep learning–based cell composition analysis from tissue expression profiles. *Science Advances*, *6*(30). <https://doi.org/10.1126/sciadv.aba2619>

Newman, A. M., Steen, C. B., Liu, C. L., Gentles, A. J., Chaudhuri, A. A., Scherer, F., Khodadoust, M. S., Esfahani, M. S., Luca, B. A., Steiner, D., Diehn, M., & Alizadeh, A. A. (2019). Determining cell type abundance and expression from bulk tissues with digital cytometry. *Nature Biotechnology*, *37*(7), 773–782.

<https://doi.org/10.1038/s41587-019-0114-2>

Zhao, B., Erwin, A., & Xue, B. (2018). How many differentially expressed genes: A perspective from the comparison of genotypic and phenotypic distances. *Genomics*, *110*(1), 67–73. <https://doi.org/10.1016/j.ygeno.2017.08.007>

REVIEWER COMMENTS

Reviewer #1 (Remarks to the Author):

I would like to congratulate the authors for their very thorough revision. All my concerns were addressed.